# How do Minimum-Norm Shallow Denoisers Look in Function Space?

**Chen Zeno**
Electrical and Computer Engineering
Technion

**Greg Ongie**
Department Mathematical and Statistical Sciences
Marquette University

**Yaniv Blumenfeld, Nir Weinberger, Daniel Soudry**
Electrical and Computer Engineering
Technion

{chenzeno,yanivbl}@campus.technion.ac.il, gregory.ongie@marquette.edu
nirwein@technion.ac.il, daniel.soudry@gmail.com

## Abstract

Neural network (NN) denoisers are an essential building block in many common tasks, ranging from image reconstruction to image generation. However, the success of these models is not well understood from a theoretical perspective. In this paper, we aim to characterize the functions realized by shallow ReLU NN denoisers — in the common theoretical setting of interpolation (i.e., zero training loss) with a minimal representation cost (i.e., minimal $\ell^2$ norm weights). First, for univariate data, we derive a closed form for the NN denoiser function, find it is contractive toward the clean data points, and prove it generalizes better than the empirical MMSE estimator at a low noise level. Next, for multivariate data, we find the NN denoiser functions in a closed form under various geometric assumptions on the training data: data contained in a low-dimensional subspace, data contained in a union of one-sided rays, or several types of simplexes. These functions decompose into a sum of simple rank-one piecewise linear interpolations aligned with edges and/or faces connecting training samples. We empirically verify this alignment phenomenon on synthetic data and real images.

## 1 Introduction

The ability to reconstruct an image from a noisy observation has been studied extensively in the last decades, as it is useful for many practical applications (e.g., Hasinoff et al. [2010]). In recent years, Neural Network (NN) denoisers commonly replace classical expert-based approaches as they achieve substantially better results than the classical approaches (e.g., Zhang et al. [2017]). Beyond this natural usage, NN denoisers also serve as essential building blocks in a variety of common computer vision tasks, such as solving inverse problems [Zhang et al., 2021] and image generation [Song and Ermon, 2019, Ho et al., 2020]. To better understand the role of NN denoisers in such complex applications, we first wish to theoretically understand the type of solutions they converge to.

In practice, when training denoisers, we sample multiple noisy samples for each clean image and minimize the Mean Squared Error (MSE) loss for recovering the clean image. Since we sample numerous noisy samples per clean sample, the number of training samples is typically larger than the number of parameters in the network. Interestingly, even in such an under-parameterized regime, the loss has multiple global minima corresponding to distinct denoiser functions which achieve zero

37th Conference on Neural Information Processing Systems (NeurIPS 2023).

loss on the observed data. To characterize these functions, we study, similarly to previous works [Savarese et al., 2019, Ongie et al., 2020], the shallow NN solutions that interpolate the training data with minimal representation cost, i.e., where the $\ell^2$-norm of the weights (without biases and skip connections) is as small as possible. This is because we converge to such *min-cost* solutions when we minimize the loss with a vanishingly small $\ell^2$ regularization on these weights.

We first examine the univariate input case: building on existing results [Hanin, 2021], we characterize the min-cost interpolating solution and its generalization to unseen data. Next, we aim to extend this analysis to the multivariate case. However, this is challenging, since, to the best of our knowledge, there are no results that explicitly characterize these min-cost solutions for general multivariate shallow NNs — except in two basic cases. In the first case, the input data is co-linear [Ergen and Pilanci, 2021]. In the second case, the input samples are identical to their target outputs, so the trivial min-cost solution is the identity function. The NN denoisers' training regime is 'near' the second case: there, the input samples are noisy versions of the clean target outputs. Interestingly, we find that this regime leads to non-trivial min-cost solutions far from identity — even with an infinitesimally small input noise. We analytically investigate these solutions here.

**Our Contributions.** We study the NN solutions in the setting of interpolation of noisy samples with min-cost, in a practically relevant "low noise regime" where the noisy samples are well clustered. In the univariate case,

- We find a closed-form solution for the minimum representation cost NN denoiser. Then, we prove this solution generalizes better than the empirical minimum MSE (eMMSE) denoiser.
- We prove this min-cost NN solution is contractive toward the clean data points, that is, applying the denoiser necessarily reduces the distance of a noisy sample to one of the clean samples.

In the multivariate case,

- We derive a closed-form solution for the min-cost NN denoiser in multivariate case under various assumptions on the geometric configuration of the clean training samples. To the best of our knowledge, this is the first set of results to explicitly characterize a min-cost interpolating NN in a non-basic multivariate setting.
- We illustrate a general alignment phenomenon of min-cost NN denoisers in the multivariate setting: the optimal NN denoiser decomposes into a sum of simple rank-one piecewise linear interpolations aligned with edges and/or faces connecting clean training samples.

## 2 Preliminaries and problem setting

**The denoising problem.** Let $\boldsymbol{y} \in \mathbb{R}^d$ be a noisy observation of $\boldsymbol{x} \in \mathbb{R}^d$, such that $\boldsymbol{y} = \boldsymbol{x} + \boldsymbol{\epsilon}$ where $\boldsymbol{x}$ and $\boldsymbol{\epsilon}$ are statistically independent, and $\mathbb{E}[\boldsymbol{\epsilon}] = \boldsymbol{0}$. Commonly, this noise is Gaussian with covariance matrix $\sigma^2 \mathrm{I}$. The ultimate goal of a denoiser $\hat{\boldsymbol{x}}(\boldsymbol{y})$ is to minimize the MSE loss over the joint probability distribution of the data and the noisy observation ("population distribution"), i.e., to minimize

$$\mathcal{L}(\hat{\boldsymbol{x}}) = \mathbb{E}_{\boldsymbol{x},\boldsymbol{y}} \|\hat{\boldsymbol{x}}(\boldsymbol{y}) - \boldsymbol{x}\|^2 . \tag{1}$$

The well-known optimal solution for (1) is the minimum mean square error (MMSE) denoiser, i.e.,

$$\hat{\boldsymbol{x}}^*(\boldsymbol{y}) = \mathbb{E}_{\boldsymbol{x}|\boldsymbol{y}}[\boldsymbol{x} \mid \boldsymbol{y}] \in \arg\min_{\hat{\boldsymbol{x}}(\boldsymbol{y})} \mathbb{E}_{\boldsymbol{x},\boldsymbol{y}} \|\boldsymbol{x} - \hat{\boldsymbol{x}}(\boldsymbol{y})\|^2 . \tag{2}$$

Since we do not have access to the distribution of the data, and hence not to the posterior distribution, we rely on a finite amount of clean data $\{\boldsymbol{x}_n\}_{n=1}^N$ in order to learn a good approximation for the MMSE estimator. One approach is to assume an empirical data distribution and derive the optimal solution of (1), i.e., the empirical minimum mean square error (eMMSE) denoiser,

$$\hat{\boldsymbol{x}}^{\mathrm{eMMSE}}(\boldsymbol{y}) \in \arg\min_{\hat{\boldsymbol{x}}(\boldsymbol{y})} \frac{1}{N} \sum_{n=1}^N \mathbb{E}_{\boldsymbol{y}|\boldsymbol{x}_n} \|\hat{\boldsymbol{x}}(\boldsymbol{y}) - \boldsymbol{x}_n\|^2 . \tag{3}$$

If the noise is Gaussian with a covariance of $\sigma^2 \mathrm{I}$, an explicit solution to the eMMSE is given by

$$\hat{\boldsymbol{x}}^{\mathrm{eMMSE}}(\boldsymbol{y}) = \frac{\sum_{n=1}^N \boldsymbol{x}_n \exp\left(-\frac{\|\boldsymbol{y} - \boldsymbol{x}_n\|^2}{2\sigma^2}\right)}{\sum_{n=1}^N \exp\left(-\frac{\|\boldsymbol{y} - \boldsymbol{x}_n\|^2}{2\sigma^2}\right)} . \tag{4}$$

An alternative approach to computing the eMMSE directly is to draw $M$ noisy samples for each clean data point, as $\boldsymbol{y}_{n,m} = \boldsymbol{x}_n + \boldsymbol{\epsilon}_{n,m}$, where $\boldsymbol{\epsilon}_{n,m} \sim \mathcal{N}\left(\mathbf{0}, \sigma^2 \boldsymbol{I}\right)$ are independent and identically distributed, and to minimize the following loss function

$$\mathcal{L}_{\text{offline}, M}\left(\hat{\boldsymbol{x}}\right) = \frac{1}{MN} \sum_{m=1}^{M} \sum_{n=1}^{N} \|\hat{\boldsymbol{x}}\left(\boldsymbol{y}_{n,m}\right) - \boldsymbol{x}_n\|^2 \ . \tag{5}$$

**Denoiser model and algorithms.** In practice, we approximate the optimal denoiser $\hat{\boldsymbol{x}}\left(\boldsymbol{y}\right)$ using a parametric model $\boldsymbol{h}_{\boldsymbol{\theta}}\left(\boldsymbol{y}\right)$, typically a NN. We focus on a shallow ReLU network model with a skip connection of the form

$$\boldsymbol{h}_{\theta}(\boldsymbol{y}) = \sum_{k=1}^{K} \boldsymbol{a}_k [\boldsymbol{w}_k^\top \boldsymbol{y} + b_k]_+ + \boldsymbol{V}\boldsymbol{y} + \boldsymbol{c} \tag{6}$$

where $\theta = ((\theta_k)_{k=1}^{K}; \boldsymbol{c}, \boldsymbol{V})$ with $\theta_k = (b_k, \boldsymbol{a}_k, \boldsymbol{w}_k) \in \mathbb{R} \times \mathbb{R}^d \times \mathbb{R}^d$ and $\boldsymbol{c} \in \mathbb{R}^d, \boldsymbol{V} \in \mathbb{R}^{d \times d}$. We train the model on a finite set of clean data points $\{\boldsymbol{x}_n\}_{n=1}^{N}$. The common practical training method is based on an online approach. First, we sample a random batch (with replacement) from the data $\mathcal{B} \subseteq \{\boldsymbol{x}_n\}_{n=1}^{N}$. Then, for each clean data point $\boldsymbol{x}_n \in \mathcal{B}$, we draw a noisy sample $\boldsymbol{y}_n = \boldsymbol{x}_n + \boldsymbol{\epsilon}_n$, where $\boldsymbol{\epsilon}_n \sim \mathcal{N}\left(\mathbf{0}, \sigma^2 \mathrm{I}\right)$ are independent of the clean data points and other noise samples. At each iteration $t$ out of $T$ iterations, we update the model parameters according to a stochastic gradient descent rule, with a vanishingly small regularization term $\lambda C\left(\boldsymbol{\theta}\right)$, that is,

$$\boldsymbol{\theta}_{t+1} = \boldsymbol{\theta}_t - \eta \nabla_{\boldsymbol{\theta}_t} \frac{1}{|\mathcal{B}|} \sum_{n \in \mathcal{B}} \|\boldsymbol{h}_{\boldsymbol{\theta}_t}\left(\boldsymbol{y}_n\right) - \boldsymbol{x}_n\|^2 - \eta \lambda \nabla_{\boldsymbol{\theta}_t} C\left(\boldsymbol{\theta}_t\right) \ . \tag{7}$$

Another training method [Chen et al., 2014] is based on an offline approach. We sample $M$ noisy sample for each clean data point and minimize (5) plus a regularization term

$$\mathcal{L}_{\text{offline}, M}\left(\boldsymbol{\theta}\right) = \frac{1}{MN} \sum_{m=1}^{M} \sum_{n=1}^{N} \|\boldsymbol{h}_{\theta}\left(\boldsymbol{y}_{n,m}\right) - \boldsymbol{x}_n\|^2 + \lambda C\left(\boldsymbol{\theta}\right) \ . \tag{8}$$

Similarly to previous works [Savarese et al., 2019, Ongie et al., 2020], we assume an $\ell^2$ penalty on the weights, but not on the biases and skip connections, i.e.,

$$C(\theta) = \frac{1}{2} \sum_{k=1}^{K} \left(\|\boldsymbol{a}_k\|^2 + \|\boldsymbol{w}_k\|^2\right) \ . \tag{9}$$

**Low noise regime.** In this paper, we study the solution of the NN denoiser when the clusters of noisy samples around each clean point are well-separated, a setting which we refer to as the "low noise regime". This is a rather relevant regime since denoisers are practically used when the noise level is mild. Indeed, common image-denoising benchmarks test on low (but not negligible) noise levels. For instance, in the commonly used denoising benchmark BSD68 [Roth and Black, 2009], the noise level $\sigma = 0.1$ is in the low noise regime.[1] Moreover, this setting is important, for example, in diffusion-based image generation, since at the end of the reverse denoising process, new images are sampled by denoising smaller and smaller noise levels.[2]

## 3 Basic properties of neural network denoisers

**Offline v.s. online NN solutions.** NN denoisers are traditionally trained in an online fashion (7), using a finite amount of $T$ iterations. Consequently, only a finite number of noisy samples are used for each clean data point. We empirically observe that the solutions in the offline and online settings are similar. Specifically, in the univariate case, we show in Figure 1 that denoisers based on offline and

---

[1]The minimum distance between two images in BSD68 is about 97 while the image resolution is $d = 481 \times 321$. Also, the norm of the noise concentrates around the value of $\sqrt{d}\sigma \approx \sqrt{481 \cdot 321} \cdot 0.1 \approx 40 < 97$. Therefore, the clusters of noisy samples around each clean point are generally well-separated.

[2]Interestingly, it was suggested that the "useful" part of the diffusion dynamics happens only below some critical noise level [Raya and Ambrogioni, 2023].

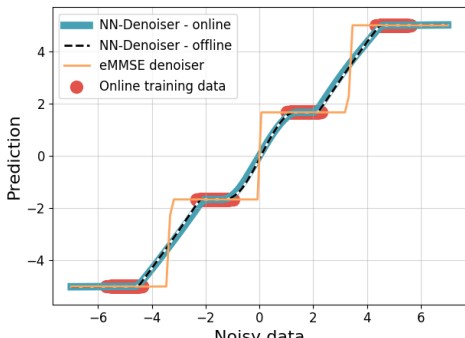

Figure 1: **NN denoiser vs eMMSE denoiser.** We trained a one-hidden-layer ReLU network with a skip connection on a denoising task. The clean dataset has four points equally spaced in the interval $[-5, 5]$, and the noisy samples are generated by adding zero-mean Gaussian noise with $\sigma = 1.5$. We use $\lambda = 10^{-5}$ in both setting. The Figure shows the denoiser output as a function of its input for: (1) NN denoiser trained online using (7) for $100K$ iterations, (2) NN denoiser trained offline using (8) with $M = 9000$ and $20K$ epochs, and (3) the eMMSE denoiser (4).

online loss functions converge to indistinguishable solutions. For the multivariate case, we observe (Figure 11 in Appendix D) that the offline and online solutions achieve approximately the same test MSE when trained on a subset of the MNIST dataset. The comparison is made using the same number of iterations for both training methods, while using much less noisy samples in the offline setting. Evidently, this lower number of samples does not significantly affect the generalization error. Hence, in the rest of the paper, we focus on offline training (i.e., minimizing the offline loss $\mathcal{L}_{\text{offline},M}$), as it defines an explicit loss function with solutions that can be theoretically analyzed, as in [Savarese et al., 2019, Ongie et al., 2020].

**The empirical MMSE denoiser.** The law of large numbers implies that the denoiser minimizing the offline loss $\mathcal{L}_{\text{offline},M}$ approaches the eMMSE estimator in the limit of infinitely many noisy samples,

$$\hat{\boldsymbol{x}}^{\text{eMMSE}}(\boldsymbol{y}) \in \arg\min_{\hat{\boldsymbol{x}}} \lim_{M \to \infty} \mathcal{L}_{\text{offline},M}(\hat{\boldsymbol{x}}) . \tag{10}$$

Therefore, it may seem that for a reasonable number of noise samples $M$, a large enough model, and small enough regularization, the denoiser we get by minimizing the offline loss (6) will also be similar to the eMMSE estimator. However, Figure 1 shows that the eMMSE solution and the NN solutions (both online and offline) are quite different. The eMMSE denoiser has a much sharper transition and maps almost all inputs to a value of a clean data point. This is because in the case of low noise the eMMSE denoiser (4) approximates the one nearest-neighbor (1-NN) classifier, i.e.,

$$\lim_{\sigma \to 0^+} \hat{\boldsymbol{x}}^{\text{eMMSE}}(\boldsymbol{y}) = \arg\min_{\boldsymbol{x} \in \{\boldsymbol{x}_i\}_{i=1}^N} \|\boldsymbol{y} - \boldsymbol{x}\| . \tag{11}$$

In contrast, the NN denoiser maps each noisy sample to its corresponding clean sample only in a limited "noise ball" around the clean point, and interpolates near-linearly between the "noise balls". Hence, we may expect that the smoother NN denoiser typically generalizes better than the eMMSE denoiser. We prove that this is indeed true for the univariate case in Section 4.

Why the NN denoiser does not converge to the eMMSE denoiser? Note that the limit in (10) is not practically relevant for the low-level noise regime, since we need an exponentially large $M$ in order to converge in this limit. For example, in the case of univariate Gaussian noise, we have that $P(|\epsilon| > t) \leq 2 \exp(-\frac{t^2}{2\sigma^2})$, $\forall t > 0$. Therefore, during training, we effectively observe only noisy samples that are in a bounded interval of size $2\sigma\sqrt{\log M}$ around each clean sample (see Figure 1). In other words, in the low-noise regime and for non-exponential $M$, there is no way to distinguish if the noise is sampled from some distribution with limited support of from a Gaussian distribution. The denoiser minimizing the loss with respect to a bounded-support distribution can be radically different from the eMMSE denoiser in the regions outside the "noise balls" surrounding the clean samples, where the denoiser function is not constrained by the loss. This leads to a large difference between the NN denoiser and the MMSE estimator.

Alternatively, one may suggest that the NN denoiser does not converge to the eMMSE denoiser due to an approximation error (i.e., the shallow NN's model capacity is too small to approximate the MMSE denoiser). Nevertheless, we provide empirical evidence indicating it is not the case. Specifically, recall that in the low noise regime, the eMMSE denoiser tends to the nearest-neighbor classifier, and

such a solution does not generalize well to test data. Thus, if the NN denoiser would have converged to the eMMSE solution, then its test error would have increased with the network size, in contrast to what we observe in Figure 12 (Appendix D).

Therefore, in order to approximate the eMMSE with a NN, it seems we must have an exponentially large $M$. Alternatively, we may converge to the eMMSE if we use a loss function marginalized over the Gaussian noise. This idea was previously suggested by Chen et al. [2014], with the goal of effectively increasing the number of noisy samples and thus improving the training performance of denoising autoencoders. Therein, this improvement was obtained by approximating the marginalized loss function by a Taylor series expansion. However, for shallow denoisers, we may actually obtain an explicit expression for this marginalized loss, without any approximation. Specifically, if we assume for simplicity, that the network does not have a linear unit ($\boldsymbol{V} = \boldsymbol{0}$) and its bias terms are zero ($\boldsymbol{c} = \boldsymbol{0}, b_k = 0$), then the marginalized loss for Gaussian noise, derived in Appendix A, is given by

$$\mathcal{L}(\boldsymbol{\theta}, \sigma) = \mathbb{E}_{\boldsymbol{x}, \boldsymbol{y}} \|\boldsymbol{h}_\theta(\boldsymbol{y}) - \boldsymbol{x}\|^2 = \mathbb{E}_{\boldsymbol{x}, \boldsymbol{y}} \left\| \sum_{k=1}^K \boldsymbol{a}_k [\boldsymbol{w}_k^\top \boldsymbol{y}]_+ - \boldsymbol{x} \right\|^2 = \mathbb{E}_{\boldsymbol{x}} \mathbb{E}_{\boldsymbol{y}|\boldsymbol{x}} \left\| \sum_{k=1}^K \boldsymbol{a}_i [\boldsymbol{w}_k^\top \boldsymbol{y}]_+ - \boldsymbol{x} \right\|^2$$

$$= \mathbb{E}_{\boldsymbol{x}} \left\| \sum_{k=1}^K \boldsymbol{a}_k \tilde{\phi} \left( \hat{\boldsymbol{w}}_k^\top \boldsymbol{x}, \|\boldsymbol{w}_k\|, \sigma \right) - \boldsymbol{x} \right\|^2 + \sum_{i=1}^K \sum_{j=1}^K \boldsymbol{a}_i^\top \boldsymbol{a}_j \mathbf{H}_{ij} \left( \boldsymbol{w}_i, \boldsymbol{w}_j, \sigma^2 \right) , \qquad (12)$$

where $\boldsymbol{w}_k = \hat{\boldsymbol{w}}_k \|\boldsymbol{w}_k\|$ and $\mathbf{H} \succeq 0$, $\tilde{\phi}(\cdot)$ are defined in Appendix A. NN denoisers trained over this loss function will thus tend to the eMMSE solution as the network size is increased. However, as we explained above, this is not necessarily desirable, so we only mention (12) to show exact marginalization is feasible.

**Regularization biases toward specific neural network denoisers.** To further explore the converged solution for offline training, we note that the offline loss function $\mathcal{L}_{\text{offline}, M}(\boldsymbol{\theta})$ allows the network to converge to a zero-loss solution. This is in contrast to online training for which each batch leads to new realizations of noisy samples, and thus the training error is never exactly zero. Specifically, consider the low noise regime (well-separated noisy clusters). Then, the network can perfectly fit all the noisy samples using a finite number of neurons (see Section 4 for a more accurate description in the univariate case). Importantly, there are multiple ways to cluster the noisy data points with such neurons, and so there are multiple global training loss minima that the network can achieve with zero loss, each with a different generalization capability. In contrast to the standard case considered in the literature, this holds even in the under-parameterized case (where $NM$, the total number of noisy samples, is larger than the number of parameters).

Since there are many minima that perfectly fit the training data, we converge to specific minima which also minimize the $\ell^2$ regularization we use (even though we assumed it is vanishing). Specifically, in the limit of vanishing regularization $C(\theta)$, the minimizers of $\mathcal{L}_{\text{offline}, M}(\boldsymbol{\theta})$ also minimize the representation cost.

**Definition 1.** *Let $\boldsymbol{h}_\theta : \mathbb{R}^d \to \mathbb{R}^d$ denote a shallow ReLU network of the form* (6). *For any function $\boldsymbol{f} : \mathbb{R}^d \to \mathbb{R}^d$ realizable as a shallow ReLU network, we define its **representation cost** as*

$$R(\boldsymbol{f}) = \inf_{\theta : \boldsymbol{f} = \boldsymbol{h}_\theta} C(\theta) = \inf_\theta \sum_{k=1}^K \|\boldsymbol{a}_k\| \text{ s.t. } \|\boldsymbol{w}_k\| = 1 \, \forall k, \boldsymbol{f} = \boldsymbol{h}_\theta , \qquad (13)$$

*and a **minimizer** of this cost, i.e. the 'min-cost' solution, as*

$$\boldsymbol{f}^* \in \operatorname*{argmin}_{\boldsymbol{f}} R(\boldsymbol{f}) \text{ s.t. } \boldsymbol{f}(\boldsymbol{y}_{n,m}) = \boldsymbol{x}_n \, \forall n, m , \qquad (14)$$

where the second equality in (13) holds due to the 1-homogeneity of the ReLU activation function, and since the bias terms are not regularized (see [Savarese et al., 2019, Appendix A]). In the next sections, we examine which function we obtain by minimizing the representation cost $R(\boldsymbol{f})$ in various settings.

## 4 Closed form solution for the NN denoiser function — univariate data

In this section, we prove that NN denoisers that minimize $R(f)$ for univariate data have the specific piecewise linear form observed in Figure 1, and we discuss the properties of this form. We observe $N$

clean univariate data points $\{x_n\}_{n=1}^{N}$, s.t. $-\infty < x_1 < x_2 < \cdots < x_N < \infty$, and $M$ noisy samples (drawn from some known distribution) for each clean data point, such that $y_{n,m} = x_n + \epsilon_{n,m}$. We denote by $\epsilon_n^{\max}$ the maximal noise seen for data point $x_n$, and by $\epsilon_n^{\min}$ the minimal noise seen for data point $x_n$, i.e.,

$$\epsilon_n^{\max} \equiv \max_m \epsilon_{n,m}, \quad \epsilon_n^{\min} \equiv \min_m \epsilon_{n,m}, \tag{15}$$

and assume the following,

**Assumption 1.** *Assume the data $\{x_n\}_{n=1}^{N}$ is well-separated after the addition of noise, i.e.,*

$$\forall n \in [N-1]: \ x_n + \epsilon_n^{\max} < x_{n+1} + \epsilon_{n+1}^{\min}, \tag{16}$$

*and $\epsilon_n^{\max} > 0$, $\epsilon_n^{\min} < 0$.*

So we can state the following,

**Proposition 1.** *For all datasets such that Assumption 1 holds, the unique minimizer of $R(f)$ is*

$$f_{1D}^*(y) = \begin{cases} x_1, & y < x_1 + \epsilon_1^{\min} \\ x_n, & x_n + \epsilon_n^{\min} \leq y \leq x_n + \epsilon_n^{\max} \\ \frac{x_{n+1}-x_n}{x_{n+1}+\epsilon_{n+1}^{\min}-(x_n+\epsilon_n^{\max})}\left(y - (x_n + \epsilon_n^{\max})\right) + x_n, & x_n + \epsilon_n^{\max} < y < x_{n+1} + \epsilon_{n+1}^{\min} \\ x_N, & y > x_N + \epsilon_N^{\max} \end{cases} . \tag{17}$$

The proof (which is based on Theorem 1.2. in [Hanin, 2021]) can be found in Appendix B.1. As can be seen from Figure 1, the empirical simulation matches Proposition 1. [3] Proposition 1 states that (17) is a closed-form solution for (8) with minimal representation cost. Notice that the *minimal number of neurons* needed to represent $f_{1D}^*$ using $h_\theta(y)$ is $2N - 2$, which is less than the number of the total training samples $NM$ for $M \geq 2$.

In the case of univariate data, we can prove that the representation cost minimizer $f_{1D}^*$ (linear interpolation) generalizes better than the optimal estimator over the empirical distribution (eMMSE) for low noise levels.

**Theorem 1.** *Let $y = x + \epsilon$ where $x \sim p_x(x)$ and $\epsilon \sim \mathcal{N}\left(0, \sigma^2\right)$ where $x$ and $\epsilon$ are statistically independent. Then for all datasets such that Assumption 1 holds, and for all density probability distributions $p_x(x)$ with bounded second moment such that $p_x(x) > 0$ for all $x \in [\min_n x_n, \max_n x_n]$, the following holds,*

$$\lim_{\sigma \to 0^+} \text{MSE}\left(\hat{x}^{\text{eMMSE}}(y)\right) > \lim_{\sigma \to 0^+} \text{MSE}\left(f_{1D}^*(y)\right) .$$

See Appendix B.2 for the proof. We may deduce from Theorem 1 that for each density probability distribution $p(x)$ there exists a critical noise level for which the the representation cost minimizer $f_{1D}^*$ has strictly lower MSE than the eMMSE for all smaller noise levels (this is because the MSE is a continuous function of $\sigma$). The critical noise level can change significantly depending on $p(x)$. For example, if $p(x)$ has a high "mass" in between the training points then the critical noise level is large. However, if the density function has a low "mass" between the training points then the critical noise level is small. In Appendix D we show the MSE vs. the noise level on MNIST denoiser for NN denoiser and eMMSE denoiser (Figure 13). As can be seen there, the critical noise level in this case is not small ($\sim 5$).

Intuitively, the difference between the NN denoiser and the eMMSE denoiser is how they operate on inputs that are not close to any of the clean samples (compared to the noise standard deviation). For such a point, the eMMSE denoiser does not take into account that the empirical distribution of the clean samples does not approximate well their true distribution. Thus, for small noise, it insists on "assigning" it to the closest clean sample point. By contrast, the NN denoiser generalizes better since it takes into account that, far from the clean samples, the data distribution is not well approximated by the empirical sample distribution. Thus, its operation there is near the identity function, with a small contraction toward the clean points, as we discuss next.

---

[3]Notice that the training points in Figure 1 are used in the online setting (7) and in the offline setting (8) we observe less noisy samples.

**Minimal norm leads to contractive solutions on univariate data.** Radhakrishnan et al. [2018] have empirically shown that Auto-Encoders (AE, i.e. NN denoisers without input noise), are *locally* contractive toward the training samples. Specifically, they showed that the clean dataset can be recovered when iterating the AE output multiple times until convergence. Additionally, they showed that, as we increase the width or the depth of the NN, the network becomes more contractive toward the training examples. In addition, Radhakrishnan et al. [2018] proved that 2-layer AE models are locally contractive under strong assumptions (the weights of the input layer are fixed and the number of neurons goes to infinity). Next, we prove that a univariate shallow NN denoiser is *globally* contractive toward the clean data points without using the assumptions used by Radhakrishnan et al. [2018] (i.e., the minimizer optimizes over both layers and has a finite number of neurons).

**Definition 2.** *We say that $\boldsymbol{f} : \mathbb{R}^d \to \mathbb{R}^d$ is contractive toward a set of points $\{\boldsymbol{x}_n\}_{n=1}^N$ on $\mathcal{Y} \subseteq \mathbb{R}^d$ if there exists a real number $0 \le \alpha < 1$ such that for any $\boldsymbol{y} \in \mathcal{Y}$ there exists $i \in [N]$ so that*

$$\|\boldsymbol{f}(\boldsymbol{y}) - \boldsymbol{f}(\boldsymbol{x}_i)\| \le \alpha \|\boldsymbol{y} - \boldsymbol{x}_i\| . \tag{18}$$

**Lemma 1.** $f_{1D}^*(y)$ *is contractive toward the clean training points $\{\boldsymbol{x}_n\}_{n=1}^N$ on $\mathcal{Y} = \mathbb{R} \setminus \cup_{n \in [N-1]} \left\{ \frac{x_{n+1}\epsilon_n^{\max} - x_n\epsilon_{n+1}^{\min}}{\epsilon_n^{\max} - \epsilon_{n+1}^{\min}} \right\}$.*

The proof can be found in Appendix B.3.

## 5 Minimal norm leads to alignment phenomenon on multivariate data

In the multivariate case, min-cost solutions of (14) are difficult to explicitly characterize. Even in the setting of fitting scalar-valued shallow ReLU networks, explicitly characterizing min-cost solutions under interpolation constraints remains an open problem, except in some basic cases (e.g., co-linear data [Ergen and Pilanci, 2021]).

As an approximation to (14) that is more mathematically tractable, we assume the functions being fit are constant and equal to $\boldsymbol{x}_n$ on a closed ball of radius $\rho$ centered at each $\boldsymbol{x}_n$, i.e., $\boldsymbol{f}(\boldsymbol{y}) = \boldsymbol{x}_n$ for all $\|\boldsymbol{y} - \boldsymbol{x}_n\| \le \rho$, such that the balls do not overlap. Letting $B(\boldsymbol{x}_n, \rho)$ denote the ball of radius $\rho$ centered at $\boldsymbol{x}_n$, we can write this constraint more compactly as $\boldsymbol{f}(B(\boldsymbol{x}_n, \rho)) = \{\boldsymbol{x}_n\}$. Consider minimizing the representation cost under this constraint:

$$\min_{\boldsymbol{f}} R(\boldsymbol{f}) \quad s.t. \quad \boldsymbol{f}(B(\boldsymbol{x}_n, \rho)) = \{\boldsymbol{x}_n\} \quad \forall n \in [N]. \tag{19}$$

However, even with this approximation, explicitly describing minimizers of (19) for an arbitrary collection of training samples remains challenging. Instead, to gain intuition, we describe minimizers of (19) assuming the training samples belong to simple geometric structures that yield explicit solutions. Our results reveal a general alignment phenomenon, such that the weights of the representation cost minimizer align themselves with edges and/or faces connecting data points. We also show that approximate solutions of (14) obtained numerically by training a NN denoiser with weight decay match well with the solutions of (19) having exact closed-form expressions.

### 5.1 Training data on a subspace

In the event that the clean training samples belong to a subspace, we show the representation cost minimizer depends only on the projection of the inputs onto the subspace containing the training data, and its output is also constrained to this subspace.

**Theorem 2.** *Assume the training samples $\{\boldsymbol{x}_n\}_{n=1}^N$ belong to a linear subspace $\mathcal{S} \subset \mathbb{R}^d$, and let $\boldsymbol{P}_{\mathcal{S}} \in \mathbb{R}^{d \times d}$ denote the orthogonal projector onto $\mathcal{S}$. Then any minimizer $\boldsymbol{f}^*$ of (19) satisfies $\boldsymbol{f}^*(\boldsymbol{y}) = \boldsymbol{P}_{\mathcal{S}}\boldsymbol{f}^*(\boldsymbol{P}_{\mathcal{S}}\boldsymbol{y})$ for all $\boldsymbol{y} \in \mathbb{R}^d$.*

The proof of this result and all others in this section is given in Appendix C.

Note the assumption that the dataset lies on a subpaces is practically relevant, since, in general, large datasets are (approximately) low rank, i.e., lie on a linear subspace [Udell and Townsend, 2019]. In Appendix D we also validated that common image datasets are (approximately) low rank (Table 1).

Specializing to the case co-linear training data (i.e., training samples belonging to a one-dimensional subspace) the min-cost solution is unique and is described by the following corollary:

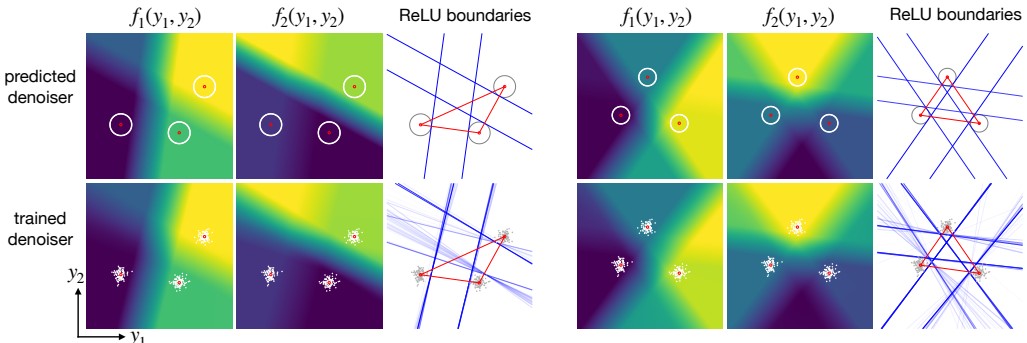

Figure 2: Predicted (top row) and empirical (bottom row) min-cost NN denoisers for $N = 3$ clean training samples in $d = 2$ dimensions. The emprical NN denoisers were trained with weight decay parameter $\lambda = 10^{-5}$ and $M = 100$ noisy samples. As predicted by our theory, the ReLU boundaries align either perpendicular to the triangle edges in the obtuse case (left panel), or parallel to the triangle edges (right panel).

**Corollary 1.** *Assume the training samples $\{\boldsymbol{x}_n\}_{n=1}^N$ are co-linear, i.e., $\boldsymbol{x}_n = c_n \boldsymbol{u}$ for some scalars $c_1 < c_2 < \cdots < c_n$ where $\boldsymbol{u} \in \mathbb{R}^d$ is a unit-vector. Then the minimizer $\boldsymbol{f}^*$ of (19) is unique and given by $\boldsymbol{f}^*(\boldsymbol{y}) = \boldsymbol{u}\phi(\boldsymbol{u}^\top \boldsymbol{y})$ where $\phi : \mathbb{R} \to \mathbb{R}$ has the same form as the 1-D minimizer (17) $f_{1D}^*$ with $x_n = c_n$ and $\epsilon_n^{\max} = -\epsilon_n^{\min} = \rho$.*

In other words, the min-cost solution has a particularly simple form in this case: $\boldsymbol{f}^*(\boldsymbol{y}) = \boldsymbol{u}\phi(\boldsymbol{u}^\top \boldsymbol{y})$, where $\phi$ is a monotonic piecewise linear function. We call any function of this form a *rank-one piecewise linear interpolator*. Below we show that many other min-cost solutions can be expressed as superpositions of rank-one piecewise linear interpolators.

## 5.2 Training data on rays

As an extension of the previous setting, we now consider training data belonging to a union of one-sided rays sharing a common origin. Assuming the rays are well-separated (in a sense made precise below) we prove that the representation cost minimizer decomposes into a sum of rank-one piecewise linear interpolators aligned with each ray.

**Theorem 3.** *Suppose the training samples $X$ belong to a union of $L$ rays plus a sample at the origin: $X = \{\boldsymbol{0}\} \cup \{\boldsymbol{x}_n^{(1)}\}_{n=1}^{N_1} \cup \cdots \cup \{\boldsymbol{x}_n^{(L)}\}_{n=1}^{N_L}$ where $\boldsymbol{x}_n^{(\ell)} = c_n^{(\ell)} \boldsymbol{u}_\ell$ for some unit vector $\boldsymbol{u}_\ell$ and constants $0 < c_1^{(\ell)} < c_2^{(\ell)} < \cdots < c_{N_\ell}^{(\ell)}$. Assume that the rays make obtuse angles with each other (i.e., $\boldsymbol{u}_\ell^\top \boldsymbol{u}_k < 0$ for all $\ell \neq k$). Then the minimizer $\boldsymbol{f}^*$ of (19) is unique and is given by*

$$\boldsymbol{f}^*(\boldsymbol{y}) = \boldsymbol{u}_1 \phi_1(\boldsymbol{u}_1^\top \boldsymbol{y}) + \cdots + \boldsymbol{u}_L \phi_L(\boldsymbol{u}_L^\top \boldsymbol{y}) \,, \tag{20}$$

*where $\phi_\ell : \mathbb{R} \to \mathbb{R}$ has the form of the 1-D minimizer (17) $f_{1D}^*$ with $x_n = c_n^{(\ell)}$, $\epsilon_n^{\max} = -\epsilon_n^{\min} = \rho$.*

Additionally, in the Appendix C.2.1 we show that this min-cost solution is stable with respect to small perturbations of the data. In particular, if the training data is perturbed from the rays, the functional form of the min-cost solution only changes slightly, such that the inner and outer-layer weight vectors align with the line segments connecting consecutive data points.

## 5.3 Special case: training data forming a simplex

Here, we study the representation cost minimizers for $N \leq d+1$ training points that form the vertices of a $(N-1)$-simplex, i.e., a $(N-1)$-dimensional simplex in $\mathbb{R}^d$ (e.g., a 2-simplex is a triangle, a 3-simplex is a tetrahedron, etc.). As we will show, the angles between vertices of the simplex (e.g., an acute versus obtuse triangle in $N = 3$) influences the functional form of the min-cost solution.

Our first result considers one extreme where the simplex has one vertex that makes an obtuse angle with *all* other vertices (e.g., an obtuse triangle for $N = 3$).

**Proposition 2.** *Suppose the convex hull of the training points $\{\boldsymbol{x}_1, \boldsymbol{x}_2, ..., \boldsymbol{x}_N\} \subset \mathbb{R}^d$ is a $(N-1)$-simplex such that $\boldsymbol{x}_1$ forms an obtuse angle with all other vertices, i.e., $(\boldsymbol{x}_j - \boldsymbol{x}_1)^\top(\boldsymbol{x}_i - \boldsymbol{x}_1) < 0$*

*for all $i \neq j$ with $i, j > 1$. Then the minimizer $\boldsymbol{f}^*$ of (19) is unique, and is given by*

$$\boldsymbol{f}^*(\boldsymbol{y}) = \boldsymbol{x}_1 + \sum_{n=2}^{N} \boldsymbol{u}_n \phi_n(\boldsymbol{u}_n^\top (\boldsymbol{y} - \boldsymbol{x}_1)) \tag{21}$$

*where $\boldsymbol{u}_n = \frac{\boldsymbol{x}_n - \boldsymbol{x}_1}{\|\boldsymbol{x}_n - \boldsymbol{x}_1\|}$, $\phi_n(t) = s_n([t - a_n]_+ - [t - b_n]_+)$, with $a_n = \rho$, $b_n = \|\boldsymbol{x}_n - \boldsymbol{x}_1\| - \rho$, and $s_n = \|\boldsymbol{x}_n - \boldsymbol{x}_1\|/(b_n - a_n)$ for all $n = 2, ..., N$.*

This result is essentially a corollary of Theorem 3, since after translating $\boldsymbol{x}_1$ to be the origin, the vertices of the simplex belong to rays making obtuse angles with each other, where there is exactly one sample per ray. Details of the proof are given in Appendix C.2.

At the opposite extreme, we consider the case where every vertex of the simplex is acute, meaning for all $n = 1, ..., N$ we have $(\boldsymbol{x}_i - \boldsymbol{x}_n)^\top (\boldsymbol{x}_j - \boldsymbol{x}_n) > 0$ for all $i, j \neq n$. In this case, we make following conjecture: the min-cost solution is instead a sum of $N$ rank-one piecewise linear interpolators, each aligned orthogonal to a different $(N - 2)$-dimensional face of the simplex.

**Conjecture 1.** *Suppose the convex hull of the training points $\{\boldsymbol{x}_1, \boldsymbol{x}_2, ..., \boldsymbol{x}_N\} \subset \mathbb{R}^d$ is a $(N - 1)$-simplex where every vertex of the simplex is acute. Then the minimizer $\boldsymbol{f}^*$ of (19) is unique, and is given by*

$$\boldsymbol{f}^*(\boldsymbol{y}) = \overline{\boldsymbol{x}} + \sum_{n=1}^{N} \boldsymbol{v}_n \phi_n(\boldsymbol{u}_n^\top (\boldsymbol{y} - \boldsymbol{z}_n)) \tag{22}$$

*where: $\boldsymbol{z}_n$ is the projection of $\boldsymbol{x}_n$ onto the unique $(N - 2)$-dimensional face of the simplex not containing $\boldsymbol{x}_n$; $\overline{\boldsymbol{x}}$ is the weighted geometric median of the vertices specified by*

$$\overline{\boldsymbol{x}} = \arg\min_{\boldsymbol{x} \in \mathbb{R}^d} \sum_{n=1}^{N} \frac{\|\boldsymbol{x}_n - \boldsymbol{x}\|}{\|\boldsymbol{x}_n - \boldsymbol{z}_n\|};$$

*$\boldsymbol{u}_n = \frac{\boldsymbol{x}_n - \boldsymbol{z}_n}{\|\boldsymbol{x}_n - \boldsymbol{z}_n\|}$, $\boldsymbol{v}_n = \frac{\boldsymbol{x}_n - \overline{\boldsymbol{x}}}{\|\boldsymbol{x}_n - \overline{\boldsymbol{x}}\|}$; and $\phi_n(t) = s_n([t - a_n]_+ - [t - b_n]_+)$ with $a_n = \rho$, $b_n = \|\boldsymbol{x}_n - \boldsymbol{z}_n\| - \rho$, and $s_n = \|\boldsymbol{x}_n - \overline{\boldsymbol{x}}\|/(b_n - a_n)$.*

Justification for this conjecture is given in Appendix C.3.2. In particular, we prove that the interpolator $\boldsymbol{f}^*$ given in (86) is a min-cost solution in the special case of three training points whose convex hull is an equilateral triangle. If true in general, this would imply a phase transition behavior in the min-cost solution when the simplex changes from having one obtuse vertex to all acute vertices, such that ReLU boundaries go from being aligned orthogonal to the edges connecting vertices, to being aligned parallel with the simplex faces. Figure 2 illustrates this for $N = 3$ training points forming a triangle in $d = 2$ dimensions. Moreover, Figure 2 shows that the empirical minimizer obtained using noisy samples and weight decay regularization agrees well with the form of the exact min-cost solution predicted by Proposition 2 and Conjecture 1.

In general, any given vertex of a simplex may make acute angles with some vertices and obtuse angles with others. This case is not covered by the above results. Currently, we do not have a conjectured form of the min-cost solution in this case, and we leave this as an open problem for future work.

## 6 Related works

Numerous methods have been proposed for image denoising. In last decade NN-based methods achieve state-of-the-art results [Zhang et al., 2017, 2021]. See [Elad et al., 2023] for a comprehensive review of image denoising. Sonthalia and Nadakuditi [2023] empirically showed a double decent behavior in NN denoisers, and theoretically proved it in a linear model. Similar to a denoiser, an Auto-Encoder (AE) is a NN model whose output dimension equals its input dimension, and is trained to match the output to the input. For AE, the typical goal is to learn an efficient lower-dimensional representation of the samples. Radhakrishnan et al. [2018] proved that a single hidden-layer AE that interpolates the training data (i.e., achieves zero loss), projects the input onto a nonlinear span of the training data. In addition, Radhakrishnan et al. [2018] empirically demonstrated that a multi-layer ReLU AE is locally contractive toward the training samples by iterating the AE and showing that the points converge to one of the training samples. Denoising autoencoders inject noise into the input

data in order to learn a good representation [Alain and Bengio, 2014]. The marginalized denoising autoencoder, proposed by Chen et al. [2014], approximates the marginalized loss over the noise (which is equivalent to observing infinitely many noisy samples) by using a Taylor approximation. Chen et al. [2014] demonstrated that by using the approximate marginalized loss we can achieve a substantial speedup in training and improved representation compared to standard denoising AE.

Many recent works aim to characterize function space properties of interpolating NN with minimal representation cost (i.e., min-cost solutions). Building off of the connection between weight decay and path-norm regularization identified in Neyshabur et al. [2015, 2017], Savarese et al. [2019] showed that the representation cost of a function realizable as a univariate two-layer ReLU network coincides with the $L^1$-norm of the second derivative of the function. Extensions to the multivariate setting were studied in Ongie et al. [2020], which identified the representation cost of a multivariate function with its $R$-norm, a Banach space semi-norm defined in terms of the Radon transform. Related work has extended the $R$-norm to other activation functions Parhi and Nowak [2021], vector-valued networks Shenouda et al. [2023], and deeper architectures Parhi and Nowak [2022]. A separate line of research studies min-cost solutions from a convex duality perspective Ergen and Pilanci [2021], incuding two-layer CNN denoising AEs Sahiner et al. [2021]. Recent work also studies properties of min-cost solutions in the case of arbitrarily deep NNs with ReLU activation Jacot [2022], Jacot et al. [2022].

Despite these advances in understanding min-cost solutions, there are few results explicitly characterizing their functional form. One exception is Hanin [2021], which gives a complete characterization of min-cost solutions in the case of shallow univariate ReLU networks with unregularized bias. This characterization is possible because the univariate representation cost is defined in terms of the 2nd derivative, which acts locally. Therefore, global minimizers can be found by minimizing the representation cost locally over intervals between data points. An extension of these results to the case of regularized bias is studied in Boursier and Flammarion [2023]. In the multivariate setting, the representation cost involves the Radon transform of the function – a highly non-local operation – that complicates the analysis. Parhi and Nowak [2021] prove a representer theorem showing that there always exists a min-cost solution realizable as a shallow ReLU network with finitely many neurons, and Ergen and Pilanci [2021] give an implicit characterization of min-cost NNs the solution to a convex optimization problem, and give explicit solutions in the case of co-linear training features. However, to the best of our knowledge, there are no results explicitly characterizing min-cost solutions in the case of non-colinear multivariate inputs, even for networks having scalar outputs.

## 7  Discussions

**Conclusions.**  We have explored the elementary properties of NN solutions for the denoising problem, while focusing on offline training of a one hidden-layer ReLU network. When the noisy clusters of the data samples are well-separated, there are multiple networks with zero loss, even in the case of under-parameterization, while having a different representation cost. In contrast, previous theoretical works focused on the over-parametrized regime. In the univariate case, we have derived a closed-form solution to such global minima with minimum representation cost. We also showed that the univariate NN solution generalizes better than the eMMSE denoiser. In the multivariate case, we showed that the interpolating solution with minimal representation cost is aligned with the edges and/or faces connecting the clean data points in several cases.

**Limitations.**  One limitation of our analysis in the multivariate case is that we assume the denoiser interpolates data on a full $d$-dimensional ball centered at each clean training sample, where $d$ is the input dimension. In practical settings, often the number of noisy samples $M \ll d$. A more accurate model would be to assume that denoiser interpolates over an $(M-1)$-dimensional disc centered at each training sample. This model may still be a tractable alternative to assuming interpolation of finitely many noisy samples. Also, our results relate to NN denoisers trained with explicit weight decay regularization, which is not always used in practice. However, recent work shows that stable minimizers of SGD must have low representation cost Mulayoff et al. [2021], Nacson et al. [2023], and so some of our analysis may provide insight for unregularized training, as well. Finally, for mathematical tractability, we focussed on the case of fully-connected ReLU networks with one hidden-layer. Extending our analysis to deeper architectures and convolutional neural networks is an important direction for future work.

## Acknowledgments

We thank Itay Hubara for his technical advice and valuable comments on the manuscript. The research of DS was Funded by the European Union (ERC, A-B-C-Deep, 101039436). Views and opinions expressed are however those of the author only and do not necessarily reflect those of the European Union or the European Research Council Executive Agency (ERCEA). Neither the European Union nor the granting authority can be held responsible for them. DS also acknowledges the support of Schmidt Career Advancement Chair in AI. The research of NW was supported by the Israel Science Foundation (ISF), grant no. 1782/22. GO was supported by the National Science Foundation (NSF) CRII award CCF-2153371.

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

# A  Marginalized loss

In this section, we derive the marginal loss for the case of a 1-hidden layer ReLU neural network. The loss function is,

$$\mathcal{L}\left(\boldsymbol{\theta}, \sigma\right) = \mathbb{E}_{\boldsymbol{x},\boldsymbol{y}} \left\| \sum_{k=1}^{K} \boldsymbol{a}_k [\boldsymbol{w}_k^\top \boldsymbol{y}]_+ - \boldsymbol{x} \right\|^2$$

$$= \mathbb{E}_{\boldsymbol{x},\boldsymbol{y}} \left[ \sum_{i=1}^{K}\sum_{j=1}^{K} \boldsymbol{a}_i^\top \boldsymbol{a}_j [\boldsymbol{w}_i^\top \boldsymbol{y}]_+ [\boldsymbol{w}_j^\top \boldsymbol{y}]_+ - 2 \sum_{i=1}^{K} \boldsymbol{x}^\top \boldsymbol{a}_i [\boldsymbol{w}_i^\top \boldsymbol{y}]_+ + \|\boldsymbol{x}\|^2 \right]$$

$$= \sum_{i=1}^{K}\sum_{j=1}^{K} \boldsymbol{a}_i^\top \boldsymbol{a}_j \mathbb{E}_{\boldsymbol{x},\boldsymbol{y}} \left[ [\boldsymbol{w}_i^\top \boldsymbol{y}]_+ [\boldsymbol{w}_j^\top \boldsymbol{y}]_+ \right] - 2 \sum_{i=1}^{K} \mathbb{E}_{\boldsymbol{x},\boldsymbol{y}} \left[ \boldsymbol{x}^\top \boldsymbol{a}_i [\boldsymbol{w}_i^\top \boldsymbol{y}]_+ \right] + \mathbb{E}\|\boldsymbol{x}\|^2$$

$$= \sum_{i=1}^{K}\sum_{j=1}^{K} \boldsymbol{a}_i^\top \boldsymbol{a}_j \mathbb{E}_{\boldsymbol{x}} \left[ \mathbb{E}_{\boldsymbol{y}|\boldsymbol{x}} \left[ [\boldsymbol{w}_i^\top \boldsymbol{y}]_+ [\boldsymbol{w}_j^\top \boldsymbol{y}]_+ \right] \right] - 2 \sum_{i=1}^{K} \mathbb{E}_{\boldsymbol{x}} \left[ \mathbb{E}_{\boldsymbol{y}|\boldsymbol{x}} \left[ \boldsymbol{x}^\top \boldsymbol{a}_i [\boldsymbol{w}_i^\top \boldsymbol{y}]_+ \right] \right] + \mathbb{E}\|\boldsymbol{x}\|^2$$

$$= \sum_{i=1}^{K}\sum_{j=1}^{K} \boldsymbol{a}_i^\top \boldsymbol{a}_j \mathbb{E}_{\boldsymbol{x}} \left[ \mathbb{E}_{\boldsymbol{y}|\boldsymbol{x}} \left[ [\boldsymbol{w}_i^\top \boldsymbol{y}]_+ \right] \mathbb{E}_{\boldsymbol{y}|\boldsymbol{x}} \left[ [\boldsymbol{w}_j^\top \boldsymbol{y}]_+ \right] \right] - 2 \sum_{i=1}^{K} \mathbb{E}_{\boldsymbol{x}} \left[ \mathbb{E}_{\boldsymbol{y}|\boldsymbol{x}} \left[ \boldsymbol{x}^\top \boldsymbol{a}_i [\boldsymbol{w}_i^\top \boldsymbol{y}]_+ \right] \right] + \mathbb{E}\|\boldsymbol{x}\|^2$$

$$+ \sum_{i=1}^{K}\sum_{j=1}^{K} \boldsymbol{a}_i^\top \boldsymbol{a}_j \mathbb{E}_{\boldsymbol{x}} \left[ \mathbb{E}_{\boldsymbol{y}|\boldsymbol{x}} \left[ [\boldsymbol{w}_i^\top \boldsymbol{y}]_+ [\boldsymbol{w}_j^\top \boldsymbol{y}]_+ \right] \right] - \sum_{i=1}^{K}\sum_{j=1}^{K} \boldsymbol{a}_i^\top \boldsymbol{a}_j \mathbb{E}_{\boldsymbol{x}} \left[ \mathbb{E}_{\boldsymbol{y}|\boldsymbol{x}} \left[ [\boldsymbol{w}_i^\top \boldsymbol{y}]_+ \right] \mathbb{E}_{\boldsymbol{y}|\boldsymbol{x}} \left[ [\boldsymbol{w}_j^\top \boldsymbol{y}]_+ \right] \right]$$

$$= \mathbb{E}_{(\boldsymbol{x})} \left\| \sum_{i=1}^{K} \boldsymbol{a}_i \tilde{\phi}\left(\boldsymbol{w}_i^\top \boldsymbol{x}\right) - \boldsymbol{x} \right\|^2 + \sum_{i=1}^{K}\sum_{j=1}^{K} \boldsymbol{a}_i^\top \boldsymbol{a}_j \mathbb{E}_{\boldsymbol{x}} \left[ \mathbb{E}_{\boldsymbol{y}|\boldsymbol{x}} \left[ [\boldsymbol{w}_i^\top \boldsymbol{y}]_+ [\boldsymbol{w}_j^\top \boldsymbol{y}]_+ \right] \right]$$

$$- \sum_{i=1}^{K}\sum_{j=1}^{K} \boldsymbol{a}_i^\top \boldsymbol{a}_j \mathbb{E}_{\boldsymbol{x}} \left[ \mathbb{E}_{\boldsymbol{y}|\boldsymbol{x}} \left[ [\boldsymbol{w}_i^\top \boldsymbol{y}]_+ \right] \mathbb{E}_{\boldsymbol{y}|\boldsymbol{x}} \left[ [\boldsymbol{w}_j^\top \boldsymbol{y}]_+ \right] \right] .$$

We denote by

$$\mathbf{H}_{ij}\left(\boldsymbol{w}_i, \boldsymbol{w}_j, \sigma^2\right) = \mathbb{E}_{\boldsymbol{x}} \left[ \mathbb{E}_{\boldsymbol{y}|\boldsymbol{x}} \left[ [\boldsymbol{w}_i^\top \boldsymbol{y}]_+ [\boldsymbol{w}_j^\top \boldsymbol{y}]_+ \right] - \mathbb{E}_{\boldsymbol{y}|\boldsymbol{x}} \left[ [\boldsymbol{w}_i^\top \boldsymbol{y}]_+ \right] \mathbb{E}_{\boldsymbol{y}|\boldsymbol{x}} \left[ [\boldsymbol{w}_j^\top \boldsymbol{y}]_+ \right] \right] .$$

Note that $\mathbf{H} \succeq 0$ since $\mathbf{H}$ is a covariance matrix. Thus we get,

$$\mathcal{L}\left(\boldsymbol{\theta}, \sigma\right) = \mathbb{E}_{\boldsymbol{x}} \left\| \sum_{k=1}^{K} \boldsymbol{a}_k \tilde{\phi}\left(\hat{\boldsymbol{w}}_k^\top \boldsymbol{x}, \|\boldsymbol{w}_k\|, \sigma\right) - \boldsymbol{x} \right\|^2 + \sum_{i=1}^{K}\sum_{j=1}^{K} \boldsymbol{a}_i^\top \boldsymbol{a}_j \mathbf{H}_{ij}\left(\boldsymbol{w}_i, \boldsymbol{w}_j, \sigma^2\right) .$$

**Lemma 2.** *In the case of the ReLU activation function and Gaussian noise the following holds,*

$$\tilde{\phi}\left(\hat{\boldsymbol{w}}_i^\top \boldsymbol{x}, \|\boldsymbol{w}_i\|, \sigma\right) = \|\boldsymbol{w}_i\| \left( \left( 1 - \Phi\left(-\frac{\hat{\boldsymbol{w}}_i^\top \boldsymbol{x}}{\sigma}\right) \right) \hat{\boldsymbol{w}}_i^\top \boldsymbol{x} + \varphi\left(-\frac{\hat{\boldsymbol{w}}_i^\top \boldsymbol{x}}{\sigma}\right) \right)$$

*where $\varphi, \Phi$ are the density and cumulative distribution of standard normal distribution, and*

$$\mathbf{H}_{ij} = \mathbb{E}_{\boldsymbol{x}} \left[ \Psi\left(\boldsymbol{x}, \boldsymbol{w}_i, \boldsymbol{w}_j, \sigma^2\right) - \tilde{\phi}\left(\hat{\boldsymbol{w}}_i^\top \boldsymbol{x}, \|\boldsymbol{w}_i\|, \sigma\right) \tilde{\phi}\left(\hat{\boldsymbol{w}}_j^\top \boldsymbol{x}, \|\boldsymbol{w}_j\|, \sigma\right) \right]$$

*where,*

$$\Psi\left(\boldsymbol{x}, \boldsymbol{w}_i, \boldsymbol{w}_j, \sigma^2\right) = P\left(z_1 > 0, z_2 > 0; \boldsymbol{\mu}^{(i,j)}, \boldsymbol{\Sigma}^{(i,j)}\right) \left( \sigma_{12} + \det\left(\boldsymbol{\Sigma}\right) \frac{f\left(-\boldsymbol{\mu}^{(i,j)}; \boldsymbol{0}, \boldsymbol{\Sigma}^{(i,j)}\right)}{P(z_1 > -\mu_1, z_2 > -\mu_2; \boldsymbol{0}, \boldsymbol{\Sigma}^{(i,j)})} \right.$$

$$\left. + \sigma_{11} F_1\left(-\mu_1\right) \mu_2 + \mu_1 \sigma_{22} F_2\left(-\mu_2\right) + \mu_1 \mu_2 \right)$$

$$F_2\left(-\mu_2\right) = \frac{\varphi\left(-\frac{\mu_2}{\sqrt{\sigma_{22}}}\right) P\left(z > -\mu_1; \frac{\Lambda_{12}\mu_2}{\Lambda_{11}}, \frac{1}{\Lambda_{11}}\right)}{P(z_1 > -\mu_1, z_2 > -\mu_2; \boldsymbol{0}, \boldsymbol{\Sigma})}$$

$$F_1\left(-\mu_1\right) = \frac{\varphi\left(-\frac{\mu_1}{\sqrt{\sigma_{11}}}\right) P\left(z > -\mu_2; \frac{\Lambda_{21}\mu_1}{\Lambda_{22}}, \frac{1}{\Lambda_{22}}\right)}{P(z_1 > -\mu_1, z_2 > -\mu_2; \boldsymbol{0}, \boldsymbol{\Sigma})}$$

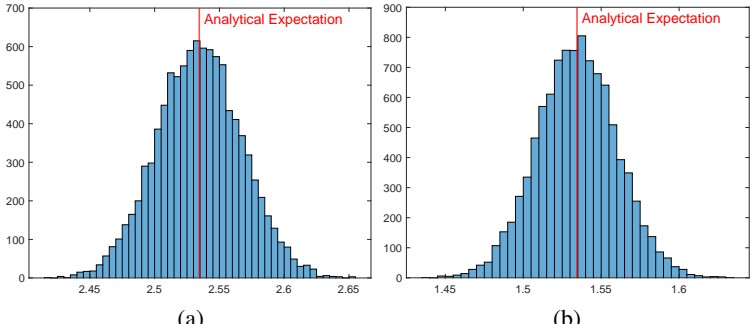

Figure 3: **Numerical evaluation of** (23) Histogram of the sample average of $\mathrm{ReLU}(x)$ for $10000$ Monte-Carlo samples. We denote by $E$ the analytical expectation and by $\bar{E}$ the sample average. Figure (a) is for $\mu = 1, \sigma = 5$, the normalized error is $\frac{|E - \bar{E}|}{E} = 0.0032\%$. Figure (b) is for $\mu = -1, \sigma = 5$, the normalized error is $\frac{|E - \bar{E}|}{E} = 0.0059\%$.

*and*

$$\boldsymbol{\mu}^{(i,j)} = \begin{pmatrix} \mu_1 \\ \mu_2 \end{pmatrix} = \begin{pmatrix} \boldsymbol{w}_i^\top \boldsymbol{x} \\ \boldsymbol{w}_j^\top \boldsymbol{x} \end{pmatrix}$$

$$\boldsymbol{\Sigma}^{(i,j)} = \begin{pmatrix} \sigma_{11} & \sigma_{12} \\ \sigma_{12} & \sigma_{22} \end{pmatrix} = \begin{pmatrix} \sigma^2 \|\boldsymbol{w}_i\|^2 & \sigma^2 \boldsymbol{w}_i^\top \boldsymbol{w}_j \\ \sigma^2 \boldsymbol{w}_j^\top \boldsymbol{w}_i & \sigma^2 \|\boldsymbol{w}_j\|^2 \end{pmatrix}$$

$$\boldsymbol{\Lambda} = \boldsymbol{\Sigma}^{-1}.$$

*Notice that we denote by $P\left(z_1 > 0, z_2 > 0; \boldsymbol{\mu}^{(i,j)}, \boldsymbol{\Sigma}^{(i,j)}\right)$ the probability that $z_1 > 0, z_2 > 0$ where $(z_1, z_2)^\top \sim \mathcal{N}\left(\boldsymbol{\mu}^{(i,j)}, \boldsymbol{\Sigma}^{(i,j)}\right)$.*

*Proof.* Let $x \sim \mathcal{N}\left(\mu, \sigma^2\right)$, then

$$E\left[[x]_+\right] = E\left[[x]_+ | x > 0\right] P\left(x > 0\right) + E\left[[x]_+ | x < 0\right] P\left(x > 0\right)$$
$$= E\left[x | x > 0\right] P\left(x > 0\right)$$
$$= E\left[x | x > 0\right] \left(1 - \Phi\left(-\frac{\mu}{\sigma}\right)\right)$$

Note that given $x > 0$ the distribution of $x$ is truncated normal [Horrace, 2015]. Therefore,

$$\mathbb{E}\left[x | x > 0\right] = \mu + \sigma \frac{\varphi\left(-\frac{\mu}{\sigma}\right)}{1 - \Phi\left(-\frac{\mu}{\sigma}\right)}$$

$$\mathbb{E}\left[[x]_+\right] = \left(1 - \Phi\left(-\frac{\mu}{\sigma}\right)\right) \mu + \sigma \varphi\left(-\frac{\mu}{\sigma}\right). \tag{23}$$

Note that given $\boldsymbol{x}$, $\boldsymbol{y}$ is a Gaussian random vector. Therefore, given $\boldsymbol{x}$,

$$\boldsymbol{w}_k^\top \boldsymbol{y} \sim \mathcal{N}\left(\boldsymbol{w}_k^\top \boldsymbol{x}, \sigma^2 \|\boldsymbol{w}_k\|^2\right)$$

and we obtain,

$$\mathbb{E}_{\boldsymbol{y}|\boldsymbol{x}}\left[[\boldsymbol{w}_k^\top \boldsymbol{y}]_+\right] = \|\boldsymbol{w}_k\| \mathbb{E}_{\boldsymbol{y}|\boldsymbol{x}}\left[[\hat{\boldsymbol{w}}_k^\top \boldsymbol{y}]_+\right]$$

$$= \|\boldsymbol{w}_k\| \left(1 - \Phi\left(-\frac{\hat{\boldsymbol{w}}_k^\top \boldsymbol{x}}{\sigma}\right)\right) \hat{\boldsymbol{w}}_k^\top \boldsymbol{x} + \sigma \varphi\left(-\frac{\hat{\boldsymbol{w}}_i^\top \boldsymbol{x}}{\sigma}\right).$$

Let $(z_1, z_2)^\top$ be a Gaussian random vector with[4]

$$\boldsymbol{\mu} = \begin{pmatrix} \mu_1 \\ \mu_2 \end{pmatrix}$$

$$\boldsymbol{\Sigma} = \begin{pmatrix} \sigma_{11} & \sigma_{12} \\ \sigma_{12} & \sigma_{22} \end{pmatrix}$$

---

[4]Note that $\sigma_{ii} = \sigma_i^2$.

then,
$$\mathbb{E}\left[[z_1]_+[z_2]_+\right] = \mathbb{E}\left[[z_1]_+[z_2]_+|z_1 > 0, z_2 > 0\right] P\left(z_1 > 0, z_2 > 0\right)$$
$$= \mathbb{E}\left[z_1 z_2 | z_1 > 0, z_2 > 0\right] P\left(z_1 > 0, z_2 > 0\right).$$

Given $z_1 > 0, z_2 > 0$ the distribution of $(z_1, z_2)^\top$ is truncated multivariate normal distribution [Manjunath and Wilhelm, 2021], therefore,

$$\mathbb{E}\left[z_1 z_2 | z_1 > 0, z_2 > 0\right] = \sigma_{12} + \sigma_{21}\left(-\mu_1\right) F_1\left(-\mu_1\right) + \sigma_{12}\left(-\mu_2\right) F_2\left(-\mu_2\right)$$

$$+ \left(\sigma_{11}\sigma_{22} - \sigma_{12}\sigma_{21}\right)\frac{f\left(-\boldsymbol{\mu}; \mathbf{0}, \boldsymbol{\Sigma}\right)}{p(z_1 > -\mu_1, z_2 > -\mu_2; \mathbf{0}, \boldsymbol{\Sigma})}$$

$$- \left(\sigma_{11} F_1\left(-\mu_1\right) + \sigma_{12} F_2\left(-\mu_2\right)\right)\left(\sigma_{21} F_1\left(-\mu_1\right) + \sigma_{22} F_2\left(-\mu_2\right)\right)$$

$$+ \left(\sigma_{11} F_1\left(-\mu_1\right) + \sigma_{12} F_2\left(-\mu_2\right) + \mu_1\right)\left(\sigma_{21} F_1\left(-\mu_1\right) + \sigma_{22} F_2\left(-\mu_2\right) + \mu_2\right)$$

$$= \sigma_{12} + \det\left(\boldsymbol{\Sigma}\right)\frac{f\left(-\boldsymbol{\mu}; \mathbf{0}, \boldsymbol{\Sigma}\right)}{p(z_1 > -\mu_1, z_2 > -\mu_2; \mathbf{0}, \boldsymbol{\Sigma})}$$

$$+ \sigma_{11} F_1\left(-\mu_1\right)\mu_2 + \mu_1\sigma_{22} F_2\left(-\mu_2\right) + \mu_1\mu_2 \qquad (24)$$

where $f\left(-\boldsymbol{\mu}; \mathbf{0}, \boldsymbol{\Sigma}\right)$ is a density function of of Gaussian random vector with mean vector $\mathbf{0}$ and covariance matrix $\boldsymbol{\Sigma}$ at the point $-\boldsymbol{\mu}$, and

$$F_2\left(z_2\right) = \frac{\int_{-\mu_1}^{\infty} f\left(z_1, z_2; \mathbf{0}, \boldsymbol{\Sigma}\right) dz_1}{P(Z_1 > -\mu_1, Z_2 > -\mu_2; \mathbf{0}, \boldsymbol{\Sigma})}$$

$$F_1\left(z_1\right) = \frac{\int_{-\mu_2}^{\infty} f\left(z_1, z_2; \mathbf{0}, \boldsymbol{\Sigma}\right) dz_2}{P(Z_1 > -\mu_1, Z_2 > -\mu_2; \mathbf{0}, \boldsymbol{\Sigma})}.$$

We denote by $\boldsymbol{\Lambda} = \boldsymbol{\Sigma}^{-1}$ thus,

$$\int_{-\mu_1}^{\infty} \exp\left(-\frac{1}{2}\left(\Lambda_{11} z_1^2 + 2\Lambda_{12} z_1 z_2 + \Lambda_{22} z_2^2\right)\right) dz_1 =$$

$$\exp\left(-\frac{1}{2}\Lambda_{22} z_2^2\right)\int_{-\mu_1}^{\infty} \exp\left(-\frac{1}{2}\Lambda_{11} z_1^2 - \Lambda_{12} z_1 z_2\right) dz_1 =$$

$$\exp\left(-\frac{1}{2}\Lambda_{22} z_2^2\right)\int_{-\mu_1}^{\infty} \exp\left(-\frac{1}{2\frac{1}{\Lambda_{11}}}\left(z_1 + \frac{\Lambda_{12} z_2}{\Lambda_{11}}\right)^2 + \frac{\Lambda_{12}^2 z_2^2}{2\Lambda_{11}}\right) dz_1 =$$

$$\exp\left(-\frac{1}{2}\Lambda_{22} z_2^2 + \frac{\Lambda_{12}^2 z_2^2}{2\Lambda_{11}}\right)\int_{-\mu_1}^{\infty} \exp\left(-\frac{1}{2\frac{1}{\Lambda_{11}}}\left(z_1 + \frac{\Lambda_{12} z_2}{\Lambda_{11}}\right)^2\right) dz_1 =$$

$$\exp\left(-\frac{1}{2}\Lambda_{22} z_2^2 + \frac{\Lambda_{12}^2 z_2^2}{2\Lambda_{11}}\right)\sqrt{\frac{2\pi}{\Lambda_{11}}} P\left(z > -\mu_1; -\frac{\Lambda_{12} z_2}{\Lambda_{11}}, \frac{1}{\Lambda_{11}}\right).$$

Thus,

$$\int_{-\mu_1}^{\infty} f\left(z_1, z_2; 0, \boldsymbol{\Sigma}\right) dz_1 = \frac{1}{2\pi\sqrt{\det\left(\boldsymbol{\Sigma}\right)}}\exp\left(-\frac{1}{2}\Lambda_{22} z_2^2 + \frac{\Lambda_{12}^2 z_2^2}{2\Lambda_{11}}\right)\sqrt{\frac{2\pi}{\Lambda_{11}}} P\left(z > -\mu_1; -\frac{\Lambda_{12} z_2}{\Lambda_{11}}, \frac{1}{\Lambda_{11}}\right)$$

$$= \frac{1}{\sqrt{2\pi\sigma_{22}}}\exp\left(-\frac{1}{2\sigma_{22}} z_2^2\right) P\left(z > -\mu_1; -\frac{\Lambda_{12} z_2}{\Lambda_{11}}, \frac{1}{\Lambda_{11}}\right).$$

Similarly, we obtain,

$$\int_{-\mu_2}^{\infty} f\left(z_1, z_2; 0, \boldsymbol{\Sigma}\right) dz_2 = \frac{1}{\sqrt{2\pi\sigma_{11}}}\exp\left(-\frac{1}{2\sigma_{11}} z_1^2\right) P\left(z > -\mu_2; -\frac{\Lambda_{21} z_1}{\Lambda_{22}}, \frac{1}{\Lambda_{22}}\right).$$

Therefore,

$$F_2\left(z_2\right) = \frac{\varphi\left(\frac{z_2}{\sqrt{\sigma_{22}}}\right) P\left(z > -\mu_1; -\frac{\Lambda_{12} z_2}{\Lambda_{11}}, \frac{1}{\Lambda_{11}}\right)}{P(Z_1 > -\mu_1, Z_2 > -\mu_2; \mathbf{0}, \boldsymbol{\Sigma})}$$

$$F_1\left(z_1\right) = \frac{\varphi\left(\frac{z_1}{\sqrt{\sigma_{11}}}\right) P\left(z > -\mu_2; -\frac{\Lambda_{21} z_1}{\Lambda_{22}}, \frac{1}{\Lambda_{22}}\right)}{P(Z_1 > -\mu_1, Z_2 > -\mu_2; \mathbf{0}, \boldsymbol{\Sigma})}.$$

$\square$

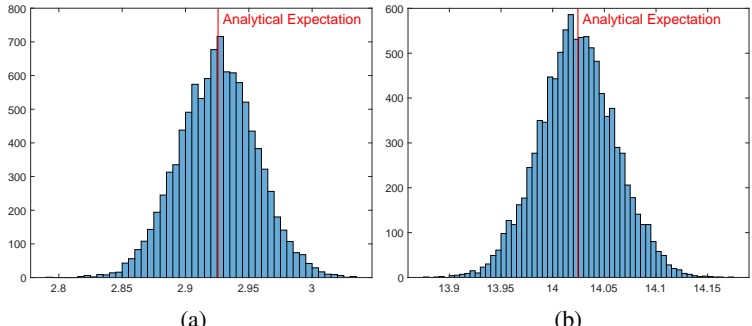

(a)                                              (b)

Figure 4: **Numerical evaluation of** (24) Histogram of the sample average of $\mathrm{ReLU}(z_1)\mathrm{ReLU}(z_2)$ for 10000 Monte-Carlo samples. We denote by $E$ the analytical expectation and by $\bar{E}$ the sample average. Figure (a) is for $\boldsymbol{\mu} = \begin{pmatrix} -4 \\ 17 \end{pmatrix}, \boldsymbol{\Sigma} = \begin{pmatrix} 13 & -9 \\ -9 & 8 \end{pmatrix}$, the normalized error is $\frac{|E - \bar{E}|}{E} = 0.0093\%$. Figure (b) is for $\boldsymbol{\mu} = \begin{pmatrix} 6 \\ 2 \end{pmatrix}, \boldsymbol{\Sigma} = \begin{pmatrix} 10 & 2 \\ 2 & 1 \end{pmatrix}$, the normalized error is $\frac{|E - \bar{E}|}{E} = 0.0044\%$.

Next, we present in Figures 3 and 4 a numerical evaluation of (23) and (24). As can be seen, the Monte-Carlo simulations verify the analytical results.

## B    Proofs of Results in Section 4

### B.1    Proof of Proposition 1

*Proof.* Let

$$\mathcal{D} = \{(y_n = x_n + \epsilon_{n,m}, x_n)\}, n \in [N], m \in [M], \quad -\infty < x_1 < x_2 < \cdots < n_N < \infty$$

such that Assumption 1 holds. We define a reduced dataset, which only contains the noisy samples with the most extreme noises

$$\bar{\mathcal{D}} = \{(x_n + \epsilon_n^{\min}, x_n), (x_n + \epsilon_n^{\max}, x_n)\}, n \in [N].$$

We define the $\ell^2$ penalty on the weights,

$$C(\theta, K) = \sum_{k=1}^{K} \left( \|\boldsymbol{a}_k\|^2 + \|\boldsymbol{w}_k\|^2 \right).$$

According to Theorem 1.2. in [Hanin, 2021], since we have opposite discrete curvature on the intervals $[x_1 + \epsilon_1^{\max}, x_2 + \epsilon_1^{\min}] \cdots [x_{N-1} + \epsilon_1^{max}, x_N + \epsilon_1^{min}]$,

$$\{h_\theta(y) \mid h_\theta(y) = x \; \forall (y, x) \in \bar{\mathcal{D}}, C(\theta, K) = C_*\} = \{f_{1D}^*(y)\},$$

where

$$C_* = \inf_{\theta, K} \{C(\theta, K) \mid \forall (y, x) \in \bar{\mathcal{D}} \; h_\theta(y) = x\}.$$

Also note that,

$$\{h_\theta(y) \mid h_\theta(y) = x \; \forall (y, x) \in \mathcal{D}, C(\theta, K) = C_*\} = \{f_{1D}^*(y)\}$$

since $f_{1D}^*(y)$ interpolates all the points in $\mathcal{D}$, and if we assume by contradiction that $C_* > \inf_{\theta, K} \{C(\theta, K) \mid \forall (y, x) \in \mathcal{D} \; h_\theta(y) = x\}$ then $C_* \neq \inf_{\theta, K} \{C(\theta, K) \mid \forall (y, x) \in \bar{\mathcal{D}} \; h_\theta(y) = x\}$, thus contradicting Theorem 1.2. in [Hanin, 2021]. Note that the minimal number of neurons needed to represent $f_{1D}^*(y)$ is $2N - 2$. So, if $K \geq 2N - 2$ then $f_{1D}^*$ is the minimizer of the representation cost (i.e., $f_{1D}^* \in \arg\min_f R(f)$). □

## B.2 Proof of Theorem 1

First we prove the following lemmas.

**Lemma 3.** *Let $y = x + \sigma\epsilon$ where $x, \epsilon \in \mathbb{R}, \sigma > 0$ then,*

$$\lim_{\sigma \to 0^+} \hat{x}^{\text{eMMSE}}(y) = \hat{x}^{1-\text{NN}}(y)$$

*Proof.* Notice that the following holds,

$$
\frac{\exp\left(-\frac{|y-x_i|^2}{2\sigma^2}\right)}{\sum_{i=1}^N \exp\left(-\frac{|y-x_i|^2}{2\sigma^2}\right)} = \frac{\exp\left(-\frac{|y-x_i|^2}{2\sigma^2}\right)\exp\left(\frac{\min_n|y-x_n|^2}{2\sigma^2}\right)}{\sum_{i=1}^N \exp\left(\frac{|y-x_i|^2}{2\sigma^2}\right)\exp\left(\frac{\min_n|y-x_n|^2}{2\sigma^2}\right)}
$$

$$
= \frac{\exp\left(-\frac{|y-x_i|^2-\min_n|y-x_n|^2}{2\sigma^2}\right)}{\sum_{i=1}^N \exp\left(-\frac{|y-x_i|^2-\min_n|y-x_n|^2}{2\sigma^2}\right)} .
$$

In addition,

$$
\lim_{\sigma \to 0^+} \exp\left(-\frac{|y-x_i|^2 - \min_n|y-x_n|^2}{2\sigma^2}\right) = \begin{cases} 1 & |y-x_i|^2 = \min_n|y-x_n|^2 \\ 0 & |y-x_i|^2 \neq \min_n|y-x_n|^2 \end{cases}
$$

so we obtain,

$$
\lim_{\sigma \to 0^+} \frac{\exp\left(-\frac{|y-x_i|^2}{2\sigma^2}\right)}{\sum_{i=1}^N \exp\left(-\frac{|y-x_i|^2}{2\sigma^2}\right)} = \begin{cases} 1 & |y-x_i|^2 = \min_n|y-x_n|^2 \\ 0 & |y-x_i|^2 \neq \min_n|y-x_n|^2 \end{cases} .
$$

Therefore,

$$
\lim_{\sigma \to 0^+} \hat{x}^{\text{eMMSE}}(y) = \lim_{\sigma \to 0^+} \frac{\sum_{i=1}^N x_i \exp\left(-\frac{|y-x_i|^2}{2\sigma^2}\right)}{\sum_{i=1}^N \exp\left(-\frac{|y-x_i|^2}{2\sigma^2}\right)} = \arg\min_n |y-x_n|^2
$$

$$
= \arg\min_n |y-x_n| = \hat{x}^{1-\text{NN}}(y) .
$$

$\square$

**Lemma 4.** *Let $y = x + \sigma\epsilon$ where $x \in \mathbb{R}, \sigma > 0$ and $\epsilon \sim \mathcal{N}(0,1)$ then,*

$$\lim_{\sigma \to 0^+} \text{MSE}\left(\hat{x}^{\text{eMMSE}}(y)\right) = \lim_{\sigma \to 0^+} \text{MSE}\left(\hat{x}^{1-\text{NN}}(y)\right)$$

*Proof.*

$$
\text{MSE}\left(\hat{x}^{\text{eMMSE}}(y)\right) = \mathbb{E}\left[\left(\hat{x}^{\text{eMMSE}}(y) - x\right)^2\right]
$$

$$
= \mathbb{E}\left[\left(\hat{x}^{\text{eMMSE}}(y) - \hat{x}^{1-\text{NN}}(y) + \hat{x}^{1-\text{NN}}(y) - x\right)^2\right]
$$

$$
= \text{MSE}\left(\hat{x}^{1-\text{NN}}(y)\right) + \mathbb{E}\left[\left(\hat{x}^{\text{eMMSE}}(y) - \hat{x}^{1-\text{NN}}(y)\right)^2\right]
$$

$$
+ 2\mathbb{E}\left[\left(\hat{x}^{1-\text{NN}}(y) - x\right)\left(\hat{x}^{\text{eMMSE}}(y) - \hat{x}^{1-\text{NN}}(y)\right)\right]
$$

Note that,

$$
\hat{x}^{\text{eMMSE}}(y) = \frac{\sum_{i=1}^N x_i \exp\left(-\frac{|y-x_i|^2}{2\sigma^2}\right)}{\sum_{i=1}^N \exp\left(-\frac{|y-x_i|^2}{2\sigma^2}\right)} \leq \frac{\sum_{i=1}^N \max_j\{x_j\}\exp\left(-\frac{|y-x_i|^2}{2\sigma^2}\right)}{\sum_{i=1}^N \exp\left(-\frac{|y-x_i|^2}{2\sigma^2}\right)}
$$

$$
= \max_j\{x_j\} \frac{\sum_{i=1}^N \exp\left(-\frac{|y-x_i|^2}{2\sigma^2}\right)}{\sum_{i=1}^N \exp\left(-\frac{|y-x_i|^2}{2\sigma^2}\right)} = \max_j\{x_j\} ,
$$

$$\hat{x}^{\text{eMMSE}}(y) = \frac{\sum_{i=1}^{N} x_i \exp\left(-\frac{|y-x_i|^2}{2\sigma^2}\right)}{\sum_{i=1}^{N} \exp\left(-\frac{|y-x_i|^2}{2\sigma^2}\right)} \geq \frac{\sum_{i=1}^{N} \min_j\{x_j\} \exp\left(-\frac{|y-x_i|^2}{2\sigma^2}\right)}{\sum_{i=1}^{N} \exp\left(-\frac{|y-x_i|^2}{2\sigma^2}\right)}$$

$$= \min_j\{x_j\} \frac{\sum_{i=1}^{N} \exp\left(-\frac{|y-x_i|^2}{2\sigma^2}\right)}{\sum_{i=1}^{N} \exp\left(-\frac{|y-x_i|^2}{2\sigma^2}\right)} = \min_j\{x_j\}.$$

Similarly,

$$\hat{x}^{1-\text{NN}}(y) = \arg\min_i |y - x_i| \leq \max_i\{x_i\}$$

$$\hat{x}^{1-\text{NN}}(y) = \arg\min_i |y - x_i| \geq \min_i\{x_i\}$$

thus $\exists M_0 > 0 \ \forall \sigma > 0$

$$|\hat{x}^{\text{eMMSE}}(y) - \hat{x}^{1-\text{NN}}(y)| < M_0.$$

According to Lemma 3

$$\lim_{\sigma \to 0^+} \hat{x}^{\text{eMMSE}}(y) = \hat{x}^{1-\text{NN}}(y)$$

almost surely. Note that,

$$\mathbb{E}_{x,y}\left[\left(\hat{x}^{\text{eMMSE}}(y) - \hat{x}^{1-\text{NN}}(y)\right)^2\right] = \mathbb{E}_{x,\epsilon}\left[\left(\hat{x}^{\text{eMMSE}}(x + \sigma\epsilon) - \hat{x}^{1-\text{NN}}(x + \sigma\epsilon)\right)^2\right]$$

Therefore, by the Dominated convergence theorem we obtain

$$\lim_{\sigma \to 0^+} \mathbb{E}_{x,\epsilon}\left[\left(\hat{x}^{\text{eMMSE}}(x + \sigma\epsilon) - \hat{x}^{1-\text{NN}}(x + \sigma\epsilon)\right)^2\right] =$$

$$\mathbb{E}_{x,\epsilon}\left[\lim_{\sigma \to 0^+}\left(\hat{x}^{\text{eMMSE}}(x + \sigma\epsilon) - \hat{x}^{1-\text{NN}}(x + \sigma\epsilon)\right)^2\right] = 0$$

Similarly,

$$\lim_{\sigma \to 0^+} \mathbb{E}_{x,\epsilon}\left[\left(\hat{x}^{1-\text{NN}}(x + \sigma\epsilon) - x\right)\left(\hat{x}^{\text{eMMSE}}(x + \sigma\epsilon) - \hat{x}^{1-\text{NN}}(x + \sigma\epsilon)\right)\right] =$$

$$\mathbb{E}_{x,\epsilon}\left[\lim_{\sigma \to 0^+}\left(\hat{x}^{1-\text{NN}}(x + \sigma\epsilon) - x\right)\left(\hat{x}^{\text{eMMSE}}(x + \sigma\epsilon) - \hat{x}^{1-\text{NN}}(x + \sigma\epsilon)\right)\right] = 0$$

Since $\hat{x}^{\text{eMMSE}}(y)$ and $\hat{x}^{1-\text{NN}}(y)$ are bounded and $\mathbb{E}[x] < \infty$. Therefore, we get

$$\lim_{\sigma \to 0^+} \text{MSE}\left(\hat{x}^{\text{eMMSE}}(y)\right) = \lim_{\sigma \to 0^+} \text{MSE}\left(\hat{x}^{1-\text{NN}}(y)\right)$$

$\square$

Next, we prove Theorem 1.

*Proof.* Assume, without loss of generality, that $N = 2$. Let $p_x(x)$ be a probability density function with bounded second moment such that $p_x(x) > 0$ for all $x \in [x_1, x_2]$. According to Lemma 4

$$\lim_{\sigma \to 0^+} \text{MSE}\left(\hat{x}^{\text{eMMSE}}(y)\right) = \lim_{\sigma \to 0^+} \text{MSE}\left(\hat{x}^{1-\text{NN}}(y)\right)$$

So we need to prove that

$$\lim_{\sigma \to 0^+} \text{MSE}\left(\hat{x}^{1-\text{NN}}(y)\right) > \lim_{\sigma \to 0^+} \text{MSE}\left(f_{1D}^*(y)\right)$$

For the case of $N = 2$, the training set includes two data points $\{x_1, x_2\}$. So we get,

$$\hat{x}^{1-\text{NN}}(y) = \begin{cases} x_1 & y < \frac{x_1+x_2}{2} \\ x_2 & \frac{x_1+x_2}{2} \leq y \end{cases}$$

$$f_{1D}^*(y) = \begin{cases} x_1 & y < x_1 + \Delta_1 \\ \frac{x_2-x_1}{x_2-x_1+\Delta_2-\Delta_1}(y - x_1 - \Delta_1) + x_1 & x_1 + \Delta_1 \leq y \leq x_2 + \Delta_2 \\ x_2 & y > x_2 + \Delta_2 \end{cases}$$

where $\max_{m \in [M]} \epsilon_{1,m} = \Delta_1 > 0, \min_{m \in [M]} \epsilon_{2,m} = \Delta_2 < 0$. Note that $\hat{x}^{1-\text{NN}}(y) = f_{1D}^*(y)$ for all $y \in (-\infty, x_1 + \Delta_1) \cup (x_2 + \Delta, \infty)$ and $\mathbb{E}[x] < \infty, \mathbb{E}[x^2] < \infty$. Therefore,

$$\text{MSE}\left(\hat{x}^{1-\text{NN}}(y)\right) - \text{MSE}\left(f_{1D}^*(y)\right) =$$

$$\int_{-\infty}^{\infty} dx \int_{x_1 + \Delta_1}^{x_2 + \Delta_2} dy \left(\hat{x}^{1-\text{NN}}(y) - x\right)^2 p_{y|x}(y|x) p_x(x) \tag{25}$$

$$-\int_{-\infty}^{\infty} dx \int_{x_1 + \Delta_1}^{x_2 + \Delta_2} dy \left(f_{1D}^*(y) - x\right)^2 p_{y|x}(y|x) p_x(x) \tag{26}$$

First, we derive (25):

$$\int_{-\infty}^{\infty} dx \int_{x_1 + \Delta_1}^{x_2 + \Delta_2} dy \left(\hat{x}^{1-\text{NN}}(y) - x\right)^2 p_{y|x}(y|x) p_x(x) =$$

$$\int_{-\infty}^{\infty} \int_{x_1 + \Delta_1}^{\frac{x_1 + x_2}{2}} (x_1 - x)^2 p_{y|x}(y|x) p_x(x)\, dy dx + \int_{-\infty}^{\infty} \int_{\frac{x_1 + x_2}{2}}^{x_2 + \Delta_2} (x_2 - x)^2 p_{y|x}(y|x) p_x(x)\, dy dx =$$

$$\mathbb{E}_x \left[ P\left(y \in \left[x_1 + \Delta_1, \frac{x_1 + x_2}{2}\right] \middle| x\right)(x_1 - x)^2 \right] + \mathbb{E}_x \left[ P\left(y \in \left[\frac{x_1 + x_2}{2}, x_2 + \Delta_2\right] \middle| x\right)(x_2 - x)^2 \right].$$

Note that,

$$\mathbb{E}\left[ P\left(y \in \left[x_1 + \Delta_1, \frac{x_1 + x_2}{2}\right] \middle| x\right)(x_1 - x)^2 \right] < \mathbb{E}\left[(x_1 - x)^2\right] < \infty$$

$$\mathbb{E}\left[ P\left(y \in \left[\frac{x_1 + x_2}{2}, x_2 + \Delta_2\right] \middle| x\right)(x_2 - x)^2 \right] < \mathbb{E}\left[(x_2 - x)^2\right] < \infty$$

thus by the Dominated convergence theorem we obtain

$$\lim_{\sigma \to 0^+} \int_{-\infty}^{\infty} dx \int_{x_1 + \Delta_1}^{x_2 + \Delta_2} dy \left(\hat{x}^{1-\text{NN}}(y) - x\right)^2 p_{y|x}(y|x) p_x(x) =$$

$$\mathbb{E}_x \left[ \lim_{\sigma \to 0^+} P\left(y \in \left[x_1 + \Delta_1, \frac{x_1 + x_2}{2}\right] \middle| x\right)(x_1 - x)^2 \right] + \mathbb{E}_x \left[ \lim_{\sigma \to 0^+} P\left(y \in \left[\frac{x_1 + x_2}{2}, x_2 + \Delta_2\right] \middle| x\right)(x_2 - x)^2 \right] =$$

$$\mathbb{E}_x \left[ (x - x_1)^2 \, 1\left\{x \in \left[x_1, \frac{x_1 + x_2}{2}\right]\right\} \right] + \mathbb{E}_x \left[ (x - x_2)^2 \, 1\left\{x \in \left[\frac{x_1 + x_2}{2}, x_2\right]\right\} \right] > C > 0$$

since $p_x(x) > 0$ for all $x \in [x_1, x_2]$. Next, we derive (26)

$$\int_{-\infty}^{\infty} dx \int_{x_1 + \Delta_1}^{x_2 + \Delta_2} dy \left(f_{1D}^*(y) - x\right)^2 p_{y|x}(y|x) p_x(x) =$$

$$\mathbb{E}_{x,y} \left[ 1\{y \in [x_1 + \Delta_1, x_2 + \Delta_2]\} \left(f_{1D}^*(y) - x\right)^2 \right] =$$

$$\mathbb{E}_{x,\epsilon} \left[ 1\{x + \sigma\epsilon \in [x_1 + \Delta_1, x_2 + \Delta_2]\} \left(f_{1D}^*(x + \sigma\epsilon) - x\right)^2 \right]$$

Note that,

$$\mathbb{E}_{x,\epsilon} \left[ 1\{x + \sigma\epsilon \in [x_1 + \Delta_1, x_2 + \Delta_2]\} \left(f_{1D}^*(x + \sigma\epsilon) - x\right)^2 \right] < \mathbb{E}_{x,\epsilon} \left[ \left(f_{1D}^*(x + \sigma\epsilon) - x\right)^2 \right] < \infty$$

thus by the Dominated convergence theorem we obtain

$$\lim_{\epsilon \to 0^+} \int_{-\infty}^{\infty} dx \int_{x_1 + \Delta_1}^{x_2 + \Delta_2} dy \left(f_{1D}^*(y) - x\right)^2 p_{y|x}(y|x) p_x(x) =$$

$$\lim_{\epsilon \to 0^+} \mathbb{E}_{x,\epsilon} \left[ 1\{x + \sigma\epsilon \in [x_1 + \Delta_1, x_2 + \Delta_2]\} \left(f_{1D}^*(x + \sigma\epsilon) - x\right)^2 \right] =$$

$$\mathbb{E}_{x,\epsilon} \left[ \lim_{\epsilon \to 0^+} 1\{x + \sigma\epsilon \in [x_1 + \Delta_1, x_2 + \Delta_2]\} \left(f_{1D}^*(x + \sigma\epsilon) - x\right)^2 \right] = 0$$

since

$$\lim_{\epsilon \to 0^+} 1\left\{x + \sigma\epsilon \in [x_1 + \Delta_1, x_2 + \Delta_2]\right\} = 1\left\{x \in [x_1, x_2]\right\}$$

$$\lim_{\epsilon \to 0^+} f_{1D}^*(x + \sigma\epsilon) = \begin{cases} x_1 & x < x_1 \\ x & x_1 \le x \le x_2 \\ x_2 & x > x_2 \end{cases}$$

$\square$

## B.3 Proof of Lemma 1

*Proof.* First we prove that for all $y \in \cup_{n \in [N-1]}\left\{\frac{x_{n+1}\epsilon_n^{\max} - x_n\epsilon_{n+1}^{\min}}{\epsilon_n^{\max} - \epsilon_{n+1}^{\min}}\right\}$ $f_{1D}^*(y) = y$. We find the intersection between the line $f_{1D}^*(y) = y$ and the linear part of (17) (the third branch)

$$\frac{x_{n+1} - x_n}{x_{n+1} + \epsilon_{n+1}^{\min} - (x_n + \epsilon_n^{\max})}\left(y - (x_n + \epsilon_n^{\max})\right) + x_n = y \tag{27}$$

$$(x_{n+1} - x_n)\left(y - (x_n + \epsilon_n^{\max})\right) + x_n\left(x_{n+1} + \epsilon_{n+1}^{\min} - (x_n + \epsilon_n^{\max})\right) = y\left(x_{n+1} + \epsilon_{n+1}^{\min} - (x_n + \epsilon_n^{\max})\right) \tag{28}$$

$$x_n\left(x_{n+1} + \epsilon_{n+1}^{\min} - (x_n + \epsilon_n^{\max})\right) - (x_{n+1} - x_n)(x_n + \epsilon_n^{\max}) = y\left(\epsilon_{n+1}^{\min} - \epsilon_n^{\max}\right) \tag{29}$$

$$x_n\left(x_{n+1} + \epsilon_{n+1}^{\min}\right) - x_{n+1}(x_n + \epsilon_n^{\max}) = y\left(\epsilon_{n+1}^{\min} - \epsilon_n^{\max}\right) \tag{30}$$

$$x_n\epsilon_{n+1}^{\min} - x_{n+1}\epsilon_n^{\max} = y\left(\epsilon_{n+1}^{\min} - \epsilon_n^{\max}\right) \tag{31}$$

$$y = \frac{x_{n+1}\epsilon_n^{\max} - x_n\epsilon_{n+1}^{\min}}{\epsilon_n^{\max} - \epsilon_{n+1}^{\min}}. \tag{32}$$

Note that

$$x_n + \epsilon_n^{\max} < \frac{x_{n+1}\epsilon_n^{\max} - x_n\epsilon_{n+1}^{\min}}{\epsilon_n^{\max} - \epsilon_{n+1}^{\min}} < x_{n+1} + \epsilon_{n+1}^{\min} \tag{33}$$

Since,

$$x_n + \epsilon_n^{\max} < \frac{x_{n+1}\epsilon_n^{\max} - x_n\epsilon_{n+1}^{\min}}{\epsilon_n^{\max} - \epsilon_{n+1}^{\min}} \tag{34}$$

$$(x_n + \epsilon_n^{\max})\left(\epsilon_n^{\max} - \epsilon_{n+1}^{\min}\right) < x_{n+1}\epsilon_n^{\max} - x_n\epsilon_{n+1}^{\min} \tag{35}$$

$$\left(x_n + \epsilon_n^{\max} - \epsilon_{n+1}^{\min}\right)\epsilon_n^{\max} < x_{n+1}\epsilon_n^{\max} \tag{36}$$

$$x_n + \epsilon_n^{\max} - \epsilon_{n+1}^{\min} < x_{n+1} \tag{37}$$

$$x_n + \epsilon_n^{\max} < x_{n+1} + \epsilon_{n+1}^{\min} \tag{38}$$

which holds according to Assumption 1.

$$\frac{x_{n+1}\epsilon_n^{\max} - x_n\epsilon_{n+1}^{\min}}{\epsilon_n^{\max} - \epsilon_{n+1}^{\min}} < x_{n+1} + \epsilon_{n+1}^{\min} \tag{39}$$

$$x_{n+1}\epsilon_n^{\max} - x_n\epsilon_{n+1}^{\min} < \left(x_{n+1} + \epsilon_{n+1}^{\min}\right)\left(\epsilon_n^{\max} - \epsilon_{n+1}^{\min}\right) \tag{40}$$

$$-x_n\epsilon_{n+1}^{\min} < -\epsilon_{n+1}^{\min}\left(x_{n+1} - \epsilon_n^{\max} + \epsilon_{n+1}^{\min}\right) \tag{41}$$

$$x_n < x_{n+1} - \epsilon_n^{\max} + \epsilon_{n+1}^{\min} \tag{42}$$

$$x_n + \epsilon_n^{\max} < x_{n+1} + \epsilon_{n+1}^{\min} \tag{43}$$

which holds according to Assumption 1.

Next we prove that $f_{1D}^*(y)$ is contractive toward a set of the clean datapoints $\{\boldsymbol{x}_n\}_{n=1}^N$ on $\mathbb{R} \setminus \cup_{n \in [N-1]}\left\{\frac{x_{n+1}\epsilon_n^{\max} - x_n\epsilon_{n+1}^{\min}}{\epsilon_n^{\max} - \epsilon_{n+1}^{\min}}\right\}$. In the case where $y \in (-\infty, x_1 + \epsilon_1^{\min}]$ or $y \in \cup_{n \in [N]}[x_n + \epsilon_n^{\min}, x_n + \epsilon_n^{\max}]$ or $y \in [x_N + \epsilon_N^{\max}, \infty)$ $f_{1D}^*(y) \in \{x_n\}_{n=1}^N$ therefore (18) holds for all $0 \le \alpha < 1$. In the case where $y \in \cup_{n \in [N-1]}[x_n + \epsilon_n^{\max}, \frac{x_{n+1}\epsilon_n^{\max} - x_n\epsilon_{n+1}^{\min}}{\epsilon_n^{\max} - \epsilon_{n+1}^{\min}})$ we choose $i = n$,

$$|f_{1D}^*(y) - f(x_n)| = \frac{x_{n+1} - x_n}{x_{n+1} + \epsilon_{n+1}^{\min} - (x_n + \epsilon_n^{\max})}\left(y - (x_n + \epsilon_n^{\max})\right) \tag{44}$$

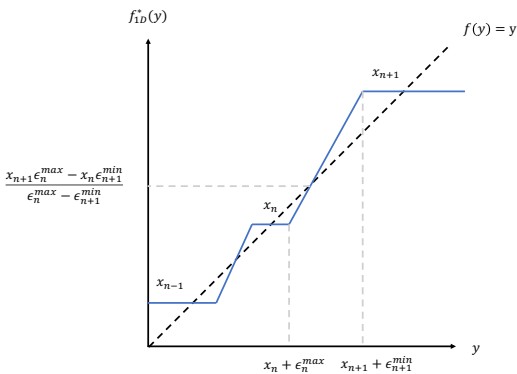

Figure 5: Illustration of $f_{1D}^*(y)$.

There exists $0 < \gamma_1 < 1$ such that

$$x_n \leq \frac{x_{n+1} - x_n}{x_{n+1} + \epsilon_{n+1}^{\min} - (x_n + \epsilon_n^{\max})} \left(y - (x_n + \epsilon_n^{\max})\right) + x_n \leq \gamma_1 y \tag{45}$$

since for $y \in \cup_{n \in [N-1]} [x_n + \epsilon_n^{\max}, \frac{x_{n+1}\epsilon_n^{\max} - x_n \epsilon_{n+1}^{\min}}{\epsilon_n^{\max} - \epsilon_{n+1}^{\min}})$ $f_{1D}^*(y)$ is bellow the line $f(y) = y$ because $f_{1D}^*(y)$ is an affine function with slope larger than 1 and $f_{1D}^*(\frac{x_{n+1}\epsilon_n^{\max} - x_n \epsilon_{n+1}^{\min}}{\epsilon_n^{\max} - \epsilon_{n+1}^{\min}}) = \frac{x_{n+1}\epsilon_n^{\max} - x_n \epsilon_{n+1}^{\min}}{\epsilon_n^{\max} - \epsilon_{n+1}^{\min}}$.
Therefore there exists $0 < \alpha_1 < 1$

$$|f_{1D}^*(y) - f(x_n)| = \frac{x_{n+1} - x_n}{x_{n+1} + \epsilon_{n+1}^{\min} - (x_n + \epsilon_n^{\max})} \left(y - (x_n + \epsilon_n^{\max})\right) \leq \gamma_1 y - x_n \leq \alpha_1 (y - x_n) \tag{46}$$

since,

$$\gamma_1 y - x_n \leq \alpha_1 (y - x_n) \tag{47}$$

$$\alpha_1 \geq \frac{\gamma_1 y - x_n}{y - x_n} . \tag{48}$$

In the case where $y \in \cup_{n \in [N-1]} (\frac{x_{n+1}\epsilon_n^{\max} - x_n \epsilon_{n+1}^{\min}}{\epsilon_n^{\max} - \epsilon_{n+1}^{\min}}, x_{n+1} + \epsilon_{n+1}^{\min}]$ we choose $i = n + 1$,

$$|f_{1D}^*(y) - f(x_{n+1})| = x_{n+1} - \frac{x_{n+1} - x_n}{x_{n+1} + \epsilon_{n+1}^{\min} - (x_n + \epsilon_n^{\max})} \left(y - (x_n + \epsilon_n^{\max})\right) - x_n . \tag{49}$$

There exists $0 < \gamma_2 < 1$ such that

$$x_{n+1} \geq \frac{x_{n+1} - x_n}{x_{n+1} + \epsilon_{n+1}^{\min} - (x_n + \epsilon_n^{\max})} \left(y - (x_n + \epsilon_n^{\max})\right) + x_n \geq \frac{1}{\gamma_2} y \tag{50}$$

since for $y \in \cup_{n \in [N-1]} (\frac{x_{n+1}\epsilon_n^{\max} - x_n \epsilon_{n+1}^{\min}}{\epsilon_n^{\max} - \epsilon_{n+1}^{\min}}, x_{n+1} + \epsilon_{n+1}^{\min}]$ $f_{1D}^*(y)$ is above the line $f(y) = y$ because $f_{1D}^*(y)$ is an affine function with slope larger than 1 and $f_{1D}^*(\frac{x_{n+1}\epsilon_n^{\max} - x_n \epsilon_{n+1}^{\min}}{\epsilon_n^{\max} - \epsilon_{n+1}^{\min}}) = \frac{x_{n+1}\epsilon_n^{\max} - x_n \epsilon_{n+1}^{\min}}{\epsilon_n^{\max} - \epsilon_{n+1}^{\min}}$.
Therefore there exists $0 < \alpha_2 < 1$

$$|f_{1D}^*(y) - f(x_{n+1})| = x_{n+1} - \frac{x_{n+1} - x_n}{x_{n+1} + \epsilon_{n+1}^{\min} - (x_n + \epsilon_n^{\max})} \left(y - (x_n + \epsilon_n^{\max})\right) - x_n \tag{51}$$

$$\leq x_{n+1} - \frac{1}{\gamma_2} y \leq \alpha_2 (x_{n+1} - y) \tag{52}$$

since,

$$x_{n+1} - \frac{1}{\gamma_2} y \leq \alpha_2 (x_{n+1} - y) \tag{53}$$

$$\alpha_2 \geq \frac{x_{n+1} - \frac{1}{\gamma_2} y}{x_{n+1} - y} . \tag{54}$$

Therefore (18) holds for $\alpha = \max\{\alpha_1, \alpha_2\}$. $\qquad\square$

# C   Proofs of Results in Section 5

Let $\boldsymbol{f} : \mathbb{R}^d \to \mathbb{R}^d$ be any function realizable as a shallow ReLU network. Consider any parametrization of $\boldsymbol{f}$ given by $\boldsymbol{f}(\boldsymbol{y}) = \sum_{k=1}^K \boldsymbol{a}_k [\boldsymbol{w}_k^\top \boldsymbol{y} + b_k]_+ + \boldsymbol{V}\boldsymbol{y} + \boldsymbol{c}$. We say such a parametrization is a *minimal representative of* $\boldsymbol{f}$ if $\|\boldsymbol{w}_k\|_2 = 1$ and $\boldsymbol{a}_k \neq 0$ for all $k = 1, ..., K$, and $R(\boldsymbol{f}) = \sum_{k=1}^K \|\boldsymbol{a}_k\|_2$. In particular, the units making up a minimal representative must be distinct in the sense that the hyperplanes describing ReLU boundaries $H_k = \{\boldsymbol{x} \in \mathbb{R}^d : \boldsymbol{w}_k^\top \boldsymbol{x} + b_k = 0\}$ are distinct, which implies that no units can be cancelled or combined.

We will also make use of the following lemma, which shows that representation costs are invariant to a translation of the training samples, assuming the function is suitably translated.

**Lemma 5.** *Let* $\boldsymbol{f} : \mathbb{R}^d \to \mathbb{R}^d$ *be any function realizable as a shallow ReLU net satisfying norm-ball interpolation constraints* $\boldsymbol{f}(B(\boldsymbol{x}_n; \rho)) = \{\boldsymbol{x}_n\}$ *for all* $n = 1, ..., N$. *Let* $\boldsymbol{x}_0 \in \mathbb{R}^d$. *Then the function* $\boldsymbol{g}(\boldsymbol{y}) = \boldsymbol{f}(\boldsymbol{y} - \boldsymbol{x}_0) + \boldsymbol{x}_0$ *is such that* $R(\boldsymbol{g}) = R(\boldsymbol{f})$ *and* $\boldsymbol{g}(B(\boldsymbol{x}_n + \boldsymbol{x}_0; \rho)) = \{\boldsymbol{x}_n + \boldsymbol{x}_0\}$ *for all* $n = 1, ..., N$.

*Proof.* First we show $\boldsymbol{g}(B(\boldsymbol{x}_n + \boldsymbol{x}_0; \rho)) = \{\boldsymbol{x}_n + \boldsymbol{x}_0\}$ for all $n = 1, ..., N$. Fix any $n$, and let $\boldsymbol{y} \in B(\boldsymbol{x}_n + \boldsymbol{x}_0; \rho)$. Then $\boldsymbol{y} = \boldsymbol{y}' + \boldsymbol{x}_0$ for some $\boldsymbol{y}' \in B(\boldsymbol{x}_n; \rho)$, and so $\boldsymbol{g}(\boldsymbol{y}) = \boldsymbol{f}(\boldsymbol{y}') + \boldsymbol{x}_0 = \boldsymbol{x}_n + \boldsymbol{x}_0$, as claimed.

To show that $R(\boldsymbol{g}) = R(\boldsymbol{f})$, let $\boldsymbol{f}(\boldsymbol{y}) = \sum_{k=1}^K \boldsymbol{a}_k [\boldsymbol{w}_k^\top \boldsymbol{y} + b_k]_+ + \boldsymbol{V}\boldsymbol{y} + \boldsymbol{c}$ be any minimal representative of $\boldsymbol{f}$. Then

$$\boldsymbol{g}(\boldsymbol{y}) = \sum_{k=1}^K \boldsymbol{a}_k [\boldsymbol{w}_k^\top (\boldsymbol{y} - \boldsymbol{x}_0) + b_k]_+ + \boldsymbol{V}(\boldsymbol{y} - \boldsymbol{x}_0) + \boldsymbol{c} + \boldsymbol{x}_0 \tag{55}$$

$$= \sum_{k=1}^K \boldsymbol{a}_k [\boldsymbol{w}_k^\top \boldsymbol{y} + \tilde{b}_k]_+ + \boldsymbol{V}\boldsymbol{y} + \tilde{\boldsymbol{c}} \tag{56}$$

with $\tilde{b}_k = b_k - \boldsymbol{w}_k^\top \boldsymbol{x}_0$ and $\tilde{\boldsymbol{c}} = \boldsymbol{c} + \boldsymbol{x}_0 - \boldsymbol{V}\boldsymbol{x}_0$. And so $R(\boldsymbol{g}) \leq \sum_{k=1}^K \|\boldsymbol{a}_k\|_2 = R(\boldsymbol{f})$. A parallel argument with the roles of $\boldsymbol{g}$ and $\boldsymbol{f}$ reversed shows that $R(\boldsymbol{f}) \leq R(\boldsymbol{g})$, and so $R(\boldsymbol{f}) = R(\boldsymbol{g})$.  □

In particular, this lemma shows that if $\boldsymbol{f}^*(\boldsymbol{y})$ is a norm-ball interpolating representation cost minimizer of the training samples $\{\boldsymbol{x}_n\}_{n=1}^N$, then $\boldsymbol{g}^*(\boldsymbol{y}) = \boldsymbol{f}^*(\boldsymbol{y} - \boldsymbol{x}_0) + \boldsymbol{x}_0$ is a norm-ball interpolating min-cost solution of the translated training samples $\{\boldsymbol{x}_n + \boldsymbol{x}_0\}_{n=1}^N$.

## C.1   Training data belonging to a subspace

The proof of Theorem 2 is a direct consequence of the following two lemmas:

**Lemma 6.** *Suppose the clean training samples* $\{\boldsymbol{x}_n\}_{n=1}^N$ *belong to a* $r$-*dimensional subspace* $\mathcal{S} \subset \mathbb{R}^d$, *and let* $\boldsymbol{P}_\mathcal{S}$ *denote the orthogonal projector onto* $\mathcal{S}$. *Let* $\boldsymbol{f}(\boldsymbol{y})$ *be any function realizable as a shallow ReLU network satisfying* $\boldsymbol{f}(B(\boldsymbol{x}_n, \rho)) = \{\boldsymbol{x}_n\}$ *for all* $n = 1, ..., N$. *Define* $\boldsymbol{g}(\boldsymbol{y}) = \boldsymbol{P}_\mathcal{S} \boldsymbol{f}(\boldsymbol{x})$. *Then* $\boldsymbol{g}(B(\boldsymbol{x}_n, \rho)) = \{\boldsymbol{x}_n\}$ *for all* $n = 1, ..., N$, *and* $R(\boldsymbol{g}) \leq R(\boldsymbol{f})$, *with strict inequality if* $\boldsymbol{f} \neq \boldsymbol{g}$.

*Proof.* First, for any $\boldsymbol{y} \in B(\boldsymbol{x}_n, \rho)$ we have $\boldsymbol{g}(\boldsymbol{y}) = \boldsymbol{P}_\mathcal{S} \boldsymbol{f}(\boldsymbol{y}) = \boldsymbol{P}_\mathcal{S} \boldsymbol{x}_n = \boldsymbol{x}_n$. Therefore, $\boldsymbol{g}(B(\boldsymbol{x}_n, \rho)) = \{\boldsymbol{x}_n\}$ for all $n = 1, ..., N$.

Now, let $\boldsymbol{f}(\boldsymbol{y}) = \sum_{k=1}^K \boldsymbol{a}_k [\boldsymbol{w}_k^\top \boldsymbol{y} + b_k]_+ + \boldsymbol{V}\boldsymbol{y} + \boldsymbol{c}$ be a minimal representative of $\boldsymbol{f}$. Then $\boldsymbol{g}(\boldsymbol{y}) = \sum_{k=1}^K \boldsymbol{P}\boldsymbol{a}_k [\boldsymbol{w}_k^\top \boldsymbol{y} + b_k]_+ + \boldsymbol{P}(\boldsymbol{V}\boldsymbol{y} + \boldsymbol{c})$. Since $\|\boldsymbol{P}\boldsymbol{a}_k\| \leq \|\boldsymbol{a}_k\|$ for all $k$, we have $R(\boldsymbol{g}) \leq \sum_{k=1}^K \|\boldsymbol{P}_\mathcal{S} \boldsymbol{a}_k\| \leq \sum_{k=1}^K \|\boldsymbol{a}_k\| = R(\boldsymbol{f})$.

Now we show $\boldsymbol{f} \neq \boldsymbol{g}$ implies $R(\boldsymbol{g}) < R(\boldsymbol{f})$. Observe that if any of the outer-layer weight vectors $\boldsymbol{a}_k \notin \mathcal{S}$ then $\|\boldsymbol{P}_\mathcal{S} \boldsymbol{a}_k\| < \|\boldsymbol{a}_k\|$, which implies $R(\boldsymbol{g}) < R(\boldsymbol{f})$. Hence, it suffices to show that $\boldsymbol{f} \neq \boldsymbol{g}$ implies some $\boldsymbol{a}_k \notin \mathcal{S}$.

First, consider the case where $\boldsymbol{P}_\mathcal{S} \boldsymbol{V} = \boldsymbol{V}$ and $\boldsymbol{P}_\mathcal{S} \boldsymbol{c} = \boldsymbol{c}$. Then in this case $\boldsymbol{f} \neq \boldsymbol{g}$ if only if $\sum_{k=1}^K (\boldsymbol{P}_\mathcal{S} \boldsymbol{a}_k - \boldsymbol{a}_k)[\boldsymbol{w}_k^\top \boldsymbol{y} + b_k]_+ \neq 0$ for all $\boldsymbol{y} \in \mathbb{R}^d$, which implies $\boldsymbol{P}_\mathcal{S} \boldsymbol{a}_k \neq \boldsymbol{a}_k$ for some $k$, or equivalently $\boldsymbol{a}_k \notin \mathcal{S}$.

Next, assume either $P_S V \neq V$ or $P_S c \neq c$. Fix any training sample index $n$, and let $A_n$ denote the index set of units active over the ball $B(x_n, \rho)$. Then, since $f(y)$ is constant for all $y \in B(x_n, \rho)$, the Jacobian $\partial f(y) = \sum_{k \in A_n} a_k w_k^\top + V = 0$ for all $y \in B(x_n, \rho)$. This gives $V = -\sum_{k \in A_n} a_k w_k^\top$. Therefore, if $P_S V \neq V$ then at least one of the $a_k \notin S$. On the other hand, if $P_S c \neq c$ then for all $y \in B(x_n, \rho)$ we have

$$f(y) = \sum_{k \in A_n} a_k(w_k^\top y + b_k) + Vy + c = \sum_{k \in A_n}(a_k w_k^\top + V)y + \sum_{k \in A_n} b_k a_k + c = \sum_{k \in A_n} b_k a_k + c = x_n,$$

which implies $c = x_n - \sum_{k \in A_n} b_k a_k$, and since $x_n \in S$ this implies some $a_k \notin S$, proving the claim. $\square$

**Lemma 7.** *Suppose the clean training samples $\{x_n\}_{n=1}^N$ belong to an $r$-dimensional subspace $S \subset \mathbb{R}^d$, and let $P_S$ denote the orthogonal projector onto $S$. Let $f$ be any network satisfying $f(B(x_n, \rho)) = \{x_n\}$. Define $h(y) = f(P_S y)$. Then $h(B(x_n, \rho)) = \{x_n\}$ and $R(h) \leq R(f)$, with strict inequality if $h \neq f$.*

*Proof.* Define $P_S^{-1}(B(x_n, \rho)) := \{y \in \mathbb{R}^d : P_S y \in B(x_n, \rho)\}$, i.e., the set of points in $\mathbb{R}^d$ whose projection onto $S$ is contained in $B(x_n, \rho)$. Then clearly $h(P_S^{-1}(B(x_n, \rho))) = \{x_n\}$. Also, by properties of norm-balls, we have $B(x_n, \rho) \subset P_S^{-1}(B(x_n, \rho))$. Therefore, $h(B(x_n, \rho)) = \{x_n\}$ for all $n = 1, ..., N$.

Next, let $f(y) = \sum_{k=1}^K a_k[w_k^\top y + b_k]_+ + Vy + c$ be a minimal representative of $f$. Then $h(y) = \sum_{k=1}^K a_k[(P_S w_k)^\top y + b_k]_+ + V P_S y + c$. Since $\|P_S w_k\| \leq \|w_k\|$ for all $k$, we have $R(h) \leq \sum_{k=1}^K \|a_k\| \|P_S w_k\| \leq \sum_{k=1}^K \|a_k\| \|w_k\| = R(f)$.

Finally, we show that if $f \neq h$, then $R(h) < R(f)$. Observe that if any of the inner-layer weight vectors $w_k \notin S$ then $\|P w_k\| < \|w_k\|$, which implies $R(h) < R(f)$. Hence, it suffices to show that $f \neq h$ implies some $w_k \notin S$. First, consider the case $V P_S = V$. Then $f \neq h$ if and only if $P_S w_k \neq w_k$ for some $k$, or equivalently, $w_k \notin S$. Next, consider the case $V P_S \neq V$. Fix any training sample index $n$, and let $A_n$ denote the index set of units active over $B(x_n, \rho)$. Then, since $f(y)$ is constant for all $y \in B(x_n, \rho)$, the Jacobian $\partial f(x) = \sum_{k \in A_n} a_k w_k^\top + V = 0$ for all $y \in B_n(x_n, \rho)$. This gives $V = -\sum_k a_k w_k^\top$. Therefore, if $V P_S \neq V$ (i.e., at least one row of $V$ is not contained in $S$) then at least one of the $w_k \notin S$, proving the claim. $\square$

Finally, we now give the proof Theorem 2 and Corollary 1.

*Proof of Theorem 2.* Let $f^*$ be any min-cost solution. Applying both Lemma 6 and 7 we see it must be the case that $f^*(y) = P_S f^*(P_S y)$ for all $y \in \mathbb{R}^d$, since otherwise the representation cost of $f^*$ could be reduced. $\square$

*Proof of Corollary 1.* Suppose $S$ is one-dimensional, i.e., $S = \text{span}\{u\}$ for some unit vector $u \in \mathbb{R}^d$. Theorem 2 shows we can express any min-cost solution $f^*$ as

$$f^*(y) = \sum_{k=1}^K a_k u[s_k u^\top y + b_k]_+ + v u u^\top y + c u = u\phi(u^\top y)$$

where

$$\phi(t) = \sum_k a_k[s_k t + b_k]_+ + vt + c$$

is such that $R(f^*) = R(\phi)$. Therefore, minimizing the representation cost subject to norm-ball interpolation constraints is reduces to a univariate problem:

$$\min_\phi R(\phi) \ \ s.t. \ \ \phi((c_n - \rho, c_n + \rho)) = c_n$$

where $c_n = u^\top x_n$. The minimizing $\phi^*$ is unique and coincides with the 1-D denoiser $f_{1D}^*$ in (17) with $x_n = c_n$ and $\epsilon_n^{\max} = -\epsilon_n^{\min} = \rho$. $\square$

## C.2 Data along rays

We begin with a key lemma that is central to the proof of Theorem 3.

**Lemma 8.** *Suppose $\{u_i\}_{i=1}^n \subset \mathbb{R}^d$ is a collection of $n > 1$ unit vectors such that $u_i^\top u_j < 0$ for all $i \neq j$, and $w$ is a unit vector such that $u_i^\top w > 0$ for all $i = 1, ..., n$. Let $a \in \mathbb{R}^d$ be any vector. Then,*

$$\sum_{i=1}^n |u_i^\top a|(u_i^\top w) < \|a\|.$$

Before giving the proof of Lemma 8, we first prove an auxiliary result.

**Lemma 9.** *Let $a \in \mathbb{R}^d$ be a unit vector, and suppose $\{u_i\}_{i=1}^n \subset \mathbb{R}^d$ is a collection of unit vectors such that $u_i^\top a > 0$ for all $i$ and $u_i^\top u_j < 0$ for all $i \neq j$. Let $b = \sum_{i=1}^n u_i u_i^\top a$. Then $\|b - a/2\| \leq 1/2$ with strict inequality if $n > 1$.*

*Proof.* It suffices to show $b^\top a \geq b^\top b$, since if this is the case then

$$\|b - a/2\| = \sqrt{(b - a/2)^\top (b - a/2)} = \sqrt{b^\top b - b^\top a + \frac{1}{4}a^\top a} \leq \frac{1}{2}, \tag{57}$$

which also holds with strict inequality when $b^\top a > b^\top b$.

First, if $n = 1$, then $b = a^\top u_1 u_1^\top$, and so $b^\top b = a^\top u_1 u_1^\top u_1 u_1^\top a = a^\top u_1 u_1^\top a = b^\top a$. Now assume $n > 1$. Then we have

$$b^\top a = \sum_{i=1}^n a^\top u_i u_i^\top a \tag{58}$$

$$= \sum_{i=1}^n a^\top u_i u_i^\top u_i u_i^\top a \tag{59}$$

$$> \sum_{i=1}^n \sum_{j=1}^n a^\top u_i u_i^\top u_j u_i^\top a \tag{60}$$

$$= b^\top b \tag{61}$$

The first and last equalities hold by definition. The second equality holds because each $u_i$ is a unit vector. The last inequality holds because for $i \neq j$

$$(a^\top u_i)(u_i^\top u_j)(u_j^\top a) < 0,$$

since $u_i^\top u_j < 0$ and $a^\top u_i, a^\top u_j > 0$ by assumption. $\qquad\square$

Now we give the proof of Lemma 8.

*Proof of Lemma 8.* Let $v = \sum_{i=1}^n |u_i^\top a| u_i$. Then we have

$$S = \sum_{i=1}^n |u_i^\top a|(u_i^\top w) = v^\top w$$

WLOG, we may assume $\|a\| = 1$, in which case the lemma reduces to showing $S = v^\top w < 1$. By the Cauchy-Schwartz inequality, it suffices to show $\|v\| < 1$. Toward this end, let us write $v = v_+ + v_-$ where $v_+ = \sum_{i:u_i^\top a>0} |u_i^\top a| u_i = \sum_{i:u_i^\top a>0} u_i u_i^\top a$ and $v_- = \sum_{i:u_i^\top a<0} |u_i^\top a| u_i = \sum_{i:u_i^\top a<0} u_i u_i^\top (-a)$. By Lemma 9 we have $\|v_+ - a/2\| \leq 1/2$ and $\|v_- + a/2\| \leq 1/2$, and so

$$\|v\| = \|v_+ + v_-\| \tag{62}$$

$$= \|v_+ - a/2 + v_- + a/2\| \tag{63}$$

$$\leq \|v_+ - a/2\| + \|v_- + a/2\| \tag{64}$$

$$\leq 1. \tag{65}$$

Also, if either of the index sets $I_+ := \{i : \boldsymbol{u}_i^\top \boldsymbol{a} > 0\}$ or $I_- := \{i : \boldsymbol{u}_i^\top \boldsymbol{a} < 0\}$ has cardinality greater than one then the lemma above guarantees $\|\boldsymbol{v}_+ - \boldsymbol{a}/2\| < 1/2$ or $\|\boldsymbol{v}_+ - \boldsymbol{a}/2\| < 1/2$, and so $\|\boldsymbol{v}\| < 1$, which gives $S < 1$.

It remains to show $S < 1$ when $I_+$ or $I_-$ has cardinality less than or equal to one, i.e., $|I_+| \leq 1$ or $|I_-| \leq 1$. We consider the various possibilities:

*Case 1:* $|I_+| = 0$ & $|I_-| = 0$. Then $\boldsymbol{v} = \boldsymbol{0}$ and so $S = \boldsymbol{v}^\top \boldsymbol{w} = 0 < 1$.

*Case 2:* $|I_+| = 1$ & $|I_-| = 0$ *or* $|I_+| = 0$ & $|I_-| = 1$. Then $\boldsymbol{v} = |\boldsymbol{u}_i^\top \boldsymbol{a}|\boldsymbol{u}_i$ for some $i$, and $|\boldsymbol{u}_j^\top \boldsymbol{a}| = 0$ for all $j \neq i$. By way of contradiction, assume that $S = \boldsymbol{v}^\top \boldsymbol{w} = |\boldsymbol{u}_i^\top \boldsymbol{a}|\boldsymbol{u}_i^\top \boldsymbol{w} = 1$. Since $\boldsymbol{u}_i, \boldsymbol{a}$, and $\boldsymbol{w}$ are all unit vectors, the only way this is possible is if $|\boldsymbol{u}_i^\top \boldsymbol{a}| = 1$ and $\boldsymbol{u}_i^\top \boldsymbol{w} = 1$, which implies $\boldsymbol{a} = \pm \boldsymbol{w}$ and $\boldsymbol{u}_i = \boldsymbol{w}$, and so $\boldsymbol{a} = \pm \boldsymbol{u}_i$. However, this shows that $\boldsymbol{u}_j^\top \boldsymbol{u}_i = 0$ for all $j \neq i$, contradicting our assumption that $\boldsymbol{u}_j^\top \boldsymbol{u}_i < 0$ for all $i \neq j$. Therefore, $S < 1$ in this case as well.

*Case 3:* $|I_+| = 1$ & $|I_-| = 1$. Then $\boldsymbol{v} = |\boldsymbol{u}_i^\top \boldsymbol{a}|\boldsymbol{u}_i + |\boldsymbol{u}_j^\top \boldsymbol{a}|\boldsymbol{u}_j$ for some $i \in I_+$ and $j \in I_-$, and so

$$\|\boldsymbol{v}\| = \||\boldsymbol{u}_i^\top \boldsymbol{a}|\boldsymbol{u}_i + |\boldsymbol{u}_j^\top \boldsymbol{a}|\boldsymbol{u}_j\| \tag{66}$$

$$= \|(\boldsymbol{u}_i \boldsymbol{u}_i^\top - \boldsymbol{u}_j \boldsymbol{u}_j^\top)\boldsymbol{a}\| \tag{67}$$

$$\leq \|\boldsymbol{u}_i \boldsymbol{u}_i^\top - \boldsymbol{u}_j \boldsymbol{u}_j^\top\| \tag{68}$$

where the last inequality follows by the fact that $\|\boldsymbol{a}\| = 1$. Finally, since the matrix $\boldsymbol{u}_i \boldsymbol{u}_i^\top - \boldsymbol{u}_j \boldsymbol{u}_j^\top$ is symmetric, its operator norm $\|\boldsymbol{u}_i \boldsymbol{u}_i^\top - \boldsymbol{u}_j \boldsymbol{u}_j^\top\|$ coincides with the maximum eigenvalue (in absolute value). Eigenvectors in the span of $\boldsymbol{u}_i$ and $\boldsymbol{u}_j$ have the form $\boldsymbol{v} = c_i \boldsymbol{u}_i + c_j \boldsymbol{u}_j$ where $c_i, c_j$ satisfy the equation

$$(\boldsymbol{u}_i \boldsymbol{u}_i^\top - \boldsymbol{u}_j \boldsymbol{u}_j^\top)(c_i \boldsymbol{u}_i + c_j \boldsymbol{u}_j) = \lambda(c_i \boldsymbol{u}_i + c_j \boldsymbol{u}_j) \tag{69}$$

where $\lambda \in \mathbb{R}$ is the corresponding eigenvalue. Expanding and equating coefficients gives the system

$$\begin{bmatrix} 1 - \lambda & \boldsymbol{u}_i^\top \boldsymbol{u}_j \\ \boldsymbol{u}_i^\top \boldsymbol{u}_j & 1 + \lambda \end{bmatrix} \begin{bmatrix} c_i \\ c_j \end{bmatrix} = \begin{bmatrix} 0 \\ 0 \end{bmatrix} \tag{70}$$

which has non-trivial solutions iff

$$(1 - \lambda)(1 + \lambda) - (\boldsymbol{u}_i^\top \boldsymbol{u}_j)^2 = 0 \quad \Longleftrightarrow \quad \lambda = \pm\sqrt{1 - (\boldsymbol{u}_i^\top \boldsymbol{u}_j)^2} \tag{71}$$

and so $|\lambda| < 1$. Therefore, $\|\boldsymbol{v}\| \leq \|\boldsymbol{u}_i \boldsymbol{u}_i^\top - \boldsymbol{u}_j \boldsymbol{u}_j^\top\| < 1$ as claimed. And so $S \leq \|\boldsymbol{v}\| < 1$. $\qquad\square$

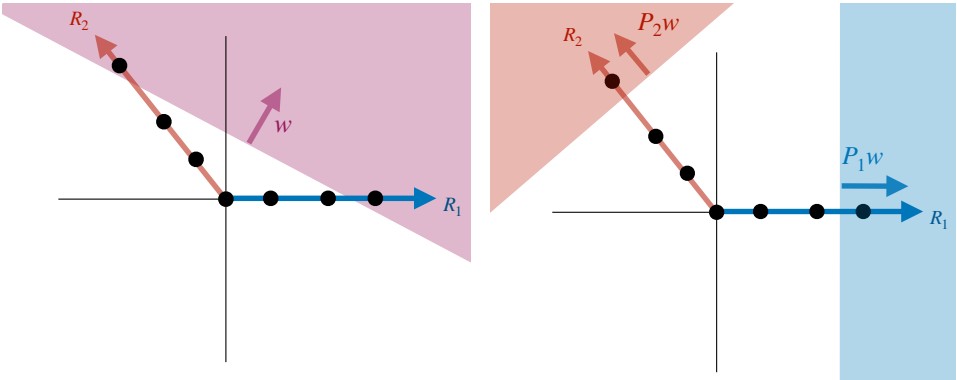

Figure 6: Illustration of "unit splitting" technique used in proof of Theorem 3. If a ReLU unit is active along two rays (left panel), it can be split into two units with lower representation cost by projecting its inner-layer weight vector $\boldsymbol{w}$ onto the two rays separately (right panel).

*Proof of Theorem 3.* Let $\boldsymbol{f}^*$ be any representation cost minimizer. We prove the claim by showing that if $\boldsymbol{f}^*$ fails to have the form specified in (20) then it is possible to construct a norm-ball interpolant $\boldsymbol{h}$ whose units are aligned with the rays that has strictly smaller representation cost than $\boldsymbol{f}^*$.

First, let $\boldsymbol{f}^*(\boldsymbol{y}) = \sum_{k=1}^{K} \boldsymbol{a}_k [\boldsymbol{w}_k^\top \boldsymbol{y} + b_k] + \boldsymbol{V}\boldsymbol{x} + \boldsymbol{c}$ be any minimal representative of $\boldsymbol{f}^*$. By properties of minimal representatives, none of the ReLU boundary sets $\{\boldsymbol{y} \in \mathbb{R}^d : \boldsymbol{w}_k^\top \boldsymbol{y} + b_k = 0\}$ can intersect any of the norm-balls centered at the training samples, since otherwise $\boldsymbol{f}^*$ would be non-constant on one of the norm-balls. Also, we may alter this parameterization in such a way that the active set of every unit avoids the ball centered at the origin $B_0 := B(\boldsymbol{0}, \rho)$ without changing the representation cost. In particular, suppose the active set of the $k$th unit contains $B_0$. Then, using the identity $[t]_+ = t - [-t]_+$ for all $t \in \mathbb{R}$, we have

$$\boldsymbol{a}_k [\boldsymbol{w}_k^\top \boldsymbol{y} + b_k]_+ = \boldsymbol{a}_k (\boldsymbol{w}_k^\top \boldsymbol{y} + b_k) - \boldsymbol{a}_k [-\boldsymbol{w}_k^\top \boldsymbol{y} - b_k]_+$$

for all $\boldsymbol{y} \in \mathbb{R}^d$. The active set of the reversed unit $-\boldsymbol{a}_k [-\boldsymbol{w}_k^\top \boldsymbol{y} - b_k]_+$ does not intersect $B_0$. Therefore, after applying this transformation to all units whose active sets contain $B_0$, we can write $\boldsymbol{f}^* = \boldsymbol{g}^* + \boldsymbol{\ell}$ where $\boldsymbol{g}^*$ is a sum of ReLU units whose active sets do not contain $B_0$ and $\boldsymbol{\ell}$ is an affine function that combines the original affine part $\boldsymbol{V}\boldsymbol{y} + \boldsymbol{c}$ with a sum of linear units of the form $\boldsymbol{a}_k (\boldsymbol{w}_k^\top \boldsymbol{y} + b_k \boldsymbol{a}_k)$ arising from the transformation above. However, because of the interpolation constraint $\boldsymbol{f}(B_0) = \{\boldsymbol{0}\}$, and by the fact that no units in $\boldsymbol{g}^*$ are active over $B_0$, for all $\boldsymbol{y} \in B_0$ we have

$$\boldsymbol{f}^*(\boldsymbol{y}) = \boldsymbol{g}^*(\boldsymbol{y}) + \boldsymbol{\ell}(\boldsymbol{y}) = \boldsymbol{\ell}(\boldsymbol{y}) = \boldsymbol{0} \tag{72}$$

which implies $\boldsymbol{\ell} \equiv \boldsymbol{0}$, i.e., $\boldsymbol{f}^*(\boldsymbol{y}) = \boldsymbol{g}^*(\boldsymbol{y})$ for all $\boldsymbol{y}$. Finally, note that the inner- and outer-layer weight vectors on the reversed units change only in sign. Therefore, this re-parameterization does not change the representation cost, and so is also a minimal representative.

Now we construct a new norm-ball interpolant $\boldsymbol{h}$ as follows. Let $\boldsymbol{f}_\ell$ be the function defined as the sum of all units making up the re-parametrized $\boldsymbol{f}^*$ that are active on at least one norm-ball centered on the $\ell$th ray. Also, let $\boldsymbol{P}_\ell = \boldsymbol{u}_\ell \boldsymbol{u}_\ell^\top \in \mathbb{R}^{d \times d}$ denote the orthogonal projector onto the $\ell$th ray. Define $\boldsymbol{h} = \sum_{\ell=1}^{L} \boldsymbol{h}_\ell$ where $\boldsymbol{h}_\ell(\boldsymbol{y}) := \boldsymbol{P}_\ell \boldsymbol{f}_\ell(\boldsymbol{P}_\ell \boldsymbol{y})$. Put in words, $\boldsymbol{h}$ is constructed by "splitting" any units active over multiple rays into a sum of multiple units aligned with each ray; see Figure 6 for an illustration.

First, we prove that $\boldsymbol{h}$ satisfies norm-ball interpolation constraints. Let $\boldsymbol{a}[\boldsymbol{w}^\top \boldsymbol{y} + b]_+$ denote any unit belonging to $\boldsymbol{f}_\ell$. Since this unit is active on a norm-ball centered on ray $\ell$, this implies that $\boldsymbol{w}^\top \boldsymbol{u}_\ell > 0$, and since the unit is inactive on $B_0$ we must have $b < 0$. Also, because we assume $\boldsymbol{u}_\ell^\top \boldsymbol{u}_m < 0$ for any $m \neq \ell$, we see that the projected unit $\boldsymbol{P}_\ell \boldsymbol{a}[\boldsymbol{w}^\top \boldsymbol{P}_\ell \boldsymbol{y} + b]_+ = \boldsymbol{P}_\ell \boldsymbol{a}[(\boldsymbol{w}^\top \boldsymbol{u}_\ell) \boldsymbol{u}_\ell^\top \boldsymbol{y} + b]_+$ is active on norm-balls centered on ray $\ell$ and inactive on norm-balls centered on any other ray. In particular, if $\boldsymbol{y}$ belongs to a norm-ball centered on ray $\ell$, then $\boldsymbol{h}_m(\boldsymbol{y}) = 0$ for all $m \neq \ell$. Therefore, if $\boldsymbol{y} \in B(\boldsymbol{x}_n^{(\ell)}, \rho)$ where $\boldsymbol{x}_n^{(\ell)}$ denotes a training sample along ray $\ell$, then

$$\boldsymbol{h}(\boldsymbol{y}) = \sum_{m=1}^{L} \boldsymbol{h}_m(\boldsymbol{y}) \tag{73}$$

$$= \boldsymbol{h}_\ell(\boldsymbol{y}) \tag{74}$$

$$= \boldsymbol{P}_\ell \boldsymbol{f}_\ell(\boldsymbol{P}_\ell \boldsymbol{y}) \tag{75}$$

$$= \boldsymbol{P}_\ell \boldsymbol{f}(\boldsymbol{P}_\ell \boldsymbol{y}) \tag{76}$$

$$= \boldsymbol{P}_\ell \boldsymbol{x}_n^{(\ell)} \tag{77}$$

$$= \boldsymbol{x}_n^{(\ell)} \tag{78}$$

which shows that $\boldsymbol{h}$ satisfies norm-ball interpolation constraints, as claimed.

Next, we show that if $\boldsymbol{h} \neq \boldsymbol{f}^*$, then $\boldsymbol{h}$ has strictly smaller representation cost. Let $\boldsymbol{u}(\boldsymbol{y}) = \boldsymbol{a}[\boldsymbol{w}^\top \boldsymbol{y} + b]_+$ denote any ReLU unit belonging to $\boldsymbol{f}^*$. Since we assume $\|\boldsymbol{w}\| = 1$, the contribution of unit $\boldsymbol{u}$ to the representation cost of $\boldsymbol{f}^*$ is $\|\boldsymbol{a}\|$. Let $\mathcal{R} \subset \{1, 2, ..., L\}$ index the subset of rays $\ell$ for which unit $\boldsymbol{u}$ is active over at least one norm-ball centered on that ray. In constructing the interpolant $\boldsymbol{h}$, the unit $\boldsymbol{u}$ get mapped to a sum of $|\mathcal{R}|$ units:

$$\sum_{\ell \in \mathcal{R}} \boldsymbol{P}_\ell \boldsymbol{a}[(\boldsymbol{P}_\ell \boldsymbol{w})^\top \boldsymbol{y} + b]_+$$

whose contribution to the representation cost of $\boldsymbol{h}$ is bounded above by

$$C = \sum_{\ell \in \mathcal{R}} \|\boldsymbol{P}_\ell \boldsymbol{a}\| \|\boldsymbol{P}_\ell \boldsymbol{w}\| = \sum_{\ell \in \mathcal{R}} |\boldsymbol{u}_\ell^\top \boldsymbol{a}| |\boldsymbol{u}_\ell^\top \boldsymbol{w}|.$$

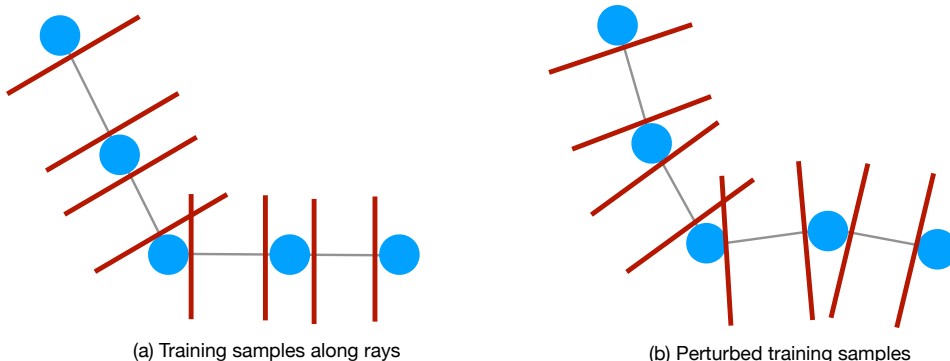

(a) Training samples along rays      (b) Perturbed training samples

Figure 7: Change in unit alignment of minimal representation cost solutions under perturbations of training samples belonging to rays. The red lines represent the ReLU boundaries of the representation cost minimizing solution satisfying norm-ball interpolation constraints. For the unperturbed samples in (a), all ReLU boundaries align perpendicular to the rays. For the perturbed samples in (b), the ReLU boundaries align perpendicular to the line segments connecting successive samples.

If $|\mathcal{R}| > 1$, then by Lemma 8 guarantees $C < \|\boldsymbol{a}\|$. And if $|\mathcal{R}| = 1$ then $C = |\boldsymbol{u}_\ell^\top \boldsymbol{a}||\boldsymbol{u}_\ell^\top \boldsymbol{w}| \leq \|\boldsymbol{a}\|$, with strict inequality unless $\boldsymbol{w}, \boldsymbol{a}$ belong to the span of $\boldsymbol{u}_\ell$. This implies that any min-cost solution must have all its units aligned with the rays, i.e., $\boldsymbol{f}^*$ must have the form $\boldsymbol{f}^*(\boldsymbol{y}) = \sum_{\ell=1}^L \boldsymbol{u}_\ell \phi_\ell(\boldsymbol{u}_\ell^\top \boldsymbol{y})$, where each $\phi_\ell$ is a univariate function. The representation cost of any function $\boldsymbol{f}^*$ in this form is the sum of the (1-D) representation costs of the $\phi_\ell$. Therefore, $\phi_\ell$ must also be the 1-D min-cost solution of the norm-ball constraints projected onto ray $\ell$, and so must have the form specified by (17).

$\square$

### C.2.1   Perturbations of samples along rays

As an extension of the above setting, suppose the training samples along rays $\boldsymbol{x}_n^{(\ell)}$ have been slightly perturbed, i.e., we consider training samples $\widetilde{\boldsymbol{x}}_n^{(\ell)} = \boldsymbol{x}_n^{(\ell)} + \boldsymbol{\epsilon}_n^{(\ell)}$ for some vectors $\boldsymbol{\epsilon}_n^{(\ell)}$ with $\|\boldsymbol{\epsilon}_n^{(\ell)}\| < \delta$ for some sufficiently small $\delta > 0$. In particular, we make the following two assumptions:

(A1) Let $\Delta_\ell := \{\widetilde{\boldsymbol{x}}_n^{(\ell)} - \widetilde{\boldsymbol{x}}_{n-1}^{(\ell)}\}_{n=1}^{N_\ell}$ be the collection of difference vectors between successive perturbed points along the $\ell$th ray (with $\widetilde{\boldsymbol{x}}_0^{(\ell)} := \boldsymbol{0}$). Assume that for all $\boldsymbol{v}_\ell \in \Delta_\ell$ and for all $\boldsymbol{v}_k \in \Delta_k$ with $k \neq \ell$, we have $\boldsymbol{v}_\ell^\top \boldsymbol{v}_k < 0$.

(A2) If $H$ is any halfspace not containing the origin that contains the ball $B(\tilde{\boldsymbol{x}}_n^{(\ell)}; \rho)$, then $H$ also contains all successive balls along that ray, i.e., $H$ contains $B(\tilde{\boldsymbol{x}}_m^{(\ell)}; \rho)$ for $m \geq n$.

Note that if the original points $\boldsymbol{x}_n^{(\ell)}$ belong to rays that satisfy the conditions of Theorem 3, then for sufficiently small $\delta$ the perturbed samples $\widetilde{\boldsymbol{x}}_n^{(\ell)}$ will satisfy assumptions (A1)&(A2) above.

We show in this case the norm-ball interpolating representation cost minimizer is a perturbed version of the min-cost solution for the unperturbed samples as identified in Theorem 3. In particular, the ReLU boundaries align with the line segments connecting successive points along the rays. See Figure 7 for an illustration in the case of $L = 2$ rays.

**Proposition 3.** *Suppose the training samples $X$ are a perturbation of data belong to a union of $L$ rays plus a sample at the origin, i.e., $X = \{\boldsymbol{0}\} \cup \{\widetilde{\boldsymbol{x}}_n^{(1)}\}_{n=1}^{N_1} \cup \cdots \cup \{\widetilde{\boldsymbol{x}}_n^{(L)}\}_{n=1}^{N_L}$ where $\widetilde{\boldsymbol{x}}_n^{(\ell)} = c_n^{(\ell)} \boldsymbol{u}_\ell + \boldsymbol{\epsilon}_n^{(\ell)}$ for some unit vector $\boldsymbol{u}_\ell$, constants $0 < c_1^{(\ell)} < c_2^{(\ell)} < \cdots < c_{N_\ell}^{(\ell)}$, and vectors $\boldsymbol{\epsilon}_n^{(\ell)}$. Assume (A1)&(A2) above hold. Then the minimizer $\boldsymbol{f}^*$ of (19) is unique and is given by*

$$\boldsymbol{f}^*(\boldsymbol{y}) = \sum_{\ell=1}^L \sum_{n=1}^{N_\ell} \boldsymbol{u}_n^{(\ell)} \phi_n^{(\ell)}((\boldsymbol{u}_n^{(\ell)})^\top (\boldsymbol{y} - \widetilde{\boldsymbol{x}}_{n-1}^{(\ell)})) \tag{79}$$

where $u_n^{(\ell)} = \frac{\widetilde{x}_n^{(\ell)} - \widetilde{x}_{n-1}^{(\ell)}}{\|\widetilde{x}_n^{(\ell)} - \widetilde{x}_{n-1}^{(\ell)}\|}$ and $\phi_n^{(\ell)}(t) = s_n^{(\ell)}([t - a_n^{(\ell)}] - [t - b_n^{(\ell)}]_+)$ with $a_n^{(\ell)} = \rho$, $b_n^{(\ell)} = \|x_n^{(\ell)} - x_{n-1}^{(\ell)}\| - \rho$, and $s_n^{(\ell)} = \|\widetilde{x}_n^{(\ell)} - \widetilde{x}_{n-1}^{(\ell)}\|/(b_n^{(\ell)} - a_n^{(\ell)})$.

*Proof.* Let $\boldsymbol{f}^*$ be any min-cost solution. Following the same steps as in the proof of Theorem 3, there is a minimal representative of $\boldsymbol{f}^*$ having the form $\boldsymbol{f}^*(\boldsymbol{y}) = \sum_k \boldsymbol{a}_k[\boldsymbol{w}_k^\top \boldsymbol{y} + b_k]_+$ where the active sets of all units in this representation have empty intersection with the ball $B(\boldsymbol{0}, \rho)$.

Let $\boldsymbol{f}_n^{(\ell)}$ denote the sum of all units in $\boldsymbol{f}^*$ whose active set boundary $\{\boldsymbol{y} \in \mathbb{R}^d : \boldsymbol{w}_k^\top \boldsymbol{y} + b_k = 0\}$ intersects the line segment connecting the training samples $\widetilde{\boldsymbol{x}}_{n-1}^{(\ell)}$ and $\widetilde{\boldsymbol{x}}_n^{(\ell)}$. By assumption (A2), this is equivalent to the sum of units that are active over balls $B(\widetilde{\boldsymbol{x}}_m^{(\ell)}, \rho)$ for $m \geq n$, and inactive for the balls with $m < n$. In particular, for $\boldsymbol{y} \in B(\widetilde{\boldsymbol{x}}_n^{(\ell)}, \rho)$, we have $\boldsymbol{f}^*(\boldsymbol{y}) = \sum_{m \leq n} \boldsymbol{f}_m^{(\ell)}(\boldsymbol{y})$. This implies $\boldsymbol{f}_1^{(\ell)}(\boldsymbol{y}) = \widetilde{\boldsymbol{x}}_1$ for all $\boldsymbol{y} \in B(\widetilde{\boldsymbol{x}}_1^{(\ell)}, \rho)$. Likewise, $\boldsymbol{f}_2^{(\ell)}(\boldsymbol{y}) = \widetilde{\boldsymbol{x}}_2^{(\ell)} - \widetilde{\boldsymbol{x}}_1^{(\ell)}$ for all $\boldsymbol{y} \in B(\widetilde{\boldsymbol{x}}_2^{(\ell)}; \rho)$, and so on, such that for all $n = 1, ..., N_\ell$ we have $\boldsymbol{f}_n^{(\ell)}(\boldsymbol{y}) = \widetilde{\boldsymbol{x}}_n^{(\ell)} - \widetilde{\boldsymbol{x}}_{n-1}^{(\ell)}$ for all $\boldsymbol{y} \in B(\widetilde{\boldsymbol{x}}_n^{(\ell)}; \rho)$.

Now we show how to construct a new interpolating function $\boldsymbol{h}$ having representation cost less than or equal to $\boldsymbol{f}^*$. Let $\boldsymbol{u}_n^{(\ell)} = \frac{\widetilde{\boldsymbol{x}}_n^{(\ell)} - \widetilde{\boldsymbol{x}}_{n-1}^{(\ell)}}{\|\widetilde{\boldsymbol{x}}_n^{(\ell)} - \widetilde{\boldsymbol{x}}_{n-1}^{(\ell)}\|}$ and define $\boldsymbol{P}_n^{(\ell)} = \boldsymbol{u}_n^{(\ell)}(\boldsymbol{u}_n^{(\ell)})^\top$, the orthogonal projector onto the span of the difference vector $\widetilde{\boldsymbol{x}}_n^{(\ell)} - \widetilde{\boldsymbol{x}}_{n-1}^{(\ell)}$. Note that the map $\boldsymbol{y} \mapsto \boldsymbol{P}_n^{(\ell)}\boldsymbol{y} + \widetilde{\boldsymbol{x}}_{n-1}^{(\ell)}$ is projection onto the affine line connecting samples $\widetilde{\boldsymbol{x}}_{n-1}^{(\ell)}$ and $\widetilde{\boldsymbol{x}}_n^{(\ell)}$. Consider the function

$$\boldsymbol{h} = \sum_{\ell=1}^{L} \sum_{n=1}^{N_\ell} \boldsymbol{h}_n^{(\ell)}$$

where

$$\boldsymbol{h}_n^{(\ell)}(\boldsymbol{y}) = \boldsymbol{P}_n^{(\ell)} \boldsymbol{f}_n^{(\ell)}(\boldsymbol{P}_n^{(\ell)}\boldsymbol{y} + \widetilde{\boldsymbol{x}}_{n-1}^{(\ell)}).$$

Put in words, $\boldsymbol{h}$ is constructed by aligning all inner-layer and outer-layer weights of units in $\boldsymbol{f}^*$ with the line segment connecting successive data points over which that unit first activates. See Figure 8 for an illustration.

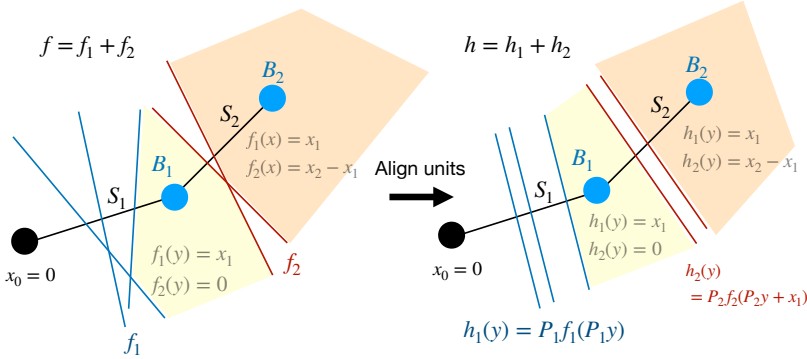

Figure 8: Illustration of "unit alignment" technique used in the proof of Proposition 3. The left panel shows an interpolant $\boldsymbol{f}$ whose ReLU boundaries (shown here as dark blue and dark red lines) are not aligned with the line segments $S_1$ and $S_2$ connecting successive training samples. The right panel shows how a new interpolant $\boldsymbol{h}$ can be constructed by projecting units in $\boldsymbol{f}$ onto the line segments $S_1$ and $S_2$, which reduces the representation cost.

Now we show that $\boldsymbol{h}$ satisfies norm-ball interpolation constraints. By assumption (A1), the function $\boldsymbol{h}_n^{(\ell)}$ is non-zero only on the norm-balls $B(\widetilde{\boldsymbol{x}}_m^{(\ell)}, \rho)$ for $m \geq n$. This implies that for all $\boldsymbol{y} \in B(\widetilde{\boldsymbol{x}}_n^{(\ell)}; \rho)$ we have $\boldsymbol{h}(\boldsymbol{y}) = \sum_{m \leq n} \boldsymbol{h}_n^{(\ell)}(\boldsymbol{y})$.

Observe that $\boldsymbol{h}(B(\boldsymbol{0}, \rho)) = \{\boldsymbol{0}\}$ since all functions $\boldsymbol{h}_n^{(\ell)}$ are zero on $B(\boldsymbol{0}, \rho)$. Also, for all $\boldsymbol{y} \in B(\widetilde{\boldsymbol{x}}_1^{(\ell)}; \rho)$ we have $\boldsymbol{P}_1^{(\ell)}\boldsymbol{y} \in B(\widetilde{\boldsymbol{x}}_1^{(\ell)}; \rho)$, and so

$$\boldsymbol{h}(\boldsymbol{y}) = \boldsymbol{h}_1^{(\ell)}(\boldsymbol{y}) = \boldsymbol{P}_1^{\ell} \boldsymbol{f}_1^{(\ell)}(\boldsymbol{P}_1^{(\ell)}\boldsymbol{y}) = \boldsymbol{P}_1^{(\ell)}\widetilde{\boldsymbol{x}}_1^{(\ell)} = \widetilde{\boldsymbol{x}}_1^{(\ell)}.$$

Similarly, the only terms in $\boldsymbol{h}$ active for $\boldsymbol{y} \in B(\widetilde{\boldsymbol{x}}_2^{(\ell)}; \rho)$ are $\boldsymbol{h}_1^{(\ell)}$ and $\boldsymbol{h}_2^{(\ell)}$, for which $\boldsymbol{h}_1^{(\ell)}(\boldsymbol{y}) = \boldsymbol{x}_1^{(\ell)}$, and $\boldsymbol{h}_2^{(\ell)}(\boldsymbol{y}) = \boldsymbol{P}_2^{(\ell)} \boldsymbol{f}_2^{(\ell)}(\boldsymbol{P}_2^{(\ell)} \boldsymbol{y} + \widetilde{\boldsymbol{x}}_1^{(\ell)}) = \boldsymbol{P}_2^{(\ell)}(\widetilde{\boldsymbol{x}}_2^{(\ell)} - \widetilde{\boldsymbol{x}}_1^{(\ell)}) = \widetilde{\boldsymbol{x}}_2^{(\ell)} - \widetilde{\boldsymbol{x}}_1^{(\ell)}$. This gives

$$\boldsymbol{h}(\boldsymbol{y}) = \boldsymbol{h}_1^{(\ell)}(\boldsymbol{y}) + \boldsymbol{h}_2^{(\ell)}(\boldsymbol{y}) = \widetilde{\boldsymbol{x}}_2^{(\ell)}.$$

Iterating this procedure for all $n = 1, ..., N_\ell$ we see that if $\boldsymbol{y} \in B(\widetilde{\boldsymbol{x}}_n^{(\ell)}, \rho)$ then $\boldsymbol{h}_n^{(\ell)}(\boldsymbol{y}) = \widetilde{\boldsymbol{x}}_n^{(\ell)} - \widetilde{\boldsymbol{x}}_{n-1}^{(\ell)}$, and so $\boldsymbol{h}(\boldsymbol{y}) = \sum_{m \leq n} \boldsymbol{h}_m^{(\ell)}(\boldsymbol{y}) = \boldsymbol{x}_n^{(\ell)}$, as claimed.

Following steps similar to the proof of Theorem 3, we can show that $R(\boldsymbol{h}) \leq R(\boldsymbol{f}^*)$ and with strict inequality if $\boldsymbol{h} \neq \boldsymbol{f}^*$. First, if any unit $\boldsymbol{u}(\boldsymbol{y}) = \boldsymbol{a}[\boldsymbol{w}^\top \boldsymbol{y} + b]_+$ in $\boldsymbol{f}^*$ is active over balls centered on multiple rays $\mathcal{R} \subset \{1, 2, ..., L\}$, then in the construction of $\boldsymbol{h}$ the unit $\boldsymbol{u}$ gets mapped to a sum of multiple units:

$$\sum_{\ell \in \mathcal{R}} \boldsymbol{u}_{n_\ell}^{(\ell)} (\boldsymbol{u}_{n_\ell}^{(\ell)})^\top \boldsymbol{a}[\boldsymbol{w}^\top \boldsymbol{u}_{n_\ell}^{(\ell)} (\boldsymbol{u}_{n_\ell}^{(\ell)})^\top (\boldsymbol{y} + \widetilde{\boldsymbol{x}}_{n_\ell-1}^{(\ell)}) + b]_+$$

for some $n_\ell$ with $1 \leq n_\ell \leq N_\ell$. The contribution of the sum of these units to the representation cost of $\boldsymbol{h}$ is less than or equal to $C = \sum_{\ell \in \mathcal{R}} |(\boldsymbol{u}_{n_\ell}^{(\ell)})^\top \boldsymbol{a}| |(\boldsymbol{u}_{n_\ell}^{(\ell)})^\top \boldsymbol{w}|$. By assumption (A1), we know $(\boldsymbol{u}_m^{(\ell)})^\top \boldsymbol{u}_n^{(p)} < 0$ for all $p \neq \ell$ and for all $m, n$. Therefore, by Lemma 8 we know $C < \|\boldsymbol{a}\|$. This shows $\boldsymbol{f}^*$ cannot have any units active over balls centered on different rays, since otherwise $\boldsymbol{h}$ has strictly smaller representation cost. Therefore, we have shown $\boldsymbol{f}^*$ as $\boldsymbol{f}^* = \sum_{\ell=1}^{L} \sum_{n=1}^{N_\ell} \boldsymbol{f}_n^{(\ell)}$. Additionally, we see that $\boldsymbol{h}_n^{(\ell)} = \boldsymbol{f}_n^{(\ell)}$, i.e., all inner- and outer-layer weight vectors of units in $\boldsymbol{f}_n^{(\ell)}$ are aligned with $\boldsymbol{u}_n^{(\ell)}$, since otherwise the representation cost of $\boldsymbol{f}^*$ could be reduced. Therefore, each $\boldsymbol{f}_n^{(\ell)}$ must have the form $\boldsymbol{f}_n^{(\ell)}(\boldsymbol{y}) = \boldsymbol{u}_n^{(\ell)} \psi_n^{(\ell)}((\boldsymbol{u}_n^{(\ell)})^\top \boldsymbol{y})$. Minimizing the $\psi_n^{(\ell)}$ separately under interpolation constraints, we arrive at the unique solution given in (79). $\qquad \square$

### C.3  Simplex Data

#### C.3.1  Simplex with one obtuse vertex

*Proof of Proposition 2.* Consider training samples $\boldsymbol{x}_1, \boldsymbol{x}_2, ..., \boldsymbol{x}_N \in \mathbb{R}^d$ whose convex hull is a $(N-1)$-simplex such that $\boldsymbol{x}_1$ makes an obtuse angle with all other vertices, i.e., $(\boldsymbol{x}_n - \boldsymbol{x}_1)^\top \boldsymbol{x}_1 < 0$ for all $n = 2, ..., N$. By Lemma 5, $\boldsymbol{f}^*$ is a norm-ball interpolating min-cost solution if and only if $\boldsymbol{g}^*(\boldsymbol{y}) = \boldsymbol{f}^*(\boldsymbol{y} - \boldsymbol{x}_1) + \boldsymbol{x}_1$ is a norm-ball interpolating min-cost solution for the translated points $\boldsymbol{0}, \boldsymbol{x}_2 - \boldsymbol{x}_1, ..., \boldsymbol{x}_N - \boldsymbol{x}_1 \in \mathbb{R}^d$. The latter configuration satisfies the hypotheses of Theorem 3 with a single training sample per ray. Therefore, the min-cost solution $\boldsymbol{g}^*$ of the translated points has units whose inner- and outer-layer weight vetors are aligned with $\boldsymbol{x}_n - \boldsymbol{x}_1$, $n = 2, ..., N$, and likewise for $\boldsymbol{f}^*$, since it is translation of $\boldsymbol{g}^*$. $\qquad \square$

#### C.3.2  Simplex with all acute vertices

*Justification of Conjecture 1.* For concreteness we focus on the case of three points $\boldsymbol{x}_1, \boldsymbol{x}_2, \boldsymbol{x}_3 \in \mathbb{R}^d$ whose convex hull is an acute triangle, and let $B_1, B_2, B_3$ be open balls of radius $\rho$ centered at $\boldsymbol{x}_1, \boldsymbol{x}_2, \boldsymbol{x}_3$ (respectively).

Let $\boldsymbol{f}$ be any norm-ball interpolating min-cost solution, and let $\boldsymbol{f}(\boldsymbol{y}) = \sum_k \boldsymbol{a}_k[\boldsymbol{w}_k^\top \boldsymbol{y} + b_k]_+ + \boldsymbol{V}\boldsymbol{x} + \boldsymbol{c}$ be any minimal representative of $\boldsymbol{f}$.

First, by properties of minimal representatives, none of the ReLU boundary sets $H_k = \{\boldsymbol{y} \in \mathbb{R}^d : \boldsymbol{w}_k^\top \boldsymbol{y} + b_k = 0\}$ intersect any of the balls centered at the training samples. Also, the active set of each unit in $\boldsymbol{f}$ must contain either one or two norm-balls, since otherwise the unit is either inactive over all balls or active over all balls, in which cases the unit can be removed or absorbed into unregularized linear part while strictly reducing the representation cost. By "reversing units", as in the proof of Theorem 3, we may transform the parameterization in such a way that the active set of every unit contains exactly one ball, and do this without changing the representation cost.

After this transformation, we may write

$$\boldsymbol{f} = \boldsymbol{f}_1 + \boldsymbol{f}_2 + \boldsymbol{f}_3 + \boldsymbol{\ell}$$

where $\boldsymbol{f}_i$ is a sum of units active only on the ball $B_i$ and no others, and where $\boldsymbol{\ell}(\boldsymbol{y}) = \boldsymbol{A}\boldsymbol{y} + \boldsymbol{b}$ is an affine function. Then we have

$$\forall \boldsymbol{y}_1 \in B_1, \quad \boldsymbol{f}(\boldsymbol{y}_1) = \boldsymbol{f}_1(\boldsymbol{y}_1) + \boldsymbol{A}\boldsymbol{y}_1 + \boldsymbol{b} = \boldsymbol{x}_1 \tag{80}$$
$$\forall \boldsymbol{y}_2 \in B_2, \quad \boldsymbol{f}(\boldsymbol{y}_2) = \boldsymbol{f}_2(\boldsymbol{y}_2) + \boldsymbol{A}\boldsymbol{y}_2 + \boldsymbol{b} = \boldsymbol{x}_2 \tag{81}$$
$$\forall \boldsymbol{y}_3 \in B_3, \quad \boldsymbol{f}(\boldsymbol{y}_3) = \boldsymbol{f}_3(\boldsymbol{y}_3) + \boldsymbol{A}\boldsymbol{y}_3 + \boldsymbol{b} = \boldsymbol{x}_3 \tag{82}$$

and so,

$$\forall \boldsymbol{y}_1 \in B_1, \quad \boldsymbol{f}_1(\boldsymbol{y}_1) = \boldsymbol{x}_1 - \boldsymbol{b} - \boldsymbol{A}\boldsymbol{y}_1 \tag{83}$$
$$\forall \boldsymbol{y}_2 \in B_2, \quad \boldsymbol{f}_2(\boldsymbol{y}_2) = \boldsymbol{x}_2 - \boldsymbol{b} - \boldsymbol{A}\boldsymbol{y}_2 \tag{84}$$
$$\forall \boldsymbol{y}_3 \in B_3, \quad \boldsymbol{f}_3(\boldsymbol{y}_3) = \boldsymbol{x}_3 - \boldsymbol{b} - \boldsymbol{A}\boldsymbol{y}_3 \tag{85}$$

Finding the min-cost solution then amounts to minimizing of the representation cost of $\boldsymbol{f}_1, \boldsymbol{f}_2, \boldsymbol{f}_3$ subject to the above constraints. This is made challenging by the fact that the constraints are coupled together by the parameters $\boldsymbol{A}$ and $\boldsymbol{b}$ of the affine part. However, *under the assumption $\boldsymbol{A} = \boldsymbol{0}$*, we prove below that the min-cost solution must have the conjectured form $\boldsymbol{f}^*$ as given in (86). However, since we cannot *a priori* assume $\boldsymbol{A} = \boldsymbol{0}$, this does not constitute a full proof.

Before proceeding, we give a lemma that will be used to lower bound $R(\boldsymbol{f})$ (assuming $\boldsymbol{A} = \boldsymbol{0}$).

**Lemma 10.** *Suppose $\boldsymbol{g}$ is a sum of ReLU units such that $\boldsymbol{g} = \boldsymbol{0}$ on a closed convex region $C \subset \mathbb{R}^d$ and $\boldsymbol{g}$ is constant and equal to $\boldsymbol{c}$ on a closed ball $B \subset \mathbb{R}^d$, and $\boldsymbol{g}$ has a minimal representative where all its units are active on $B$. Then*

$$R(\boldsymbol{g}) \geq \frac{2\|\boldsymbol{c}\|}{dist(B, C)}$$

*where $dist(B, C) = \min_{\boldsymbol{y} \in B, \boldsymbol{x} \in C} \|\boldsymbol{y} - \boldsymbol{x}\|$.*

*Proof.* Let $\boldsymbol{g}(\boldsymbol{y}) = \sum_k \boldsymbol{a}_k [\boldsymbol{w}_k^\top \boldsymbol{y} + b_k]_+$ be a minimal representative where each unit is active on $B$. For $\boldsymbol{g}$ to be constant on $B$ it must be the case that $\sum_k \boldsymbol{a}_k \boldsymbol{w}_k^\top = \boldsymbol{0}$. Let $\boldsymbol{y} = \boldsymbol{y}_0$ be any value for which $\|\partial \boldsymbol{g}(\boldsymbol{y}_0)\|$, the operator norm of the Jacobian of $\boldsymbol{g}$, is maximized. Since $\boldsymbol{g}$ is piecewise linear, we see that its Lipschitz constant $\mathrm{Lip}(\boldsymbol{g}) = \|\partial \boldsymbol{g}(\boldsymbol{y}_0)\|$. Let $I_0$ be the set of indices of active units at $\boldsymbol{y}_0$ so that $\partial \boldsymbol{g}(\boldsymbol{y}_0) = \sum_{k \in I_0} \boldsymbol{a}_k \boldsymbol{w}_k^\top$, and let $I_1$ be the complementary index set. Then because $\sum_k \boldsymbol{a}_k \boldsymbol{w}_k^\top = \boldsymbol{0}$, we have $\sum_{k \in I_0} \boldsymbol{a}_k \boldsymbol{w}_k^\top = -\sum_{k \in I_1} \boldsymbol{a}_k \boldsymbol{w}_k^\top$, and so $\|\sum_{k \in I_0} \boldsymbol{a}_k \boldsymbol{w}_k^\top\| = \|\sum_{k \in I_1} \boldsymbol{a}_k \boldsymbol{w}_k^\top\|$. Therefore,

$$2\,\mathrm{Lip}(\boldsymbol{g}) = 2\|\partial \boldsymbol{g}(\boldsymbol{y}_0)\| = 2 \left\| \sum_{k \in I_0} \boldsymbol{a}_k \boldsymbol{w}_k^\top \right\| = \left\| \sum_{k \in I_0} \boldsymbol{a}_k \boldsymbol{w}_k^\top \right\| + \left\| \sum_{k \in I_1} \boldsymbol{a}_k \boldsymbol{w}_k^\top \right\|$$
$$\leq \sum_k \|\boldsymbol{a}_k\| \|\boldsymbol{w}_k\| = R(\boldsymbol{g}).$$

Finally,

$$\mathrm{Lip}(\boldsymbol{g}) \geq \max_{\boldsymbol{x} \in C, \boldsymbol{y} \in B} \frac{\|\boldsymbol{g}(\boldsymbol{y}) - \boldsymbol{g}(\boldsymbol{x})\|}{\|\boldsymbol{y} - \boldsymbol{x}\|} = \frac{\|\boldsymbol{c}\|}{\min_{\boldsymbol{x} \in C, \boldsymbol{y} \in B} \|\boldsymbol{y} - \boldsymbol{x}\|} = \frac{\|\boldsymbol{c}\|}{dist(B, C)}.$$

Combining this with the previous inequality gives the claim. $\square$

Now, returning to the proof of the main claim, for $n = 1, 2, 3$, let $C_n$ be the closed convex region given by the intersection of all closed half-planes containing the balls $B_j$ and $B_k$, $j \neq n$, and $k \neq n$, $j \neq k$. By assumption, $\boldsymbol{f}_n$ vanishes on $C_n$ and $\boldsymbol{f}_n$ is constant and equal to $\boldsymbol{x}_n - \boldsymbol{b}$ on the closed ball $B_n$. So, by Lemma 10, we see that

$$R(\boldsymbol{f}_n) \geq \frac{2\|\boldsymbol{x}_n - \boldsymbol{b}\|}{\delta_n}$$

where $\delta_n = dist(B_n, C_n)$. Therefore, for all $\boldsymbol{b} \in \mathbb{R}^d$ we have

$$R(\boldsymbol{f}) = R(\boldsymbol{f}_1) + R(\boldsymbol{f}_2) + R(\boldsymbol{f}_3) \geq \sum_{n=1}^{3} \frac{2\|\boldsymbol{x}_n - \boldsymbol{b}\|}{\delta_n}$$

Let $\overline{x} \in \mathbb{R}^d$ be the minimizer of

$$\min_{b \in \mathbb{R}^d} \sum_{n=1}^{3} \frac{2\|x_n - b\|}{\delta_n}.$$

Then plugging in $b = \overline{x}$ above, we have the lower bound

$$R(f) \geq \sum_{n=1}^{3} \frac{2\|x_n - \overline{x}\|}{\delta_n},$$

which is independent of $b$.

Finally, simple calculations show that the conjectured min-cost solution $f^*$ specified in (86) has representation cost achieving this lower bound. Therefore, under the assumption $A = 0$, $f^*$ is a min-cost solution.

$\square$

Now we prove that, in the special case where the convex hull of the training points is an equilateral triangle, the norm-ball interpolator $f^*$ identified in Conjecture 1 is a min-cost solution (though not necessarily the unique min-cost solution). In this case, we may also give $f^*$ a more explicit form, as detailed below.

**Proposition 4.** *Suppose the convex hull of the training points $x_1, x_2, x_3 \in \mathbb{R}^d$ is an equilateral triangle. Assume the norm-balls $B_n := B(x_n, \rho)$ centered at each training point have radius $\rho < \|x_n - x_0\|/2$, $n = 1, 2, 3$, where $x_0 = \frac{1}{3}(x_1 + x_2 + x_3)$ is the centroid of the triangle. Then a minimizer $f^*$ of (19) is given by*

$$f^*(y) = u_1 \phi_1(u_1^\top (y - x_0)) + u_2 \phi_2(u_2^\top (y - x_0)) + u_3 \phi_3(u_3^\top (y - x_0)) + x_0, \quad (86)$$

*where $\phi_n(t) = s_n([t - a_n]_+ - [t - b_n]_+)$ with $u_n = \frac{x_n - x_0}{\|x_n - x_0\|}$, $a_n = -\frac{1}{2}\|x_n - x_0\| + \rho$, $b_n = \|x_n - x_0\| - \rho$, and $s_n = \|x_n - x_0\|/(b_n - a_n)$.*

*Proof.* By translation and scale invariance of min-cost solutions, without loss of generality we may assume $x_1, x_2, x_3 \in \mathbb{R}^d$ are unit-norm vectors and mean-zero (i.e., the triangle centroid $x_0 = 0$). In this case, the assumption on $\rho$ translates to $\rho < 1/2$. Additionally, it suffices to prove the claim in the case $d = 2$. This is because, if $x_1, x_2, x_3 \in \mathbb{R}^d$ are the vertices of an equilateral triangle whose centroid is at the origin, then these points are contained in a two-dimensional subspace $\mathcal{S} \subset \mathbb{R}^d$. And if we let $P \in \mathbb{R}^{d \times 2}$ be a matrix whose columns are an orthonormal basis for $\mathcal{S}$, then by Theorem 2, $f$ is a min-cost solution if and only if $f(y) = P f_0(P^\top y)$, where $f_0 : \mathbb{R}^2 \to \mathbb{R}^2$ is a min-cost solution under the constraints $f_0(B(P^\top x_n, \rho)) = \{P^\top x_n\}$ for all $n = 1, 2, 3$. Therefore, the problem reduces to finding a min-cost solution of the projected points $P^\top x_1, P^\top x_2, P^\top x_3 \in \mathbb{R}^2$ whose convex hull is an equilateral triangle in $\mathbb{R}^2$.

So now let $f : \mathbb{R}^2 \to \mathbb{R}^2$ be any min-cost solution under the assumption $x_1, x_2, x_3 \in \mathbb{R}^2$ are unit-norm, have zero mean, and $\rho < 1/2$. By reversing units as necessary, there exists a minimal representative of $f$ that can be put in the form $f(y) = f_1(y) + f_2(y) + f_3(y) + Ay + c$, such that $f_n(y)$ is a sum of ReLU units, all of which are active on $B_n$, and all of which are inactive on $B_j$ for $j \neq n$.

Let $Q$ be the reflection matrix $Q = 2x_1 x_1^\top - I$, which reflects points across the line spanned by $x_1$. In particular, $y \in B_1$ then $Qy \in B_1$ while if $y \in B_2$ then $Qy \in B_3$ (and vice versa). Also, $Qx_1 = x_1$, $Qx_2 = x_3$, and $Qx_3 = x_2$. Define $g(y) = \frac{1}{2}\left(f(y) + Q^{-1}f(Qy)\right)$. Then it is easy to check that $g$ satisfies interpolation constraints, and since $Q$ is unitary, we have $R(g) \leq \frac{1}{2}R(f) + \frac{1}{2}R(Q \circ f \circ Q^{-1}) = \frac{1}{2}R(f) + \frac{1}{2}R(f) = R(f)$. Furthermore, since $f$ is a min-cost solution, we must have $R(g) = R(f)$. Additionally, since none of the units making up $f$ are active over more than one ball, neither are the units making up $g$, which implies no pair of units belonging to $g$ combine to form an affine function. Therefore, we may write $g(y) = g_1(y) + g_2(y) + g_3(y) + By + v$, where $B = A + Q^{-1}AQ$, such that each $g_n$ is a sum of ReLU units, all of which are active on $B_n$, and all of which are inactive on $B_j$ for $j \neq n$.

Let $U$ be a rotation matrix by 120 degrees such that $x_2 = Ux_1$, $x_3 = Ux_2$, and $x_1 = Ux_3$. Consider the symmetrized version of $g$ given by $h(y) := \frac{1}{3}(g(y) + U^{-1}g(Uy) + U^{-2}g(U^2 y))$.

Since for all $\boldsymbol{y} \in B_n$ we have $\boldsymbol{U}\boldsymbol{y} \in B_{n+1}$ (with indices understood modulo 3), it is easy to verify that $\boldsymbol{h}$ satisfies interpolation constraints. Also, since $\boldsymbol{U}$ is unitary, we have $R(\boldsymbol{h}) \leq R(\boldsymbol{g}) \leq R(\boldsymbol{f})$, which implies $R(\boldsymbol{h}) = R(\boldsymbol{f})$ since $\boldsymbol{f}$ is a min-cost solution. Again, since none of the units making up $\boldsymbol{g}$ are active over more than one ball, neither are the units making up $\boldsymbol{h}$, which implies no pair of units belonging to $\boldsymbol{h}$ combine to form an affine function. And so, we may write $\boldsymbol{h}(\boldsymbol{y}) = \boldsymbol{h}_1(\boldsymbol{y}) + \boldsymbol{h}_2(\boldsymbol{y}) + \boldsymbol{h}_3(\boldsymbol{y}) + \boldsymbol{C} + \boldsymbol{u}$, such that each $\boldsymbol{h}_n$ is a sum of ReLU units, all of which are active on $B_n$, and all of which are inactive on $B_j$ for $j \neq n$.

Observe that $\boldsymbol{u} = \frac{1}{3}(\boldsymbol{I} + \boldsymbol{U}^{-1} + \boldsymbol{U}^{-2})\boldsymbol{v} = \boldsymbol{0}$. Also, we have $\boldsymbol{C} = \frac{1}{3}(\boldsymbol{B} + \boldsymbol{U}^{-1}\boldsymbol{B}\boldsymbol{U} + \boldsymbol{U}^{-2}\boldsymbol{B}\boldsymbol{U}^2)$. This implies $\boldsymbol{C}$ commutes with $\boldsymbol{U}$, because it is easy to see that $\boldsymbol{C} = \boldsymbol{U}^{-1}\boldsymbol{C}\boldsymbol{U}$. Additionally, $\boldsymbol{C}$ commutes with $\boldsymbol{Q}$, because it is easy to see $\boldsymbol{B} = \boldsymbol{Q}^{-1}\boldsymbol{B}\boldsymbol{Q}$, and by properties of rotations/reflections we have $\boldsymbol{U}\boldsymbol{Q} = \boldsymbol{Q}\boldsymbol{U}^{-1}$, which implies

$$
\begin{aligned}
\boldsymbol{Q}^{-1}\boldsymbol{C}\boldsymbol{Q} &= \frac{1}{3}(\boldsymbol{Q}^{-1}\boldsymbol{B}\boldsymbol{Q} + \boldsymbol{Q}^{-1}\boldsymbol{U}^{-1}\boldsymbol{B}\boldsymbol{U}\boldsymbol{Q} + \boldsymbol{Q}^{-1}\boldsymbol{U}^{-2}\boldsymbol{B}\boldsymbol{U}^2\boldsymbol{Q}) \\
&= \frac{1}{3}(\boldsymbol{Q}^{-1}\boldsymbol{B}\boldsymbol{Q} + \boldsymbol{Q}^{-1}\boldsymbol{U}^{-1}\boldsymbol{B}\boldsymbol{U}\boldsymbol{Q} + \boldsymbol{Q}^{-1}\boldsymbol{U}^{-2}\boldsymbol{B}\boldsymbol{U}^2\boldsymbol{Q}) \\
&= \frac{1}{3}(\boldsymbol{Q}^{-1}\boldsymbol{B}\boldsymbol{Q} + (\boldsymbol{U}\boldsymbol{Q})^{-1}\boldsymbol{B}(\boldsymbol{U}\boldsymbol{Q}) + (\boldsymbol{U}\boldsymbol{Q})^{-1}\boldsymbol{U}^{-1}\boldsymbol{B}\boldsymbol{U}(\boldsymbol{U}\boldsymbol{Q})) \\
&= \frac{1}{3}(\boldsymbol{Q}^{-1}\boldsymbol{B}\boldsymbol{Q} + \boldsymbol{U}\boldsymbol{Q}^{-1}\boldsymbol{B}\boldsymbol{Q}\boldsymbol{U}^{-1} + \boldsymbol{U}\boldsymbol{Q}^{-1}\boldsymbol{U}^{-1}\boldsymbol{B}\boldsymbol{U}\boldsymbol{Q}\boldsymbol{U}^{-1}) \\
&= \frac{1}{3}(\boldsymbol{Q}^{-1}\boldsymbol{B}\boldsymbol{Q} + \boldsymbol{U}\boldsymbol{Q}^{-1}\boldsymbol{B}\boldsymbol{Q}\boldsymbol{U}^{-1} + \boldsymbol{U}^2\boldsymbol{Q}^{-1}\boldsymbol{B}\boldsymbol{Q}\boldsymbol{U}^{-2}) \\
&= \frac{1}{3}(\boldsymbol{B} + \boldsymbol{U}\boldsymbol{B}\boldsymbol{U}^{-1} + \boldsymbol{U}^2\boldsymbol{B}\boldsymbol{U}^{-2}) \\
&= \frac{1}{3}(\boldsymbol{B} + \boldsymbol{U}^{-2}\boldsymbol{B}\boldsymbol{U}^2 + \boldsymbol{U}^{-1}\boldsymbol{B}\boldsymbol{U}) \\
&= \boldsymbol{C},
\end{aligned}
$$

which shows $\boldsymbol{C}$ commutes with $\boldsymbol{Q}$. Now we show that any matrix $\boldsymbol{C}$ that commutes with both $\boldsymbol{U}$ and $\boldsymbol{Q}$ is a scaled identity matrix. Let $\boldsymbol{x}_1^\perp$ denote a unit-vector perpendicular to $\boldsymbol{x}_1$, such that $\{\boldsymbol{x}_1, \boldsymbol{x}_1^\perp\}$ form an orthonormal basis for $\mathbb{R}^2$. First, we know $\boldsymbol{Q}$ has eigenvectors $\boldsymbol{x}_1$ and $\boldsymbol{x}_1^\perp$ with eigenvalues $1$ and $-1$, respectively. And so $\boldsymbol{Q}\boldsymbol{C}\boldsymbol{x}_1 = \boldsymbol{C}\boldsymbol{Q}\boldsymbol{x}_1 = \boldsymbol{C}\boldsymbol{x}_1$, while $\boldsymbol{Q}\boldsymbol{C}\boldsymbol{x}_1^\perp = \boldsymbol{C}\boldsymbol{Q}\boldsymbol{x}_1^\perp = -\boldsymbol{C}\boldsymbol{x}_1^\perp$, which implies $\boldsymbol{C}\boldsymbol{x}_1 = a\boldsymbol{x}_1$ and $\boldsymbol{C}\boldsymbol{x}_1^\perp = b\boldsymbol{x}_1^\perp$ for some $a, b \in \mathbb{R}$, i.e., $\boldsymbol{x}_1$ and $\boldsymbol{x}_1^\perp$ are eigenvectors of $\boldsymbol{C}$ with eigenvalues $a$ and $b$, respectively. Furthermore, we have $\boldsymbol{C}\boldsymbol{U}\boldsymbol{x}_1 = \boldsymbol{U}\boldsymbol{C}\boldsymbol{x}_1 = a\boldsymbol{U}\boldsymbol{x}_1$ and $\boldsymbol{C}\boldsymbol{U}\boldsymbol{x}_1^\perp = \boldsymbol{U}\boldsymbol{C}\boldsymbol{x}_1^\perp = b\boldsymbol{U}\boldsymbol{x}_1$. This implies $\boldsymbol{U}\boldsymbol{x}_1$ and $\boldsymbol{U}\boldsymbol{x}_1^\perp$ are also eigenvectors of $\boldsymbol{C}$ with eigenvalues $a$ and $b$, respectively. But since $\boldsymbol{x}_1$ and $\boldsymbol{U}\boldsymbol{x}_1$ are linearly independent, and likewise so are $\boldsymbol{x}_1^\perp$ and $\boldsymbol{U}\boldsymbol{x}_1^\perp$, the only way this could be true is if $a = b$, i.e., every vector in $\mathbb{R}^2$ is an eigenvector of $\boldsymbol{C}$ for the same eigenvalue $\lambda \in \mathbb{R}$, which implies $\boldsymbol{C} = \lambda\boldsymbol{I}$ for some $\lambda \in \mathbb{R}$.

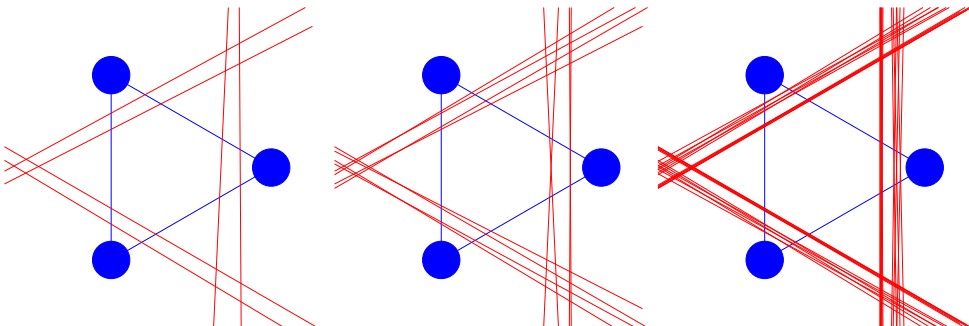

Figure 9: Illustration of ReLU boundaries (in red) of original interpolant $\boldsymbol{f}$ (left), the function $\boldsymbol{g}$ (middle) obtained by enforcing reflection symmetry, and the function $\boldsymbol{h}$ (right) obtained by enforcing both reflection and rotation symmetry.

Therefore, we have shown that every min-cost solution $\boldsymbol{f}$ maps to a min-cost solution $\boldsymbol{h}$ of the form

$$
\boldsymbol{h}(\boldsymbol{y}) = \boldsymbol{h}_1(\boldsymbol{y}) + \boldsymbol{h}_2(\boldsymbol{y}) + \boldsymbol{h}_3(\boldsymbol{y}) + \lambda\boldsymbol{I}
$$

for some $\lambda \in \mathbb{R}$, where each $\boldsymbol{h}_n$ is a sum of ReLU units, all of which are active on $B_n$ and all of which are inactive on $B_j$, $j \neq n$. In particular, $\boldsymbol{h}_n(\boldsymbol{y}) = \boldsymbol{x}_n - \lambda \boldsymbol{y}$ for all $\boldsymbol{y} \in B_n$ and $\boldsymbol{h}_n(\boldsymbol{y}) = \boldsymbol{0}$ for all $\boldsymbol{y} \in C_n$, where $C_n$ is the intersection of all half-planes containing the balls $B_j$, $j \neq n$.

This means we can write $\boldsymbol{h}(\boldsymbol{y}) = \sum_{k=1}^{K} \boldsymbol{a}_k [\boldsymbol{w}_k^\top \boldsymbol{y} + b_k]_+ + \lambda \boldsymbol{I}$ and $\boldsymbol{h}_n(\boldsymbol{y}) = \sum_{k \in \mathcal{A}_n} \boldsymbol{a}_k [\boldsymbol{w}_k^\top \boldsymbol{y} + b_k]_+$ for some index sets $\mathcal{A}_1, \mathcal{A}_2, \mathcal{A}_3$ partitioning $\{1, ..., K\}$ so that $R(\boldsymbol{h}) = \sum_{k=1}^{K} \|\boldsymbol{a}_k\| = \sum_{n=1}^{3} \sum_{k \in \mathcal{A}_n} \|\boldsymbol{a}_k\|$. Let $R_0(\cdot)$ denote the representation cost of a function computed without an unregularized linear part, i.e., define $R_0$ analogously to $R$ except where the model class $\boldsymbol{h}_\theta$ in (6) is constrained to have $\boldsymbol{V} = \boldsymbol{0}$. Then, since the realizations of the $\boldsymbol{h}_n$ functions considered above do not have a linear part, we see that $\sum_{k \in \mathcal{A}_n} \|\boldsymbol{a}_k\| \geq R_0(\boldsymbol{h}_n)$ and so $R(\boldsymbol{h}) = \sum_{n=1}^{3} \sum_{k \in \mathcal{A}_n} \|\boldsymbol{a}_k\| \geq R_0(\boldsymbol{h}_1) + R_0(\boldsymbol{h}_2) + R_0(\boldsymbol{h}_3)$.

Now we show how lower bound $R_0(\boldsymbol{h}_n)$ for all $n = 1, 2, 3$. Let $\boldsymbol{x}_n^\perp$ denote a unit-vector perpendicular to $\boldsymbol{x}_n$, such that $\{\boldsymbol{x}_n, \boldsymbol{x}_n^\perp\}$ form an orthonormal basis for $\mathbb{R}^2$. Consider the univariate functions $h_n^\parallel(t) := \boldsymbol{x}_n^\top \boldsymbol{h}_n(\boldsymbol{x}_n t)$ and $h_n^\perp(t) := (\boldsymbol{x}_n^\perp)^\top \boldsymbol{h}_n(\boldsymbol{x}_n^\perp t + \boldsymbol{x}_n)$. Here $h_n^\parallel$ is the projection of $\boldsymbol{h}$ onto the line spanned by $\boldsymbol{x}_n$, and $h_n^\perp$ is a projection onto the line perpendicular to $\boldsymbol{x}_n$ passing through the point $\boldsymbol{x}_n$. In particular, by the constraints on $\boldsymbol{h}_n$, we see that $h_n^\parallel$ and $h_n^\perp$ satisfy the constraints

$$h_n^\parallel(t) = \begin{cases} 0 & \text{if } t < -1/2 + \rho \\ 1 - \lambda t & \text{if } t > 1 - \rho \end{cases} \tag{87}$$

and $h_n^\perp(t) = -\lambda t$ if $|t| \leq \rho$.

**Claim 1:** For all $n = 1, 2, 3$, $R_0(\boldsymbol{h}_n) \geq R_0(h_n^\parallel) + R_0(h_n^\perp)$. *Proof:* Let $\boldsymbol{h}_n(\boldsymbol{y}) = \sum_k \boldsymbol{a}_k [\boldsymbol{w}_k^\top \boldsymbol{y} + b_k]_+ + \boldsymbol{c}$ be any realization of $\boldsymbol{h}_n$, whose representation cost is $C = \sum_k \frac{1}{2} (\|\boldsymbol{a}_k\|^2 + \|\boldsymbol{w}_k\|^2)$. Then realizations of $h_n^\parallel$ and $h_n^\perp$ are given by

$$h_n^\parallel(t) = \boldsymbol{x}_n^\top \boldsymbol{h}_i(\boldsymbol{x}_i t) = \sum_k (\boldsymbol{x}_n^\top \boldsymbol{a}_k)[(\boldsymbol{x}_n^\top \boldsymbol{w}_k)t + b_k]_+ + \boldsymbol{x}_n^\top \boldsymbol{c}. \tag{88}$$

$$h_n^\perp(t) = (\boldsymbol{x}_n^\perp)^\top \boldsymbol{h}_n(\boldsymbol{x}_n^\perp t + \boldsymbol{x}_n) = \sum_k ((\boldsymbol{x}_n^\perp)^\top \boldsymbol{a}_k)[((\boldsymbol{x}_n^\perp)^\top \boldsymbol{w}_k)t + b_k + \boldsymbol{w}_k^\top \boldsymbol{x}_n]_+ + (\boldsymbol{x}_n^\perp)^\top \boldsymbol{c}, \tag{89}$$

whose representation costs $C^\parallel$ and $C^\perp$, respectively, are given by

$$C^\parallel = \sum_k \frac{1}{2} \left( (\boldsymbol{x}_n^\top \boldsymbol{a}_k)^2 + (\boldsymbol{x}_n^\top \boldsymbol{w}_k)^2 \right) \tag{90}$$

$$C^\perp = \sum_k \frac{1}{2} \left( ((\boldsymbol{x}_n^\perp)^\top \boldsymbol{a}_k)^2 + ((\boldsymbol{x}_n^\perp)^\top \boldsymbol{w}_k)^2 \right) \tag{91}$$

and by the Pythagorean Theorem we see that $C = C^\parallel + C^\perp$. Therefore, $C \geq R_0(h_n^\parallel) + R_0(h_n^\perp)$ and finally minimizing over all realizations of $\boldsymbol{h}_n$ gives the claim.

**Claim 2:** For all $n = 1, 2, 3$, $R_0(h_n^\parallel) \geq \left| \frac{1 - \lambda\beta}{\beta - \alpha} \right| + \left| \frac{1 - \lambda\alpha}{\beta - \alpha} \right|$, where $\beta = 1 - \rho$ and $\alpha = -1/2 + \rho$.

*Proof:* By results in Savarese et al. [2019], $R_0(h_n^\parallel) \geq R_0(p)$ where $p(t)$ is the function satisfying the same constraints as $h_n^\parallel$ given in (87) while linearly interpolating over the interval $t \in [\alpha, \beta]$. From the formula $R_0(p) = \max\{\int |p''(t)|dt, |p'(\infty) + p'(-\infty)|\}$ as established in Savarese et al. [2019], we can show directly that $R_0(p) = \left| \frac{1 - \lambda\beta}{\beta - \alpha} \right| + \left| \frac{1 - \lambda\alpha}{\beta - \alpha} \right|$, which gives the claimed bound.

**Claim 3:** For all $n = 1, 2, 3$, $R_0(h_n^\perp) \geq |\lambda|$. *Proof:* The function $q(t)$ with minimal $R_0$-cost satisfying the constraint $q(t) = -\lambda t$ for $|t| \leq \rho$ is a single ReLU unit plus a constant: $q(t) = -\lambda[t + \rho]_+ + \lambda\rho$, which has $R_0$-cost $|\lambda|$.

Putting the above claims together, we see that

$$R_0(\boldsymbol{h}_n) \geq \left| \frac{1 - \lambda\beta}{\beta - \alpha} \right| + \left| \frac{1 - \lambda\alpha}{\beta - \alpha} \right| + |\lambda| \tag{92}$$

$$\geq \left| \frac{2 - \lambda(\beta + \alpha)}{\beta - \alpha} \right| + |\lambda| \tag{93}$$

$$\geq \frac{2 - |\lambda|(\beta + \alpha)}{\beta - \alpha} + |\lambda| \tag{94}$$

$$= \frac{2}{\beta - \alpha} + \frac{-2\alpha}{\beta - \alpha} |\lambda| \tag{95}$$

$$\geq \frac{2}{\beta - \alpha} \tag{96}$$

where in the last inequality we used the fact that $\frac{-2\alpha}{\beta - \alpha} > 0$ since $\alpha = -1/2 + \rho < 0$ and $\beta - \alpha = 3/2 - 2\rho > 0$.

Therefore, $R(\boldsymbol{f}) = R(\boldsymbol{h}) \geq R_0(\boldsymbol{h}_1) + R_0(\boldsymbol{h}_2) + R_0(\boldsymbol{h}_3) \geq \frac{6}{\beta - \alpha}$. Also, the function $\boldsymbol{f}^*$ given by

$$\boldsymbol{f}^* = \boldsymbol{f}_1^* + \boldsymbol{f}_2^* + \boldsymbol{f}_3^*$$

where

$$\boldsymbol{f}_n^*(\boldsymbol{y}) = \frac{\boldsymbol{x}_n}{\beta - \alpha} ([\boldsymbol{x}_n^\top \boldsymbol{y} - \alpha]_+ - [\boldsymbol{x}_n^\top \boldsymbol{y} - \beta]_+)$$

satisfies norm-ball interpolation constraints and $R(\boldsymbol{f}^*) = \frac{6}{\beta - \alpha}$. Hence, $\boldsymbol{f}^*$ is a min-cost solution. $\quad\square$

# D  Additional simulations

We train a one-hidden-layer ReLU network *without* a skip connection using the setting we used in Figure 1. As can be seen from Figure 10 we get a similar result for the NN denoiser with and without the skip connection. Therefore in Appendix D.1 we use one-hidden-layer ReLU network *without* a skip connection.

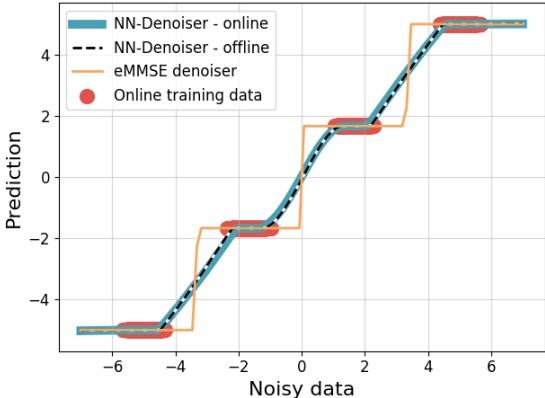

Figure 10:  **NN denoiser vs eMMSE denoiser.** We trained a one-hidden-layer ReLU network on a denoising task. The clean dataset has four points equally spaced in the interval $[-5, 5]$, and the noisy samples are generated by adding zero-mean Gaussian noise with $\sigma = 1.5$. We use $\lambda = 10^{-5}$ in both setting. The figure shows the denoiser output as a function of its input for: (1) NN denoiser trained online using (7) for $100K$ iterations, (2) NN denoiser trained offline using (8) with $M = 9000$ and $20K$ epochs, and (3) the eMMSE denoiser (4).

## D.1  MNIST

We use the MNIST dataset to verify various properties. First, the offline and online solutions achieve approximately the same test MSE when trained on a subset of the MNIST dataset (Figure 11). Second, to show that the fact that NN denoiser does not converge to the eMMSE denoiser is not due to approximation error (Figure 12). Lastly, to present the critical noise level in which representation cost minimizer $f_{1D}^*$ has strictly lower MSE than the eMMSE, for all smaller noise levels (Figure 13).

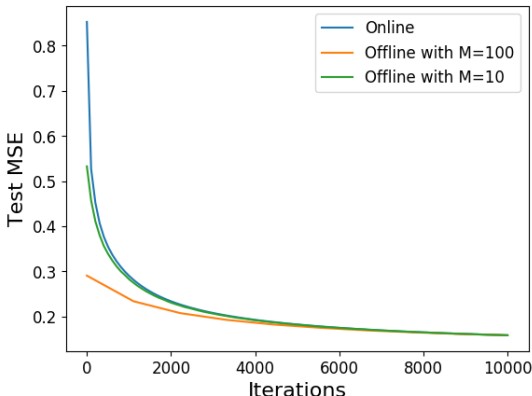

Figure 11: **Online setting vs offline setting for MNIST denoiser.** We train a one-hidden layer ReLU network on a subset of $N = 100$ MNIST images for 10K iterations. We use a Gaussian noise with zero mean and $\sigma = 0.1$.

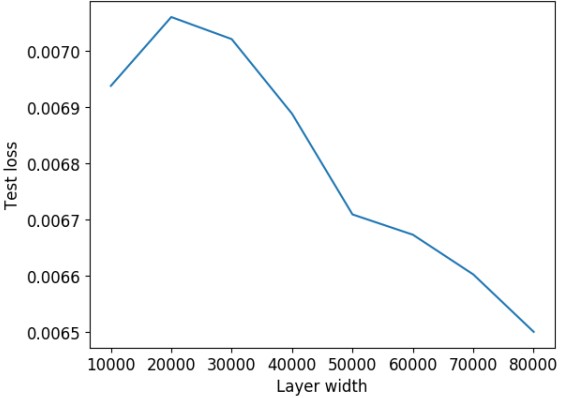

Figure 12: **Test loss vs. layer width for MNIST denoiser.** We train a one hidden layer ReLU network on MNIST denoiser task using 7 for $93K$ iterations with a fixed learning rate. We use a Gaussian noise with zero mean and $\sigma = 0.1$. The figure shows the test loss vs. layer width.

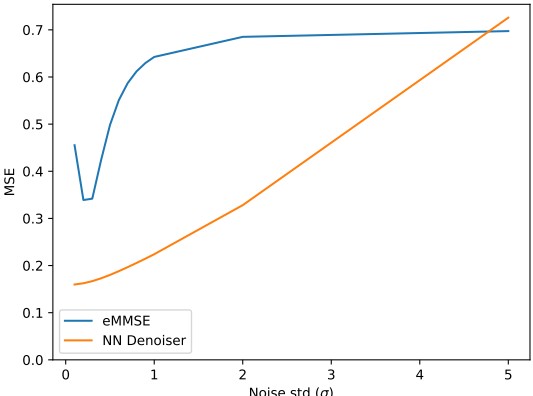

Figure 13: **MSE vs. Noise std.** We train a one-hidden layer ReLU network on a subset of $N = 100$ MNIST images (the range of each pixel is $[0, 1]$) for 10K iterations with fixed LR. We use a Gaussian noise with zero mean. The figure shows the MSE vs noise std ($\sigma$) for NN denoiser (orange line) and for eMMSE denoiser (blue line). Note that the eMMSE is dependent on $\sigma$ (4). For low noise levels, the eMMSE output is one of the training set images. For moderate noise levels, the eMMSE output is a weighted sum of the training set images. For high noise levels, the eMMSE output is the mean of the training set images.

### D.2 Three non-colinear training samples

We show in Figures 14 and 15 that for $N = 3$ training points from the MNIST dataset that forming a triangle in $d = 2$ dimensions the empirical minimizer obtained using noisy samples and weight decay regularization agrees well with the form of the exact representation cost minimizer predicted by Proposition 2 and Conjecture 4.

### D.3 Empirical validation of the subspace assumption

We validated that the following image datasets are (approximately) low rank:

- CIFAR10
- CINIC10
- Tiny ImageNet (a lower resolution version of ImageNet, enabling us to use SVD)
- BSD (a denoising benchmark composed from $128X1600$ patches of size $40X40$ cropped from $400$ images [Zhang et al., 2017])

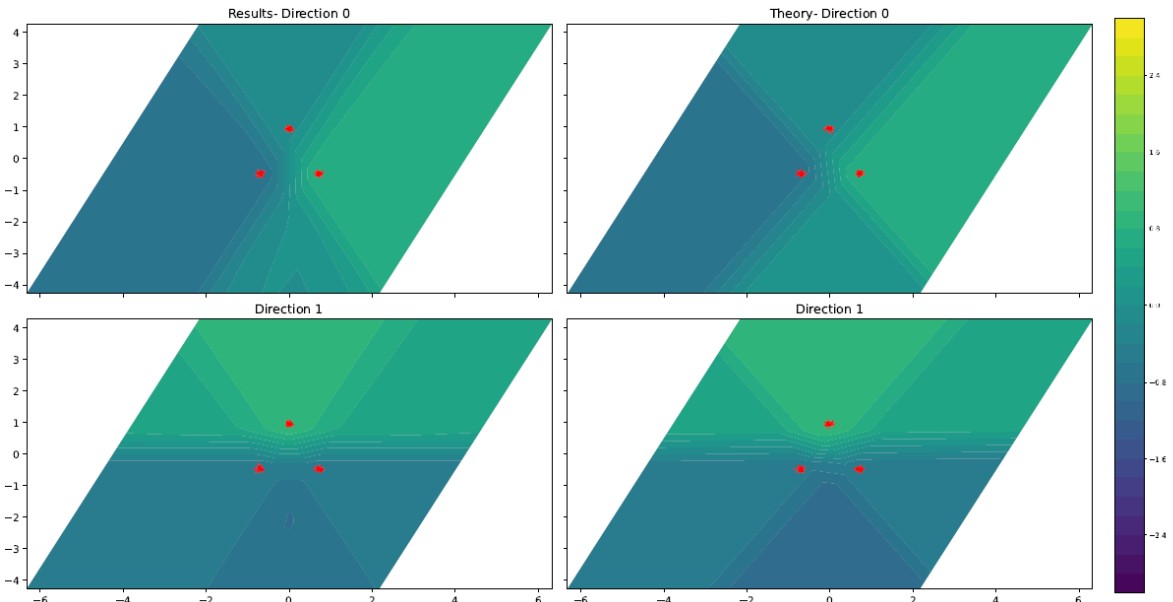

Figure 14: **Function space view of a denoiser, trained for** $3$ **MNIST data points (Acute Angle).** Here, we compare empirical results (left) with our theoretical results (right). We compare the function-space view for inputs from the data plane, with respect to the model output in each of the data directions. For the empirical results, we choose 3 random MNIST data points under the same label, and under the condition that they form an acute triangle ($64°$, for this figure). We trained a single-layer FC ReLU network with linear residual connection for 1M epochs, with weight decay of $1E-8$ (as described in our model), and ADAM optimizer with learning rate $1E-5$.

Table 1: We applied a Singular Value Decomposition (SVD) for each of the above datasets, and calculated the relative number of Singular Values (SV) needed to achieve a given percentile of the energy (for the average vector).

| Dataset | 95% | 99% | 99.9% |
|---|---|---|---|
| CIFAR10 | 0.8% | 7.5% | 30% |
| CINIC10 | 1% | 23% | 41% |
| Tiny ImageNet | 1.6% | 20% | 36% |
| BSD | 0.1% | 1.6% | 4.5% |

As can be seen from Table 1 all the datasets that we used are (approximately) low rank.

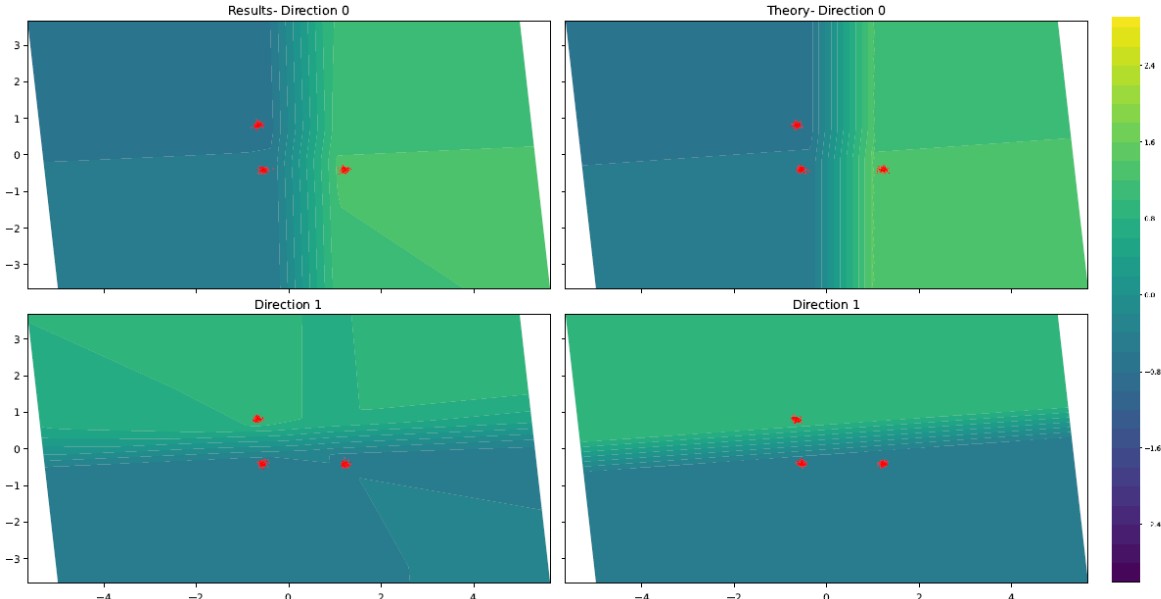

Figure 15: **Function space view of a denoiser, trained for 3 MNIST data points (Obtuse Angle).** Here, we compare empirical results (left) with our theoretical results (right). We compare the function-space view for inputs from the data plane, with respect to the model output in each of the data directions. For the empirical results, we added a single data point to the two previously chosen for Figure 14 under the condition that the three points form an obtuse triangle (95°, for this figure). We trained a single-layer FC as described in Figure 14. As predicted, the function we have converged to for data forming an obtuse triangle is noticeably different form the function we converged to when the data was forming an acute triangle.

