# OpenReview forum: "How do Minimum-Norm Shallow Denoisers Look in Function Space?"
_NeurIPS.cc/2023/Conference — NeurIPS 2023 poster_

### Official Review · Reviewer_7Tj7 · 2023-06-28

**Soundness:** 3 good
**Presentation:** 4 excellent
**Contribution:** 3 good
**Rating:** 6
**Confidence:** 3

**Summary:**

The paper studies the shape and properties of one layer networks on a denoising problem. Specifically, the authors compute a closed-form solution for a NN trained offline with regularization and show that its ability to generalize is better than the eMMSE estimator -- which acts as a piece-wise constant function -- in a low-noise regime and in 1d. Then they provide results on multivariate cases where the training data points are contained in a lower dimensional subspace, showing that the image of the network is also contained in the subspace, and deriving closed-forms for specific cases where the data are aligned.

**Strengths:**

Regarding the presentation, the paper is clearly written and each theoretical result is well-explained, making the reading easier. Regarding the content, the theorems seem new and provide interesting insights on the behavior of simple networks on the denoising problem. The authors provide a complete study of the univariate case when the noise level is reasonable, and give insights on what happens in higher dimensions. Numerical illustrations support the theoretical claims.

**Weaknesses:**

The study is limited to low noise levels (noisy samples supports can't intersect), and it would have been interesting to have the authors opinion on whether removing assumption 1 change or not the behavior shown in figure 1 (theoretical development or even empirical illustrations). In particular, theorem 1 is valid in the limit when $\sigma \to 0$, while we would have preferred a "threshold" (as the authors mention, there exists a threshold for the strict inequality but it is potentially very small).

**Questions:**

* Could the authors illustrate what happens in practice (in 1d or in higher dimension) when the noise level increases ? In particular, do we still have a gain using a NN rather than the eMMSE ?

* Is it possible to give bounds on $\sigma$ for which theorem 1 is valid ? Otherwise, could the authors provide MSE curves highlighting the difference between NN and eMMSE depending on $\sigma$ (in the case where the noisy samples support are disjoint and in the general case) ?

**Limitations:**

The limitations are addressed by the authors.

---

> ### Author Rebuttal · Authors · 2023-08-09
>
> __Answers to the questions:__
>
> __C:__  What happens in practice when the noise level increases?
>
> __R:__ Please see Figure 1 (in the pdf file attached to the 'global' rebuttal) for the MSE of NN denoiser and eMMSE vs $\sigma$ (the noise level). As can be seen from Figure 1, the NN denoiser performs better for noise levels in which we still have visible information in the noisy image.
>
> __C:__ Is it possible to give bounds on which theorem 1 is valid?
>
> __R:__ Given the probability density function of x (the clean image) we can calculate the critical noise level for which Theorem 1 holds. The critical noise level can change significantly depending on the probability density function of x. For example, if the probability density function has high “mass” in between the training points then the critical noise level is large. However, if the density function has low “mass” between the training points the critical noise level is small.
> Please see Figure 1 (in the PDF file) for the MSE curves. As can be seen the critical noise level in this case is large ($\sigma$ ~ 5).

---

> > ### Comment · Reviewer_7Tj7 · 2023-08-14
> > **Thanks for the answer**
> >
> > Thanks to the authors for their answer and the clarifications regarding the noise level.

---

### Official Review · Reviewer_kiMs · 2023-07-04

**Soundness:** 4 excellent
**Presentation:** 4 excellent
**Contribution:** 3 good
**Rating:** 7
**Confidence:** 3

**Summary:**

This paper studies how two-layer ReLU denoising networks look like when minimizing common losses such as the empirical minimum mean square error, an empirical alternative that draws finitely many noisy samples, as well as representation costs that find the data interpolating function with minimal norm of the weights. It is proven that - for univariate data - representation cost minimizers generalize better than empirical minimum mean square error minimizers in the low noise regime. Moreover, the solution to the representation costs is explicitly stated in the univariate case as well as in the multivariate case for specific geometric configurations, including one conjecture. Small numerical experiments illustrate that theory is correct and the conjecture is justified.

**Strengths:**

- The paper is very well written and easy to follow despite its theoretical nature
- It characterizes several interesting and important properties of shallow denoising networks in the low-noise regime for univariate data.
- It gives inspiring insights on the behavior in the multivariate case (under particular assumptions).
- It encourages further research in the investigated direction by posting one (partially open) conjecture, as well as by accurately stating several limitations (tight to very interesting directions of future research).
- It contributes to our understanding of denoisers in function space

**Weaknesses:**

I am not familiar enough with the explicit characterization of networks that minimize representation costs to judge the novelty and impact of the contribution. I was a little surprised to read "Hanin [2021] gave a full characterization of univariate representation cost minimizers subject to data interpolation constraints", and that there are follow-up extensions, which seem very relevant to the paper at hand.

Beyond this, except for the limitations that have already been stated by the authors themselves, I do not see any major weaknesses. In practical terms, I am now curious to what extent larger (deeper + more sophisticated, e.g. convolutional or normalization including) models inherit some of the shown properties, but can understand that this goes beyond the scope of the paper. Also, is the M<<d problem discussed in line 333 a reason why adversarial examples exist? The strict interpolation on a ball would prevent them.

**Questions:**

- Please clarify in what way your work extends previous characterizations of minimal representation cost solutions.
- And just out of curiosity - if "Savarese et al. [2019] showed that the representation cost of a function realizable as a univariate two-layer ReLU network coincides with the L1-norm of the second derivative of the function" (and assuming that the second derivative of a ReLU network has some meaning like the total variation of the (step-function-like) first derivative), doesn't proposition 1 follow from this quite directly?

**Limitations:**

The authors have addressed the limitations well.

---

> ### Author Rebuttal · Authors · 2023-08-09
>
> __Response to the weaknesses points:__
>
> __C:__ Do larger models inherit some of the shown properties?
>
> __R:__ There are several properties that we hypothesize to hold for larger models (but it goes beyond the scope of this paper). (1) We generalized Proposition 2 to the case of obtuse Simplex (See the “global” response). An interesting insight we deduce from our results on the obtuse simplex is that the NN denoiser (in this case) is bounded, which is a desired property for a denoiser. We hypothesize it is also true for larger models. (2) We proved that the minimizer is contractive towards the training points (for univariate data). This is akin to phenomenons shown empirically in diverse settings (multilayer Auto-Encoders, CNNs, and FCNs [A]). We hypothesize that larger models are locally contractive towards the training points.
>
> [A] Radhakrishnan, A., Yang, K., Belkin, M. and Uhler, C., 2018. Memorization in overparameterized autoencoders.
>
> __C:__ is the M<<d problem discussed in line 333 a reason why adversarial examples exist?
>
> __R:__ It is an interesting point. You are right, if the denoiser output is constant and equal to $x_n$ on a ball of radius $\rho$ around $x_n$, then the adversarial would be prevented altogether. It is an interesting future research topic to empirically verify if NN denoisers are more robust to adversarial attacks when we increase the number of noisy samples per image.
>
> __Answers to the questions:__
>
> __C:__ Please clarify in what way your work extends previous characterizations of minimal representation cost solutions.
>
> __R:__ For univariate data, the characterization in Hanin [2021] is possible because the representation cost reduces to the L1-norm of the 2nd derivative of the function, as shown in Savarese et al. [2019]. Since the 2nd derivative only acts locally, the minimum representation cost can be found by minimizing this quantity separately over intervals between data points. In the multivariate setting, the representation cost is more complicated, and involves the Radon transform of the function – a highly non-local operation – that complicates the analysis. Parhi and Nowak [2020] prove a representer theorem showing that there always exists a minimum representation cost interpolant realizable as a shallow ReLU network with finitely many neurons, and Ergen and Pilanci [2021] give an implicit characterization of representation cost minimizers as the solution to a convex optimization problem. However, to the best of our knowledge, there are no results in the literature explicitly characterizing representation cost minimizers in the case of multivariate inputs, even for networks having scalar outputs. Therefore, for this paper we had to develop new tools and approximations. To simplify the problem, we assume norm-ball interpolation constraints (Equation (19) in the paper), in place of finite noisy realizations. This type of approximation is novel, and allows us to give explicit characterizations of representation cost minimizers under specific geometric assumptions on the training points. Thank you for your comment, we will add this discussion to the revised paper.
>
> __C:__ Doesn't proposition 1 follow from Savarese et al. [2019] directly?
>
> __R:__ Savarese et al. [2019] considered the case of shallow ReLU networks with an unbounded number of neurons. In Proposition 1 we found an explicit form to the minimizer of the representation cost with a __finite__ amount of neurons. In addition, Savarese et al. [2019] consider the case of one hidden layer ReLU network __without__ a skip connection. Lastly, in the case of denoising, Hanin’s [2021] result guarantees a unique minimizer for the representation cost.

---

> > ### Comment · Reviewer_kiMs · 2023-08-16
> > **Thanks**
> >
> > Thanks a lot for the detailed response and the additional results which further strengthen the paper! I clearly recommend the acceptance of this work!

---

### Official Review · Reviewer_qSF7 · 2023-07-06

**Soundness:** 3 good
**Presentation:** 2 fair
**Contribution:** 2 fair
**Rating:** 5
**Confidence:** 3

**Summary:**

The elementary properties of NN solutions for the denoising problem have been explored, with a focus on offline training of a one hidden-layer ReLU network. When the noisy clusters of the data samples are well-separated, there exist multiple networks with zero loss, even in the case of under-parameterization, while having a different representation cost.  In the univariate case, a closed-form solution to such global minima with minimum representation cost has been derived. It has also been demonstrated that the univariate NN solution generalizes better than the eMMSE denoiser. In the multivariate case, it has been shown that the interpolating solution with minimal representation cost is aligned with the edges and/or faces connecting the clean data points in several basic cases.

**Strengths:**

The NN solutions is studied in the setting of interpolation of noisy samples with minimal representation cost, in a practically relevant “low noise regime” where the noisy samples are well clustered.

In the univariate case, a closed-form solution for the minimal representation cost NN denoiser is derived and shown to have better generalizition behavior than the empirical minimum MSE  denoiser.

In the multivariate case, a closed-form solution for the minimal representation cost NN denoiser in multivariate case under various assumptions on the geometric configuration of the clean training samples. Moreover,  a general alignment phenomenon of minimal representation cost NN denoisers is illustrated in the multivariate setting.

**Weaknesses:**

Weakness 1: Empirical effectiveness.  This paper proposes a shallow denoiser and analyizes its theoretical performance.

W1.1 Despite the interesting theoretical properties of the shallow denoiser, it is not clear whether the proposed model performs well in real image denoising applications due to the lack of sufficient empirical evaluation and analysis.

W1.2 In the multi-variate settings, the authors consider training data on a subspace. However, it is not clarified in which typical settings this subspace assumption holds. Thus, the empirical reasonability of the subspace-based analysis is not well addressed.

Weakness 2: The theoretical findings may be not sufficiently new. For example, the NN solution is shown contractive towards the clean data points, which has already been empirically observed in Autoencoders by Radhakrishnan et al.  [2018].


**Questions:**

Please see and address the weaknesses.

**Limitations:**

I think the authors have well discussed the limitations.

---

> ### Author Rebuttal · Authors · 2023-08-09
>
> __Response to the weaknesses points:__
>
> __Weakness 1.1:__
>
> __C:__  Is the proposed model perform well in real image-denoising applications?
>
> __R:__ Practically successful image-denoising architectures are deep and not fully connected. On such architectures, it is very challenging to obtain any theoretical guarantee without very strong assumptions (e.g. working in a linearized regime, such as the neural tangent kernel), especially before understanding simpler architectures. Therefore, deep learning theory papers researching a new question usually start with the simplest relevant model, as we do here. Specifically, in this paper, we have focused on a shallow NN denoiser, which is both simple enough to start a theoretical inquiry, yet rich enough to provide insightful conclusions about modern denoisers architectures.
>
> One such insight that we proved is that the minimizer is contractive towards the training points (for univariate data). This is akin to phenomenons shown empirically in diverse settings (multilayer Auto-Encoders, CNNs, and FCNs [C]). It is an interesting future research topic to empirically verify that practical image denoisers are also locally contractive toward the training points (Please see our answer to Weakness 2 for an additional explanation regarding the novelty of this result).
>
> Another interesting insight is that the NN denoiser is bounded, which is a desired property for a denoiser. This property hold in all our results, and also in the generalized Proposition 2 to the case of obtuse Simplex (See the “global” response), which provides an explicit solution to a high-dimensional case.
>
>
> __Weakness 1.2:__
>
> __C:__  In which typical settings this subspace assumption holds?
>
> __R:__ In general, natural images are commonly assumed to lie on a low-dimensional nonlinear manifold [A]. There are several applications where the images are assumed to lie on a subspace (linear manifold). For example, in face recognition and handwritten digits, classical algorithms use PCA and achieve good results, which indicates that the data lies on a linear subspace.
>
> __Weakness 2:__
>
> __C:__ The NN solution is shown contractive towards the clean data points, which has already been empirically observed in Autoencoders.
>
> __R:__ Our results differ from [B] in the following:
>
> * [B] focused on __empirically__ showing contractivity, while we __proved__ it. We believe there is value in proving important empirical observations, especially in deep learning, where the lack of established theory is a well-recognized problem.
>
> * [B] also proved that 2-layer auto-encoder models are contractive, but under rather __unrealistic__ assumptions: (1) the weights of the input layer are fixed and (2) the number of neurons goes to infinity. In contrast, in our theoretical results in the univariate case, the minimizer of the representation cost is contractive towards the training samples without these assumptions (i.e., the minimizer optimizes over both layers and has a finite number of neurons).
>
> * [B] used a weaker definition of contraction. Specifically, [B] only showed __local__ contraction, based on the eigenvalues of the Jacobian (linearization of a dynamical system). In contrast, we show __global__ contraction towards the training samples (the word “locally” in definition 2 in the submitted paper is a typo).
>
> * [B] examined auto-encoder models, while we examine denoisers models (i.e., auto-encoders trained with noisy inputs).
>
> We will clarify this in the revised paper.
>
> In addition, we have several new theoretical findings. For example,
>
> * we prove Generalization results for the univariate case (better generalization than the eMMSE).
>
> * A closed-form solution for the minimizer of the representation cost __in the multivariate case__. As far as we know, there are no such results in the literature, even for networks having scalar outputs.
>
> [A] Pope, P., Zhu, C., Abdelkader, A., Goldblum, M. and Goldstein, T., 2021. The intrinsic dimension of images and its impact on learning.
>
> [B] Radhakrishnan, A., Yang, K., Belkin, M. and Uhler, C., 2018. Memorization in overparameterized autoencoders.

---

> > ### Comment · Reviewer_qSF7 · 2023-08-14
> > **Response to the reply**
> >
> > Thanks for the authors' feedback.
> > While addressing my weakness 1.2 for the empirical reasonability of the subspace-based analysis, the authors give examples in face recognition and handwritten digits, where PCA achieves good performance. However, it is more convincing to analyze the subspace assumption on large scale datasets today like ImageNet.

---

> > > ### Author Response · Authors · 2023-08-17
> > >
> > > As shown in [A], it is a general phenomenon that large datasets are (approximately) low rank [A], i.e., lie on a linear subsapce.
> > >
> > > Following the reviewer׳s suggestion, we also validated the subspace assumption on the following common image datasets:
> > > * CIFAR10
> > > * CINIC10
> > > * Tiny ImageNet (a lower resolution version of ImageNet, enabling us to use SVD)
> > > * BSD (a denoising benchmark composed from 128X1600 patches of size 40X40 cropped from 400 images [B])
> > >
> > > We applied a Singular Value Decomposition (SVD) for each of the above datasets, and calculated the relative number of Singular Values (SV) needed to achieve a given percentile of the energy (for the average vector).
> > >
> > > __CIFAR10__:  95%, 99%, and 99.9% of the energy is concentrated on 0.8%, 7.5%, and 30% of the SV, respectively.
> > >
> > > __CINIC10__: 95%, 99%, and 99.9% of the energy is concentrated on 1%, 23%, and 41% of the SV, respectively.
> > >
> > > __Tiny ImageNet__: 95%, 99%, and 99.9% of the energy is concentrated on 1.6%, 20%, and 36% of the SV, respectively.
> > >
> > > __BSD__:  95%, 99%, and 99.9% of the energy is concentrated on 0.1%, 1.6%, and 4.5% of the SV, respectively.
> > >
> > > As can be seen from the results, the subspace assumption holds for all the datasets we used. We hope we were now able to completely address all the reviewer's concerns, if there are any remaining concerns, please let us know.
> > >
> > > [A] Udell, M. and Townsend, A., 2019. Why are big data matrices approximately low rank?.
> > >
> > > [B] Zhang, K., Zuo, W., Chen, Y., Meng, D. and Zhang, L., 2017. Beyond a gaussian denoiser: Residual learning of deep cnn for image denoising.

---

> > > > ### Comment · Reviewer_qSF7 · 2023-08-21
> > > >
> > > > Many thanks for the authors' feedback. I would like to keep my,score unchanged.

---

### Official Review · Reviewer_4XGM · 2023-07-06

**Soundness:** 3 good
**Presentation:** 3 good
**Contribution:** 2 fair
**Rating:** 6
**Confidence:** 3

**Summary:**

This paper looks at the denoising problem and compares the neural network denoiser to what the paper calls the eMMSE denoiser. The eMMSE denoiser is the optimal denoiser (in function space) given finite data and known noise distribution. In contrast, the Neural Network denoiser has access to a finite number of copies of each of the finite data points and minimizes the empirical noise.

The paper shows that even in simple settings (univariate data), the minimum norm neural network denoiser has better generalization than the eMMSE denoiser, as it interpolates in between the data points whereas the eMMSE denoiser acts as a nearest neighbor denoiser. The paper further explores some other settings such as low rank and data on rays.



**Strengths:**

The main strength of the paper is finding the explicit form of the minimum norm two-layer denoiser. This is interesting as it identifies the best point in function space that can be represented as a two-layer network. Hence gives the global minimizer for the neural network which is traditionally not easy. Hence this is important.

Other strengths of the paper include that they show these denoisers are contracting and generalize better than some "optimal" denoisers.

**Weaknesses:**

The are a few weaknesses of the paper.

1) I think the presentation of the paper can be improved significantly. In many cases, the paper switches between functional representations of the function and the parametric representation of the function. While both are interesting, it would be nice to have to a clear distinction between the two and a way of translating from one representation to the other.

2) While the results of the paper are interesting, there are limitations in the types of data that they consider. Specifically, univariate data, data on a line, and data on the union of lines are the cases in which the paper can identify the min norm solutions. While these cases are interesting and I understand that more general cases are challenging. Hence I don't think it is needed to solve more general cases. However, if the paper could extract some insights from these theoretical cases that might apply to more general cases, this would help strengthen the paper.

3) I think the related works section is missing various theoretical works on denoising. [A-D] look at denoising via factorizations which fit in nicely with the low-rank structure that the paper looks at, and [E] looks at denoising regression case.


[A] Raj R. Nadakuditi. OptShrink: An Algorithm for Improved Low-Rank Signal Matrix Denoising by Optimal, Data-Driven Singular Value Shrinkage. IEEE Transactions on Information Theory, 2014

[B] Marc Lelarge and Léo Miolane. Fundamental Limits of Symmetric Low-Rank Matrix Estimation. In Proceedings of the 2017 Conference on Learning Theory, 2017

[C] Antoine Maillard, Florent Krzakala, Marc Mézard, and Lenka Zdeborová. Perturbative Construction of Mean-Field Equations in Extensive-Rank Matrix Factorization and Denoising. Journal of Statistical Mechanics: Theory and Experiment, 2022

[D] Emanuele Troiani, Vittorio Erba, Florent Krzakala, Antoine Maillard, and Lenka Zdeborov’a. Optimal Denoising of Rotationally Invariant Rectangular Matrices. ArXiv, abs/2203.07752, 2022

[E] Rishi Sonthalia and Raj Rao Nadakuditi. Training data size induced double descent for denoising feedforward neural networks and the role of training noise. Transactions on Machine Learning Research, 2023



**Questions:**

1. The paper presents the network with a skip connection, but as far as I can tell, the skip connection version is not studied in the paper. As $R(f)$ (Definition 1, Equation 19) is without the skip connection.

2. The denoiser in Corollary 1, what would be the neural network representation of that function with ReLU activation?

3. The fact that the online denoiser and the offline are equivalent is interesting. Do you have any intuition as to why the offline captures the online phenomena despite having much fewer samples?

---

> ### Author Rebuttal · Authors · 2023-08-09
>
> __Response to the weaknesses points:__
>
> __C: (1)__ the paper switches between functional representations and the parametric representation
>
> __R:__ We are not sure what is the source of the confusion. For parametric representation we only use $h_{\theta}$, as defined in Equation 6 in Section 2. For function space representation only use $f$, as defined in Definition 1 in Section 3 (and we focus on $f$ from that point onwards). This was also the standard notation in previous works (e.g. Savarese et al.).
>
> __C: (2)__ insights from the theoretical results which apply to a general case
>
> __R:__ We generalized Proposition 2 to the case of obtuse Simplex (See the “global” response), and this provides an explicit solution to a high-dimensional case. We believe that this result will facilitate deriving generalization guarantees of NN denoisers. One interesting insight we deduce from our results on the obtuse simplex is that the NN denoiser (in this case) is bounded, which is a desired property for a denoiser.
>
> __C:__ (3) The related works section is missing various theoretical works on denoising
>
> __R:__ The papers [A]-[D] indeed also consider the denoising problem with a low-rank prior on the signal, but mostly focus on optimal MMSE denoising, and do not consider neural networks. Our main focus is the structural properties of neural network denoisers because this sheds light on their generalization properties. As denoising is a very large topic, we have simply referred the reader to general surveys on the denoising problem (e.g., the one by Elad et al. 2023) in order to keep the paper focused. The paper [E] is indeed relevant to neural networks, and we will add a discussion on the results obtained in this paper on the double-descent phenomenon in a rank-1 denoising setup in the revised version.
>
>
> __Answers to the questions:__
>
> __C: (1)__ The skip connection version is not studied in the paper
>
> __R:__ There appears to be some misunderstanding. As stated in Definition 1, the representation cost is defined for $h_{\theta}$ as defined in Equation 6, which is a shallow ReLU network __with a skip connection__. All later results find minimizers of $R(f)$ as defined in Definition 1, and therefore also assume shallow ReLU networks with a skip connection.
>
> __C: (2)__  What is the neural network representation of the denoiser in Corollary 1?
>
> __R:__ Suppose the collinear points $x_n = c_n u$  are ordered such that $c_1 < c_2 < … < c_N$. Then a neural network representation of the denoiser in Corollary 1 is:
>
> $f^*(y) = \sum_{n=1}^{N-1} a_n u ([u^T y - (c_n + \rho)]\_+ - [u^T y - (c_{n+1} - \rho)]\_+)$
>
> where
>
> $a_n = (c_{n+1}-c_n)/(c_{n+1}-c_n-2\rho)$ and $\rho$ is the radius of the norm-ball constraints (i.e., we assume the denoiser maps every point in the ball of radius $\rho$ centered at $x_n$ to the point $x_n$).
>
> __C: (3)__  An intuition as to why the offline captures the online?
>
> __R:__ On an intuitive level, the cause is the exponential tail of the Gaussian distribution. Given that the number of noisy samples per image in the offline setting is large enough, we need exponentially more noisy samples per image in the online setting in order to get a different solution.

---

> > ### Comment · Reviewer_4XGM · 2023-08-12
> > **Thank you for the clarifications.**
> >
> > I will increase my score. I did have one more question.
> >
> > I think why I thought (13) did not have the skip connection ($V$) is because the norm of the skip connection is not regularized. Is there a reason this is the case? Do the authors know what might happen if $V$ included in $R(f)$?

---

> > > ### Author Response · Authors · 2023-08-16
> > >
> > > Thank you for increasing the score.
> > >
> > > We followed the same setting as in [A] and [B], which did not regularize the skip connection. Previous works also considered other cases: without skip connection ([A] Theorem 2), and regularization on the bias [C].
> > > Adding regularization over the skip connection is an interesting case to consider, which was not directly covered in previous works: [D] and [E] proved related results but in a different setting (constrained path/optimization path in classification vs. regularization path in regression in our case). From these works, it intuitively appears that a “large” target function is “cheaper” to implement without a skip connection (recovering the case without the skip connection), since we can distribute the function scale over two layers instead of one. However, for general target functions the situation is more complicated. We can prove (see below) that the cost of realizing the linear function $L(x) = Vx$ with the regularized skip unit is greater than the cost of realizing it with regularized ReLU's if and only if $2||V||\_* \leq ||V||\_F^2$, where $||V||\_*$ is the nuclear norm. This will generally hold for large norm matrices (following the intuition above), but it fails for matrices with sufficiently small norms. We are not sure yet what holds for general (nonlinear) target functions. This is an interesting open question to explore.
> > >
> > > __Proof:__
> > >
> > > Let $C\_{relu}$ be the minimum cost needed to realize $L(x) = Vx$ with ReLU units over some compact domain.
> > > We can write every such realization as
> > >
> > >  $L(x) = A[W^Tx+b]\_+ -Ab$,
> > >
> > > where $[W^Tx+b]\_+ = W^Tx + b$ on the domain, and  $V = AW^T$.
> > >
> > > This gives
> > >
> > > $C\_{relu} = min\_{V = AW^T} \sum_k (||A||\_F^2 +  ||W||\_F^2)$.
> > >
> > > But by the variational characterization of the nuclear norm, we see that $C\_{relu} = 2||V||\_*$.
> > >
> > > On the other hand, to realize $L(x)$ with a regularized skip connection costs $C\_{skip} = ||V||\_F^2$.
> > > So as long as $2||V||\_* \leq ||V||\_F^2$, we have $C\_{relu} \leq C\_{skip}$.
> > >
> > > [A] Ongie, G., Willett, R., Soudry, D. and Srebro, N., 2019. A function space view of bounded norm infinite width relu nets: The multivariate case.
> > >
> > > [B] Hanin, B., 2021. Ridgeless Interpolation with Shallow ReLU Networks in $1 D $ is Nearest Neighbor Curvature Extrapolation and Provably Generalizes on Lipschitz Functions.
> > >
> > > [C] Boursier, E. and Flammarion, N., 2023. Penalising the biases in norm regularisation enforces sparsity.
> > >
> > > [D] Nacson, M.S., Gunasekar, S., Lee, J., Srebro, N. and Soudry, D., 2019, May. Lexicographic and depth-sensitive margins in homogeneous and non-homogeneous deep models.
> > >
> > > [E] Kunin, D., Yamamura, A., Ma, C. and Ganguli, S., 2022. The asymmetric maximum margin bias of quasi-homogeneous neural networks.

---

### Author Rebuttal · Authors · 2023-08-09

We thank the reviewers for the helpful feedback and remarks, and for your interest in the paper. We have addressed all of them, and we will revise the paper, ‌as detailed below. The rebuttal is in comment-response (C/R) format.

Also, ‌after the initial submission we have strengthened some of our results:

1) In Theorem 3, we have removed the restriction $L \le 3$. In other words, we generalized it to an arbitrary number of $L$ rays making obtuse angles with each other.

2) Using the above result, we generalized Proposition 2 to the case of an obtuse simplex (i.e. one point in the training data that forms an obtuse angle with all other points).

3) We proved Conjecture 1 in the special case that the training set forms an equilateral triangle.

We believe these results will further strengthen the paper, and therefore plan to add them to the final version (unless there is any objection).

---

### Decision · Program_Chairs · 2023-09-21

**Decision:**

Accept (poster)

**Comment:**

The paper studies the structure of neural network denoisers. It provides closed form solutions for optimal neural network denoisers in two settings (i) for univariate data, and (ii) for multivariate data with structure. In particular, for univariate data it gives an expression for the optimal two layer denoiser in the interpolation (zero training error) regime with weight decay, and proves that for small noise this denoiser has better generalization error the eMMSE denoiser (nearest neighbor).

 Reviewers expressed appreciation for the paper’s theoretical results -- in particular, for its closed form expression for the optimal denoiser,  and for identifying scenarios where neural nets outperform empirical MMSE. Points of discussion include technical limitations of the results (small noise), and the limited effectiveness of shallow denoisers in practice. The latter is less of a direct concern for this paper’s theoretical contributions, although it would be useful to have theoretical insights into the structure of deep denoisers. After considering author responses, reviewers converged to a recommendation to accept. Denoising is important in its own right, and a critical building block for generative models and inverse problem solvers. The paper provides clear theoretical insights, and raises some conjectures which could inspire future work.